# A dynamic, spatially periodic, micro-pattern of HES5 underlies neurogenesis in the mouse spinal cord

Veronica Biga[1],[†] (iD), Joshua Hawley[1],[†] (iD), Ximena Soto[1] (iD), Emma Johns[1] (iD), Daniel Han[2] (iD), Hayley Bennett[1], Antony D Adamson[1] (iD), Jochen Kursawe[3] (iD), Paul Glendinning[2], Cerys S Manning[1],[*],[†],[‡] (iD) & Nancy Papalopulu[1],[**],[†],[‡] (iD)

## Abstract

Ultradian oscillations of HES Transcription Factors (TFs) at the single-cell level enable cell state transitions. However, the tissue-level organisation of HES5 dynamics in neurogenesis is unknown. Here, we analyse the expression of HES5 *ex vivo* in the developing mouse ventral spinal cord and identify microclusters of 4–6 cells with positively correlated HES5 level and ultradian dynamics. These microclusters are spatially periodic along the dorsoventral axis and temporally dynamic, alternating between high and low expression with a supra-ultradian persistence time. We show that Notch signalling is required for temporal dynamics but not the spatial periodicity of HES5. Few Neurogenin 2 cells are observed per cluster, irrespective of high or low state, suggesting that the microcluster organisation of HES5 enables the stable selection of differentiating cells. Computational modelling predicts that different cell coupling strengths underlie the HES5 spatial patterns and rate of differentiation, which is consistent with comparison between the motoneuron and interneuron progenitor domains. Our work shows a previously unrecognised spatiotemporal organisation of neurogenesis, emergent at the tissue level from the synthesis of single-cell dynamics.

**Keywords** Hes5; neurogenesis; notch; oscillations; patterning

**Subject Categories** Development; Neuroscience

**Mol Syst Biol. (2021) 17: e9902**

## Introduction

Neurogenesis is the developmental process which generates the variety of neuronal cell types that mediate the function of the nervous system. Neurogenesis takes place over a period of days during mouse embryogenesis; thus, the transition from progenitor maintenance to differentiation needs to be balanced for development to occur normally. Neurogenesis relies on the integration of positional information with the transcriptional programme of neuronal differentiation. In the spinal cord, notable progress has been made in understanding the role and regulation of the dorsoventral (D-V) positional system, that relies on secreted morphogens and transcriptional networks to generate the stereotyped array of different types of neurons along this axis (Briscoe & Small, 2015; Sagner & Briscoe, 2019). The transcriptional programme that mediates neurogenesis is also well understood in the spinal cord, particularly with the application of single-cell sequencing (Paridaen & Huttner, 2014; Delile *et al*, 2019; Sagner & Briscoe, 2019).

Recent live imaging studies of cell fate decisions during neurogenesis have added a new dimension to this knowledge (Vilas-Boas *et al*, 2011; Das & Storey, 2012, 2014; Manning *et al*, 2019; Nelson *et al*, 2020; Soto *et al*, 2020). They have shown the importance of understanding transcription factor (TF) expression dynamics in real time, including the key transcriptional basic helix–loop–helix repressors Hairy and enhancer of split (HES)1 and 5 (Ohtsuka *et al*, 1999; Imayoshi & Kageyama, 2014; Bansod *et al*, 2017), in regulating state transitions. We have previously shown that in spinal cord tissue, HES5 exhibits ultradian periodicity of 3–4 h in about half of the progenitor population with the remaining progenitors showing aperiodic fluctuations (Manning *et al*, 2019). The percentage of cells that show oscillations rises in cells that enter the differentiation pathway; such cells show a transient phase of more coherent oscillations before the level of HES5 is downregulated in differentiated cells (Manning *et al*, 2019). Furthermore, our studies of a zebrafish paralogue Her6 showed that the transition from aperiodic to oscillatory expression is needed for neuronal differentiation, suggesting that oscillatory expression has an enabling role for cell state

1  Faculty of Biology Medicine and Health, The University of Manchester, Manchester, UK
2  Department of Mathematics, School of Natural Sciences, Faculty of Science and Engineering, The University of Manchester, Manchester, UK
3  School of Mathematics and Statistics, University of St Andrews, St Andrews, UK
  *Corresponding author. Tel: +44 161 2757221; E-mail: cerys.manning@manchester.ac.uk
  **Corresponding author. Tel: +44 161 3068907; E-mail: Nancy.Papalopulu@manchester.ac.uk
  †These authors contributed equally to this work
  ‡These authors contributed equally to this work as co-corresponding authors

transitions (Soto *et al,* 2020) as we have previously predicted computationally (Bonev *et al,* 2012; Goodfellow *et al,* 2014; Phillips *et al,* 2016).

Although these studies revealed an unappreciated dynamic behaviour at the level of HES TF protein expression, these live imaging studies are based on recording dynamics from sparsely distributed single cells in the tissue context. Therefore, little is known about how single-cell dynamics are synthesised to tissue-level dynamics. Do cells interact with their neighbours in order to coordinate their cell state transitions and if so, how and what is the mechanism?

Notch is of particular interest in this context because it is a highly conserved cell-to-cell signalling pathway that is well known for generating complex spatial patterns of cell fates in tissue development (Cohen *et al,* 2010; Shaya & Sprinzak, 2011; Hunter *et al,* 2016; Corson *et al,* 2017; Henrique & Schweisguth, 2019). Activation of Notch receptors by Notch ligands, including DLL1 and JAG1, results in downstream expression of HES1 and HES5. HES TFs can influence Notch activity on neighbouring cells by repressing Notch ligand expression either directly (Kobayashi *et al,* 2009; preprint: de Lichtenberg *et al,* 2018) or indirectly through the repression of proneural TFs such as Neurogenin1/2 (NGN1/2) (Ma *et al,* 1998). We argue that in order to understand how the balance of HES progenitor factors can be tipped in favour of proneural factors giving rise to a decision point in neural progenitor cells, we need to address tissue-level patterns of HES expression and use computational models that can integrate the complexity of interactions at multiple scales.

The effects of Notch–Delta signalling combined with HES oscillations have been investigated during somitogenesis. Live imaging of dissociated PSM cells *in vitro* has shown that single-cell oscillators can self-organise through Notch-dependent synchronisation to generate waves in gene expression similar to those observed *in vivo* (Tsiairis & Aulehla, 2016). A model of mRNA and protein production and self-repression with transcriptional delay explains the emergence of autonomous oscillations of Her1 and Her7 as well as synchronisation by Notch activity observed during the formation of somites (Lewis, 2003; Özbudak & Lewis, 2008; Webb *et al,* 2016). A more abstract Kuramoto-style model with time delays explains how a population of initially asynchronous and autonomous oscillators can evolve to adopt the same frequency and phase in order to periodically form somites (Morelli *et al,* 2009; Oates, 2020). The period of the oscillations determines the size of the somite and Notch abundance controls dynamic parameters such as the time to synchronisation (Herrgen *et al,* 2010). Apart from a limited number of studies suggesting an anti-phase relationship of DLL1 oscillations in neighbouring neural cells (Shimojo *et al,* 2016), whether and how neural progenitor cells coordinate fate decisions and dynamic HES activity with their neighbours remains unknown.

In this study, we observe spatially periodic HES5 micro-patterns which are generated through positive correlations in the levels of HES5 between neighbouring cells and by local synchronisation of low coherence single-cell oscillators present in spinal cord tissue. These patterns are maintained in a dynamic way through Notch mediated cell–cell interactions. A computational model predicts that coupling strength changes spatial patterns of expression and, in turn, the probability of progenitor differentiation. We confirm that between adjacent progenitor domains in the spinal cord, the rate of differentiation correlates with spatial patterns of HES5 and cell–cell coupling strength. Thus, organisation of neural progenitors in HES5

phase-synchronised and level-matched progenitors is an exquisite spatiotemporal mechanism conferring tissue-level regulation of the transition of single cells from neural progenitor to neuron.

# Results

## Positive correlations in Venus::HES5 intensity are indicative of microclusters in spinal cord tissue

Within the peak of spinal cord neurogenesis (E9.5–E11.5), HES5 is expressed in two broad domains in the dorsal and ventral embryonic mouse spinal cord (Sagner *et al,* 2018; Manning *et al,* 2019). Previously, we have characterised the single-cell dynamic behaviour of the more ventral HES5 expression domain that covers the ventral interneuron (p0–p2) and motorneuron progenitors (pMN) (Manning *et al,* 2019). Thus, to understand how the single-cell expression dynamics contributes to tissue-level behaviour we have focussed here on the same ventral area of HES5 expression (Figs 1 and EV1a). In this area, all progenitor cells (marked by SOX2) show HES5 expression (Fig EV1B). To characterise the spatial pattern of HES5 protein expression in this progenitor domain, we made *ex vivo* slices of E9.5-E11.5 Venus::HES5 knock-in mouse embryo spinal cord (Imayoshi *et al,* 2013). In snapshot images of this domain, we noticed multiple local clusters of neural progenitor cells with similar levels of nuclear HES5 (Fig 1A) which we refer to as "microclusters". These are notable after manual segmentation using a Draq5 live nuclear stain and averaging HES5 intensity across the nucleus (Fig 1A–C). The differences in Venus::HES5 intensity between nuclei did not correlate with the Draq5 nuclear staining indicating this was not related to global effects or effects of imaging through tissue (Fig EV1C). By measuring the number of nuclei in microclusters with high Venus::HES5 levels (see Materials and Methods "Microcluster quantification"), we found that they consisted of 3–4 cells wide in the apical-basal (A-B) and 2–4 cells wide in the dorsoventral (D-V) direction (3–7 cells in total, Fig EV1D) and were similar in size between E9.5 and E11.5 (Fig 1D). Randomisation controls of the nuclear intensities showed that microclusters were significantly larger than is expected by chance (Fig EV1D and Materials and Methods). Consistent with the presence of microclusters of cells with similar levels, nuclei showed a positive correlation in Venus::HES5 between close neighbours that drops with increasing neighbour number (Fig 1E). We took a more quantitative approach and correlated mean nuclear HES5 levels between all pairs of nuclei and found that nuclei close to each other were highly positively correlated and this correlation dropped with increasing distance, becoming negative at distances over 50 μm (Fig 1F). This relationship was similar across E9.5–E11.5 (Fig EV1E) and substantially different to the correlation coefficients calculated from randomisations of the nuclei intensities but keeping the same nuclear spatial arrangement (Fig 1F) which indicates the presence of a pattern in HES5 levels.

The longer-range negative correlations may arise from gradients in HES5 expression in A-B and D-V direction. Indeed, the images indicate the presence of a radial gradient emanating from an area of highly expressing cells (Fig 1G and H, Fig EV1F and Appendix Fig S1A). Such a radial gradient could be due to the downregulation of HES5 as cells differentiate and move basally from the progenitor

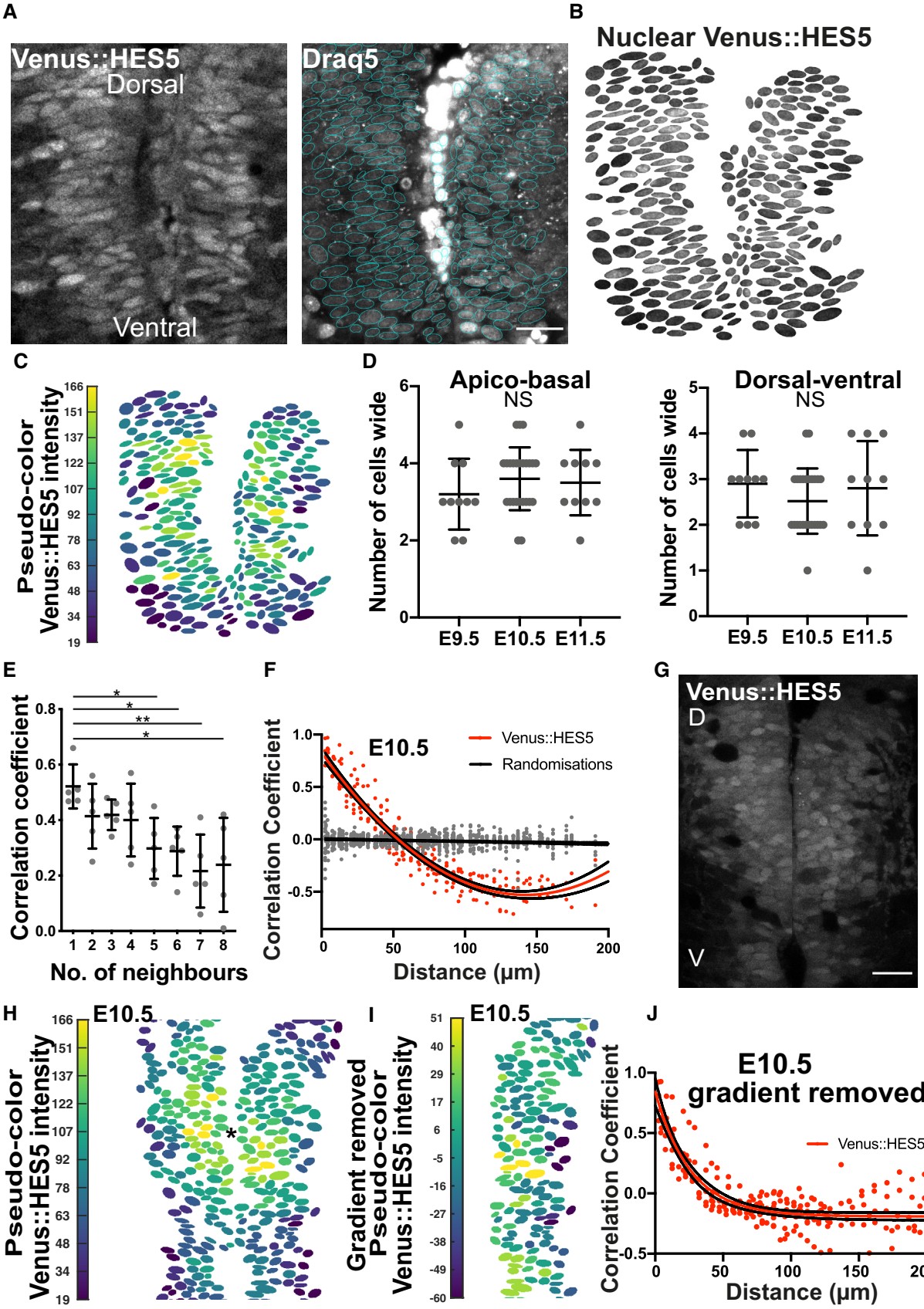

Figure 1.

**Figure 1.  Microclusters of spinal cord neural progenitor cells have positively correlated HES5 levels.**

A   Transverse slice of live E10.5 Venus::HES5 homozygous knock-in mouse showing the ventral HES5 domain in spinal cord *ex vivo* (left panel); Draq5 live nuclear stain with nuclear segmentation overlay (right panel); scale bar 30 μm.

B   Venus::HES5 nuclear signal corresponding to tissue in (A) obtained by applying nuclear segmentation onto Venus channel.

C   Pseudo-color look-up table applied to mean nuclear Venus::HES5 intensity (Materials and Methods) corresponding to segmented image in (B).

D   Dimensions of microclusters in cell numbers with high and similar levels of HES5 in apical–basal axis (left panel) and dorsoventral axis (right panel) at E9.5 (10 microclusters, 3 slices, 3 exps), E10.5 (10 microclusters, 9 slices, 3 exps) and E11.5 (10 microclusters, 3 slices, 3 exps). NS—no significant difference in one-way ANOVA $P = 0.46$ (A-B), $P = 0.38$ (D-V). Bars show mean $\pm$ SD.

E   Pearson correlation coefficient observed in segmented E10.5 homozygous Venus::HES5 spinal cord *ex vivo* slices showing correlation between mean nuclear Venus:: HES5 intensity in any cell compared with up to eight nearest neighbours (see Materials and Methods); dots indicate average per slice; bars indicate mean and standard deviation of five slices from three experiments (data set is different from (D)).

F   Pearson correlation coefficient of mean nuclear Venus::HES5 intensity in relationship to distance; red dots indicate average Venus::HES5 correlation per slice of 12 slices from three experiments with corresponding red line indicating polynomial fit (order 2); grey dots with black line indicate correlations and polynomial fit from five randomisations of intensities analysed in the same way (see Materials and Methods).

G   Transverse slice of live E10.5 Venus::HES5 homozygous knock-in mouse showing the ventral HES5 domain in spinal cord *ex vivo*. Scale bar 30 μm, D—dorsal, V— ventral.

H   Pseudo-color look-up table applied to mean nuclear Venus::HES5 intensity of (G); centre of intensity shown with *.

I    Pseudo-color look-up table applied to mean nuclear Venus::HES5 intensity in (H) (only one side of ventricle) after radial gradient removal (see Materials and Methods).

J    Pearsons correlation coefficient of mean nuclear Venus::HES5 intensity with distance after subtraction of radial gradient in Venus::HES5 intensity; red dots represent average in each of 12 slices from three experiments.

Source data are available online for this figure.

domain as well as to D-V differences in the level of expression (see below) and is not further investigated here. To ask whether the local positive correlations in HES5 levels are an artefact of this larger-scale domain expression pattern, we measured and subsequently removed a radial gradient across the tissue from the segmented single-cell images (see Materials and Methods). However even after removing a radial gradient, mean nuclear HES5 levels at E9.5–E11.5 remained highly positively correlated at distances less than 40–50 μm (Figs 1I and J, and EV1G and H). Therefore, a global tissue gradient of HES5 cannot fully explain the detailed spatial pattern and further factors, such as microclusters of cells with similar HES5 levels, must contribute to the formation of the HES5 spatial pattern.

## HES5 microclusters are spatially periodic along dorsoventral axis of spinal cord

The high-resolution analysis of single-cell snapshots showed the presence of multiple microclusters in HES5 expression in the ventral domain. Next, we asked whether these microclusters have a regular spatial arrangement. To do this, we drew line profiles 15 μm wide, parallel to the ventricle, in the ventral to dorsal direction (Figs 2A and EV2A and B) and plotted the Venus::HES5 intensity along this line (Fig 2B) from lower resolution 20× images of *ex vivo* slice cultures. Throughout the paper, the 0 distance is the ventral-most point of the HES5 domain, and distance extends dorsally (Materials and Methods). Detrending the signal removed a bell-shaped curve

**Figure 2.  HES5 microclusters are spatially periodic along the dorsal–ventral axis of spinal cord.**

A   20x snapshot of an *ex vivo* slice culture of E10.5 spinal cord from Venus::HES5 heterozygous knock-in mouse, transverse section; delineated region (blue) correspond to data shown in (B, C). D—dorsal, V—ventral.

B   Spatial profile of Venus::HES5 intensity averaged over 2.5 h with 0 distance representing the ventral end of kymograph; black line represents the trend in Venus::HES5 data across the domain produced using an polynomial order 6 (see Materials and Methods).

C   Detrended spatial profile of Venus::HES5 corresponding to data shown in (B).

D   Lomb-Scargle Periodogram analysis of detrended Venus::HES5 data in (C); horizontal line indicates Lomb-Scargle significance level $P = 0.0001$; red arrowhead indicate significant peaks.

E   Auto-correlation analysis of detrended Venus::HES5 spatial profile in (C) with multiple peaks indicating spatial periodicity; significant peaks (red arrowhead) lie outside grey area indicating 95% significance based on bootstrap approach (see Materials and Methods) and non-significant peaks (black arrowhead).

F   Peak to peak distance in auto-correlation from detrended Venus::HES5 signal collected in apical regions of spinal cord between E9.5-E11.5; bars indicate mean and SD of individual slices from three independent experiments; Kruskal–Wallis test not significant, $P = 0.44$.

G   Representative example of auto-correlation from detrended Draq5 nuclear signal with peak to peak distances indicative of inter-nuclear distance in live tissue; grey area denotes 95% confidence area for Draq5.

H   Peak to peak distance in auto-correlation of detrended Draq5 spatial profile in apical regions of spinal cord between E9.5-E11.5; bars indicate mean and SD of individual slices from three independent experiments; Kruskal–Wallis test not significant, $P = 0.3$.

I    Schematic of multiple non-overlapping regions of interest identified as Apical, Intermediate and Basal in the spinal cord tissue; width of regions in the apical-to-basal direction was 15 μm.

J    Peak to peak distance in auto-correlation of detrended Venus::HES5 spatial profile corresponding to apical, intermediate and basal regions of spinal cord at E10.5; dataset is different from (H); markers indicate average distance per experiment with a minimum of three z-stacks per experiment and two repeats (left and right of ventricle) analysed per z-stack; bars indicate mean and SD of six independent experiments; Kruskal–Wallis test not significant, $P = 0.115$; distances correspond to 4–5 cells considering the inter-nuclear distance in DV quantified in (H).

Source data are available online for this figure.

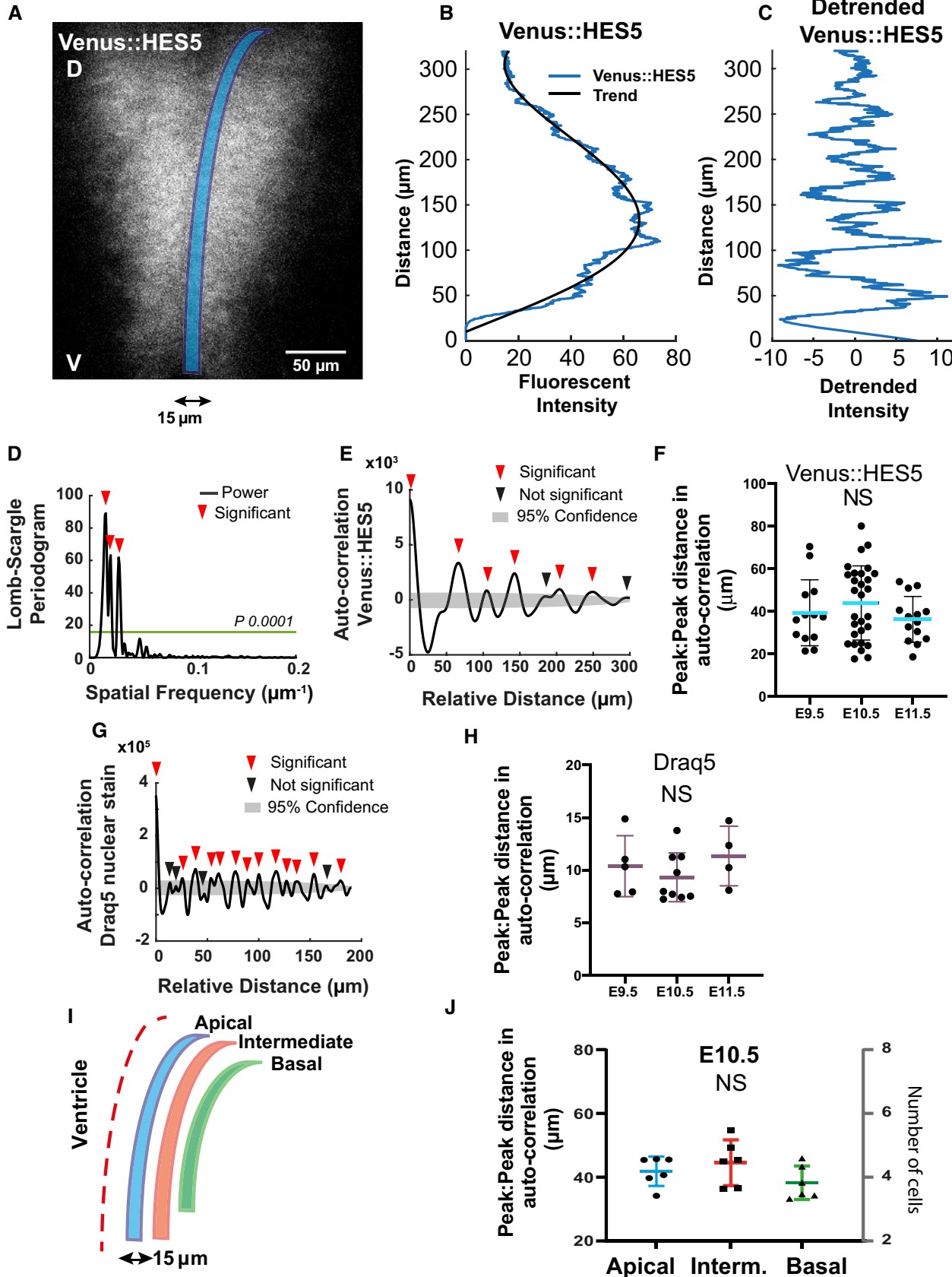

Figure 2.

of expression that is a result of different HES5 levels along the D-V axis and is not further investigated here (Figs 2B and EV2C). The microclusters could be detected as intensity peaks in the detrended Venus::HES5 intensity profile (direct comparison in Fig EV2A–C) and we observed multiple intensity peaks across the D-V axis (Fig 2 C). Periodicity analysis of the detrended spatial Venus::HES5 expression profile revealed the presence of dominant frequency peaks in the power spectrum (Fig 2D) and multiple significant peaks in the auto-correlation function in all tissues analysed (Fig 2E, Appendix Fig S1B and Appendix Table S1). A significant peak in the auto-correlation function shows that the signal has similarity to itself at the relative distance (or lag) indicated in the x-axis. Multiple significant peaks in an auto-correlation function are indicative of a periodic signal with the peak to peak distance in the auto-correlation corresponding to the period of the signal. The spatial period in Venus::HES5 expression was 30–60 μm with a median period of 40 μm and no significant difference between E9.5 to E11.5. Periodicity measurements from auto-correlation functions and the power spectrum corresponded well (Fig 2F and Appendix Fig S1C).

To understand how our observed spatial periodicity relates to nuclear count in the tissue, we analysed the spatial profile for the Draq5 nuclear stain from snapshot images of Venus::HES5 *ex vivo* slices. We observed peaks in Draq5 in regions of low Venus::HES5 indicating that the lower Venus::HES5 regions did not correspond to a lack of nuclei at this position of the tissue (Fig EV2D). As expected, the Draq5 signal observed along the D-V axis also showed multiple significant peaks in the auto-correlation that corresponded to a spatial period of 10 μm, a single-cell width and was consistent over developmental time (Fig 2G and H, and Fig EV2E and Appendix Fig S1D). Using this value, we estimate that the periodic occurrence of microclusters of cells with correlated levels of Venus:: HES5 has a median period of four cells. This corresponded well to the distance between microclusters in the high-resolution analysis of single-cell snapshots (Fig EV2F).

Since the apical region of spinal cord contains proliferative neural progenitors with high levels of HES5 that become downregulated when progenitors begin to migrate towards basal regions, we interrogated if the spatial periodicity persisted in the A-B axis. Venus::HES5 expression profiles collected from apical, intermediate and basal regions (Fig 2I) within the HES5 expression domain at E10.5 all showed spatial periodicity (both power spectrum, Fig EV2G and auto-correlation, Fig 2J) with the period varying from approximately four cells in the apical side to three cells in the basal region (Fig 2J, Appendix Fig S1E). These results suggest that proliferative progenitors (localised apically) as well as differentiating progenitors (localised more basally) show local spatial correlation in Venus::HES5 levels between neighbouring cells where 3–7 neighbouring cells can be in a high or low state in synchrony with each other and that these clusters are repeated periodically in the D-V axis.

To test whether clusters extended in the anterior–posterior (A-P) axis, we took longitudinal cryosections of the spinal cord and performed auto-correlations of the Venus::HES5 spatial profile along the A-P axis (Fig EV2H–J). Peaks in the auto-correlation show spatial periodicity in A-P axis of around 30 μm (Fig EV2J). Thus, the scale of cluster size in A-P is comparable to that observed in D-V. We confirmed this in our existing kymograph data by correlating the expression of HES5 at subsequent z-positions extending in the A-P axis in the same slice (Materials and Methods). Indeed,

correlations in A-P persisted at less than 30 μm but were lost further away (Fig EV2K).

The microclusters could be set up earlier on in development with fewer or single cells and then clonally expand through cell division. However, the similar microcluster size and Venus::HES5 spatial periodicity between E9.5, E10.5, and E11.5 argues against a clonal expansion mechanism. Coordinated cell behaviours such as nuclear motility may also contribute. We found weak positive correlation in the movement of nuclei in apico-basal axis between cell pairs less than 30 μm apart, but there was a large variation in correlations, and the correlation dropped between cells further apart (Appendix Fig S1F). This weak correlation in apical–basal nuclear movement may contribute weakly to maintaining the microcluster pattern.

### The HES5 spatial pattern is dynamic over time

Given that single-cell Venus::HES5 expression dynamically fluctuates (Manning *et al*, 2019), we next investigated whether the spatially periodic pattern in Venus::HES5 is dynamic over a time scale of hours. To do this, we generated kymographs, single images that represent spatial intensity profiles in the same region of tissue over time, from 15 μm wide ventral–dorsal lines in movies of E10.5 Venus::HES5 spinal cord *ex vivo* slices (Fig 3A and B, Appendix Fig S2A and B, and Movie EV1). We noticed stripes in the kymograph, corresponding to the spatially periodic Venus::HES5 pattern (Fig 3 B). To investigate how long high HES5 and low HES5 microclusters persist over time we split the kymograph into adjacent 20 μm regions (half of the 40 μm spatial periodicity, chosen to capture the size of a microcluster) along the D-V axis and followed their levels over time (Materials and Methods). Hierarchical clustering of the dynamic behaviour of the kymograph regions revealed changes from low to high Venus::HES5, high to low, or re-occurring high–low–high, showing that clusters of cells can interconvert between low and high HES5 states (Fig 3C and additional examples Appendix Fig S2C). To exclude the possibility of sample drift in the DV axis being responsible for these dynamics, we used single-cell tracking from the same videos as the kymographs to determine that global DV drift is minimal (<20 μm per 12 h, Appendix Table S2) and only one in 10 tissues was excluded from temporal analysis. Thus, we could proceed to analyse the persistence of a microcluster in the high or low state and we found that it was on average 6–8 h with no difference between persistence of high or low states in the same region (Fig 3D and Appendix Fig S2D). We confirmed these results using a second method that detected high/low regions in the first 2 h of kymograph and fixing ROIs around these regions whereby we continued to observe changes in intensity over time with similar persistence (Appendix Fig S2E). This shows that the microstripes of HES5 expression are not stable but are dynamic over time.

Since the HES5 expression is periodic along the D-V axis, it can be represented as a spatial oscillator. Therefore, we used its phase characteristics denoting the position in the spatial cycle, to analyse how the HES5 signal changes from high to low in the same region over time. We transformed the detrended spatial Venus::HES5 intensity (Fig EV2L, Appendix Fig S3A) along the D-V axis to phase of the spatial oscillator of Venus::HES5 using the Hilbert transform (Materials and Methods). All experiments showed a dynamic pattern with changes in phase in any area of the tissue over the 12–24 h movies

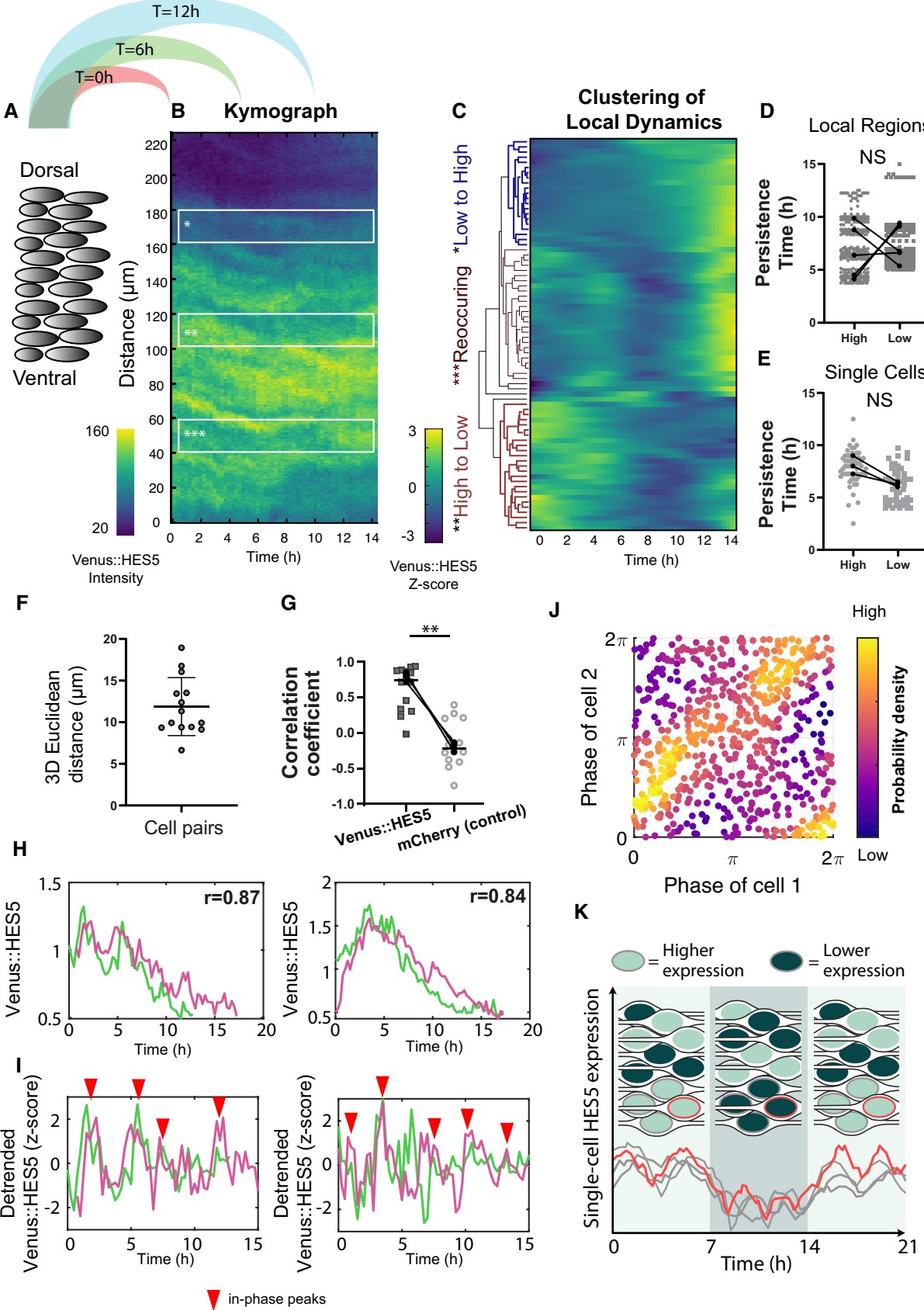

Figure 3.

**Figure 3. HES5 protein is expressed in a dynamical spatial periodic pattern modulated by Notch.**

A   Schematic of extracting kymograph information from tissue data by averaging Venus::HES5 intensities observed in E10.5 heterozygous spinal cord slices to generate one intensity profile in the dorsal–ventral axis per timepoint (see Materials and Methods).

B   Representative kymograph data showing spatiotemporal Venus::HES5 expression profile along ventral–dorsal direction in a 15 μm wide apical region and observed over 14 h; local bands of 20 μm width in D-V; region of interest markers indicate: *low to high, **high to low and ***re-occurring high/low activity in the same area.

C   Hierarchical clustering of apical Venus::HE5 expression from one representative experiment showing behaviour in the same area over time; columns represent fluctuations in Venus::HES5 intensity in small local areas (bands) obtained by dividing the spatial signal into non-overlapping 20 μm regions and normalising to the mean and standard deviation of each region over time (z-scoring); data have been subject to a Gaussian blur pre-processing step (see Appendix Fig S2B and Materials and Methods).

D   Persistence of Venus::HES5 in 20 μm regions expressed as continuous time intervals when signal in the band is high or low compared with its mean (see Materials and Methods); individual datapoints (grey) indicate quantification of high and low persistence time obtained from over 300 thin bands collected from multiple tissues with 2 z-stacks per tissue and two repeats (left and right of ventricle) per z-stack; dots indicate paired medians of five independent experiments; statistical test is paired *t*-test of median per experiment with two-tail significance and $P = 0.7171$.

E   Persistence of Venus::HES5 levels in high and low states taken from 60 tracked single cells collected from three independent experiments; paired *t*-test not significant $P = 0.0533$.

F   Relative distance between cell pairs computed from relative 3D Euclidean distance between nuclei over 12–15 h; dots indicate median distance over tracking period; horizontal lines show mean and SD of 14 cell pairs from three experiments.

G   Spearman correlation coefficients computed in the same cell pairs from Venus::HES5 and H2B::mCherry (control) nuclear intensity timeseries; markers in each condition indicate pairs; black dots indicate median correlation coefficients per experiment (four pairs, three pairs and seven pairs); lines show median of 14 pairs from three experiments; paired *t*-test with significance $P = 0.0058$.

H   Representative example timeseries of Venus::HES5 in cells pairs identified as remaining in close proximity; *r*-values indicate Spearman correlation coefficients between time traces over all co-existing timepoints.

I   Detrended Venus::HES5 fluorescent intensity timeseries (after z-scoring) corresponding to examples in (H); red arrows indicate in-phase peaks.

J   Density phase plots from instantaneous Hilbert phase reconstruction at multiple timepoints over a 12–14 h period; dots indicate the phase angle in Cell 1 and Cell 2 from 14 pairs collected from three experiments; colormap indicates probability density showing accumulation of phase values predominantly along the (0,0) and (2π, 2π) diagonal; light colours indicate most frequent.

K   Graphic representation of a neuroepithelial tissue with nuclei colour-coded to indicate clusters of high or low HES5 expression. The tissue is illustrated at three different time points to depict how clusters of cells can dynamically switch from high to low or low to high while the periodic spatial pattern is maintained. In the example time traces (corresponding to the three grey and one red highlighted nuclei), synchronised ultradian oscillations are shown as being overlayed on the slow-varying higher-amplitude switching dynamics.

Source data are available online for this figure.

(Appendix Fig S3B). Regions could be identified that maintained a similar phase over several hours followed by a change, indicating a switch in state of the Venus::HES5 pattern (Fig EV2L). Phase waves could be observed in some movies, indicated by the diagonal lines of similar colours in the spatial phase map (Appendix Fig S3B); however, these were variable across the data and did not have a consistent direction in the D-V axis between experiments. In summary, we find microclusters of cells with correlated Venus::HES5 levels that are a maximum of 2–3 cells wide in D-V and 3–4 in A-B axes and are arranged in a spatially periodic pattern. The pattern is also temporally dynamic with a persistence in a high or low level expression of 6–8 h but no consistent phase wave travel in D-V.

**Single cells in a microcluster coordinate HES5 expression at two different timescales**

We next addressed how the dynamic tissue pattern may be synthesised from single-cell Venus::HES5 expression. We have previously tracked Venus::HES5 in single nuclei of E10.5 spinal cord *ex vivo* slices and reported that about 40% of the progenitors show oscillations of 3–4 h periodicity (Manning *et al*, 2019). However, we also observed changes in the mean expression level of apical progenitors that varied at a time scale longer than 3–4 h (Manning *et al*, 2019). This slowly varying signal in progenitors was not investigated further at the time (Manning *et al*, 2019). Indeed, when we re-analysed single-cell Venus::HES5 expression data of apical progenitors we found that the slowly varying fluctuations have similar "persistence" time as the microclusters (Fig 3E vs D and Appendix Fig S2E). The distinction between the dynamics at shorter timescale (ultradian oscillations) and

the longer timescale fluctuations in mean HES5 levels is that slow-varying dynamics have larger amplitude compared with the ultradian (Appendix Fig S4A and examples in Manning *et al*, 2019; Appendix Fig S7). As such, both slow-varying and ultradian changes in HES5 could contribute to the formation and dynamic nature of microclusters with the slower varying fluctuations in HES5 mean levels specifically modulating the microcluster persistence time.

To investigate the single-cell expression inside a microcluster, we identified cell pairs that were in close proximity over 12 h (median Euclidean distance < 20 μm) (Fig 3F and Appendix Fig S4B). We found that 10/14 cell pairs showed a high positive correlation in their mean Venus::HES5 levels (Fig 3G: median 0.74 and examples Fig 3H) and this was reproducibly higher than the experimental control of nuclear H2B:mCherry in the same experiment (Fig 3G, median −0.2). Thus, single cells in a microcluster coordinate their HES5 levels over time. We then turned our attention to the ultradian HES5 activity in cell pairs by utilising detrending and subsequent phase reconstruction (Materials and Methods). The instantaneous phase of Venus::HES5 expression in cell pairs persistently showed in-phase peaks (examples Fig 3I red arrowhead). Phase–phase visualisation maps of all pairs at all recorded timepoints exhibited a large accumulation of Venus::HES5 instantaneous phases along the diagonal between (0,2π) and (2π,2π) indicating prevalence of in-phase behaviour at single-cell level in the same pair (Fig 3J). We also noted imperfections with some phase activity around (0,2π) and the presence of anti-phase peaks (Fig 3I and J); however, this was transient and not characteristic of any particular pair (Appendix Fig 4C–E and Materials and Methods). Moreover, we performed a cross-pairing control which showed that while in-phase

activity is reproducibly observed in neighbouring cell pairs, this effect is lost when pairing cells located further away in the same tissue (Appendix Fig S4F and G and Materials and Methods). These findings demonstrate that ultradian activity between neighbouring cells in a microcluster is predominantly in-phase; however, it does not translate to global synchrony across the tissue and we refer to this as "local in-phase".

To summarise, inside a HES5 expressing microcluster, cells predominantly show synchronised ultradian oscillations (or fluctuations) of 3–4 h; on top of that, each microcluster has a persistence time in a high or low state of about 6–8 h, a timescale coincident with the slower varying fluctuations observed in single-cell traces. Each HES5 expressing microcluster is a composite of these two dynamic activities observed at different timescales (diagram in Fig 3K).

## Notch inhibition extinguishes dynamic changes in Venus::HES5 microclusters between high and low states

We hypothesised that the periodic microclusters of HES5 are generated through Notch–Delta interactions that locally synchronise dynamic HES5 expression between neighbouring cells. To test this, we treated spinal cord slice cultures with the Notch inhibitor DBZ and performed kymograph analysis in the apical region of DMSO and DBZ treated slices. In Notch inhibitor conditions, the HES5 levels reduce continuously over time (Fig 4A) indicating that the DBZ is effective. The most noticeable difference in the spatiotemporal HES pattern was that the temporal transitions of microclusters from high to low Venus::HES5 were impaired by DBZ. We observed this at a temporal resolution at which single cells are unlikely to leave the region of interest (Appendix Table S3). We saw fewer changes in the phase of the spatial periodic Venus::HES5 pattern indicating the spatial pattern remained stable (Figs 4B–D and EV3A and B). This was quantified with a phase synchronisation index (see Materials and Methods), where low values indicate the presence of

phase changes at the same D-V locations. The phase synchronisation index was significantly higher in DBZ-treated tissue (Fig 4E) indicating that in the absence of Notch signalling, HES5 microclusters were more persistent in the same region and that the dynamic changes in Venus::HES5 microclusters between high and low levels are mediated by Notch signalling. The phase detection method (Hilbert transform) is not dependent on the level of expression and so the reduction in HES5 levels in DBZ does not affect the analysis of microcluster high-to-low and low-to-high phase switches. However, we did account for loss of periodicity in DBZ (discussed below) by comparing phase only over time intervals when spatial periodicity was still detected (see Materials and Methods).

We analysed the spatial periodicity of HES5 and found that the amplitude between high and low microclusters appears diminished compared with control DMSO treated conditions (Fig 4F). Spatial periodicity could be detected at the start of the movie, immediately after DBZ addition; however, the spatial periodicity was gradually extinguished through loss of Venus::HES5 levels and spatial amplitude death (Fig EV3C). Approximately 45% of the DBZ-treated slices did not show significant peaks in the auto-correlation of detrended spatial Venus::HES5 profile by 10–12 h of treatment (Fig 4G) whereas periodicity was maintained in all DMSO conditions. Spatial periodicity in detrended Venus::HES5 levels that could be detected in DBZ treatment at early time points frequently appeared higher in Notch inhibitor treated *ex vivo* slices than in DMSO control (Figs 4H and EV3D). Cell density also decreased in Notch inhibitor conditions suggesting this increase in spatial period was partially due to changes in the spatial arrangement of cells (Fig EV3E).

We also investigated how Notch inhibition may affect ultradian dynamics at single-cell level. We had previously reported that under DBZ conditions, single neural progenitors continue to show oscillations and fluctuations in HES5 before undergoing amplitude death (Manning *et al*, 2019). However, here we wanted to interrogate how DBZ affects the way cells coordinate their activity in the tissue. To do this, we used the Kuramoto Order Parameter (KOP, also known

---

**Figure 4. Notch inhibition increases HES5 pattern persistence.**

A   Start:Finish Venus::HES5 intensity ratio in E10.5 Venus::HES5 heterozygous spinal cord slices treated with control (DMSO) and Notch inhibitor DBZ (2 μM) observed over 16 h; bars indicate mean and standard deviation of DMSO ($n = 3$ experiments) and DBZ ($n = 4$ experiments); 2-tailed $t$-test ****$P = 0.0001$.

B   Representative spatiotemporal plots of the detrended Venus::HES5 pattern along ventral–dorsal direction in DMSO control (left panel) and DBZ conditions (right panel) obtained by averaging kymographs data in the same region over 2-h time intervals.

C   Schematic indicating the correspondence between Venus::HES5 spatial oscillator represented as detrended level and phase angle characteristics; the spatial oscillator traverses repeated cycles including start (HES5 low-orange arrowhead), middle (HES5-teal arrowhead) and end (HES5 low-red arrowhead) which in phase space corresponds to phase angles 0, π and 2π, respectively.

D   Phase maps corresponding to DMSO (left panel) and DBZ (right panel) detrended Venus::HES5 data shown in (B).

E   Phase synchronisation measure (see Materials and Methods) of the detrended Venus::HES5 spatial oscillator measured over time in E10.5 Venus::HES5 spinal cord periodic slices treated in DMSO vs DBZ conditions up to 10 h; dots indicate DMSO (21 kymographs, $n = 3$ experiments) and DBZ (19 kymographs, $n = 4$ experiments); bars indicate mean and SD; 2-tailed Mann–Whitney test with significance ****$P < 0.0001$.

F   Spatial peak: trough fold change in Venus::HES5 intensity profile in the D-V axis measured at 2 h and 10 h in DMSO and DBZ-treated E10.5 Venus::HES5 spinal cord slices; dots indicate average over three z-slices from DMSO ($n = 3$) and DBZ ($n = 4$) experiments; lines indicate median per condition; 1 tailed unpaired $t$-test with significance *$P < 0.05$.

G   Percentage of *ex vivo* slices with significant spatial period detected after 10–12 h of DMSO and DBZ conditions; significant spatial period defined as multiple significant peaks in auto-correlation detected above the 95% confidence bounds; dots indicate % per experiment; bars denote median and inter-quartile range of DMSO ($n = 3$) and DBZ ($n = 4$) experiments; 1-tailed $t$-test with significance **$P = 0.0062$.

H   Peak to peak distance in auto-correlation plots of detrended Venus::HES5 spatial profile in DMSO and DBZ-treated E10.5 Venus::HES5 spinal cord slices; grey dots represent significant mean peak to peak distance of DMSO (100) and DBZ (105) auto-correlation functions collected from three z-stacks per slice and two repeats (left and right of ventricle) with multiple timepoints; bars indicate median per experiments from DMSO ($n = 3$) and DBZ ($n = 4$) experiments; error bars indicate SD; 2-tailed Mann–Whitney test ****$P < 0.0001$.

Source data are available online for this figure.

---

                          

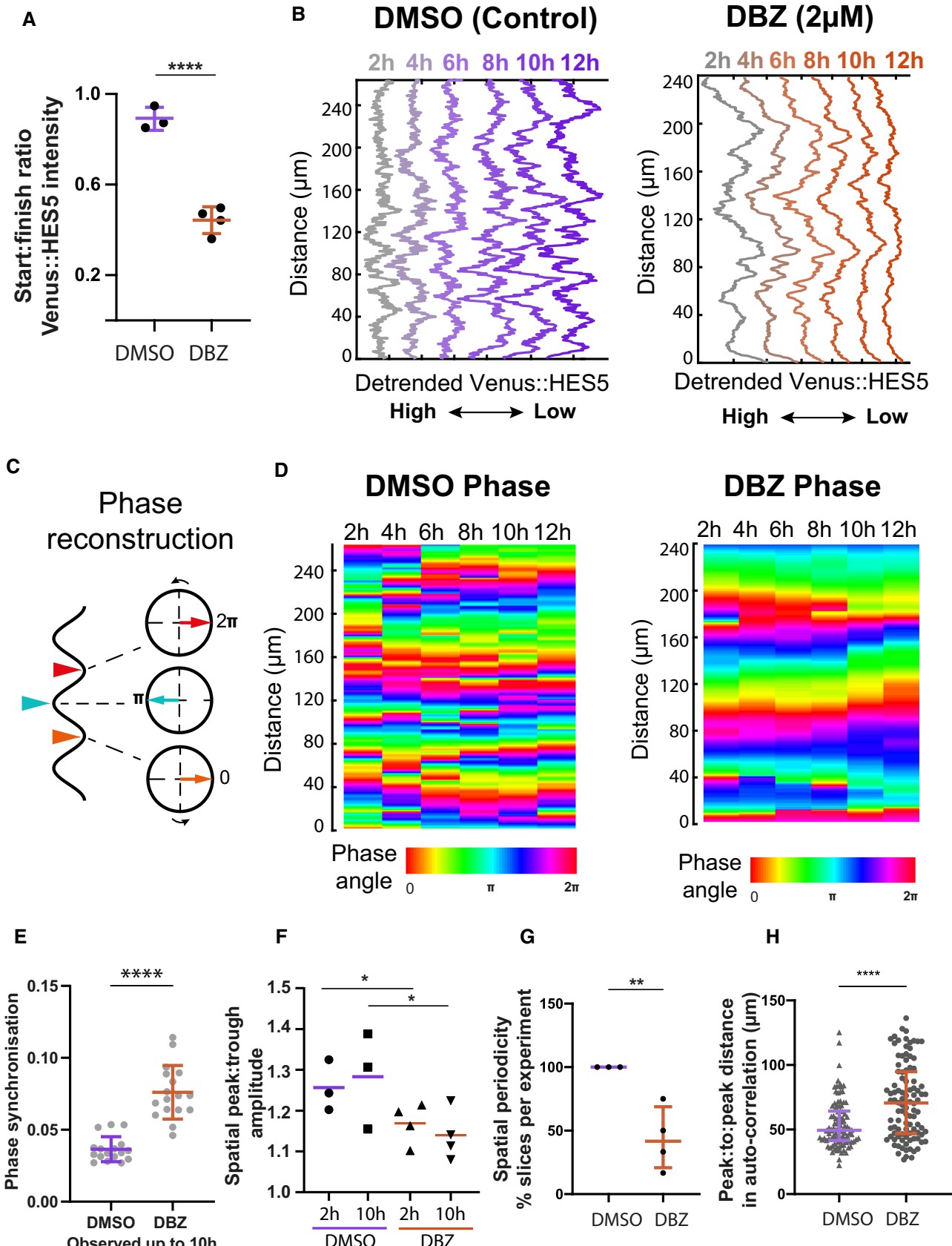

**Figure 4.**

as mean-field value) a population measure of synchrony (Choi *et al*, 2000). High KOP levels close to 1 are indicative of global in-phase activity whereas low KOP values close to 0 are indicative of no in-phase synchrony (see Materials and Methods). We found that KOP of single progenitors showed weak levels of synchrony under DMSO conditions (Appendix Fig S4H, mean 0.36) consistent with our findings of local in-phase activity but not indicative of global synchrony. Furthermore, we observed a significant reduction in KOP values in DBZ conditions (Appendix Fig S4H, mean 0.15).

Taken together, these findings show that Notch signalling is responsible for certain aspects of the pattern, such as the dynamic switching between high/low HES5 microcluster states over time. However, inhibition of Notch does not seem to abolish the existence of microclusters or their spatial periodicity, as they can still be detected until amplitude death occurs and the HES5 levels are depleted. At single-cell level, we observe that Notch signalling is likely to promote local in-phase ultradian coordination between cells within a microcluster.

### A model of Notch–Delta with HES5 auto-repression containing stochasticity and delays recapitulates the existence of local in-phase HES5 dynamics

We used computational modelling to help us understand how positively correlated, spatially periodic, and dynamic microclusters of cells may emerge in the spinal cord. At single-cell level, HES5 protein expression oscillations are due to HES5 self-repression, an intra-cellular transcriptional time delay ($\tau_H$) and short protein and mRNA half-lives (Jensen *et al*, 2003; Monk, 2003; Momiji & Monk, 2008). We represented the auto-repressive interactions between HES5 mRNA and protein using stochastic differential equations with time delay, as previously described in (Galla, 2009; Phillips *et al*, 2016; Manning *et al*, 2019). This single-cell model has been shown to faithfully recapitulate statistics of single-cell HES5 expression dynamics collected from spinal cord tissue (Manning *et al*, 2019). We extended the single-cell mathematical description of HES5 to a coupled dynamical model by incorporating a repressive interaction

in the form of a Hill function, that describes how HES5 protein in one cell represses Hes5 transcription in a neighbouring cell via Delta-Notch signalling (Figs 5A and EV4A). We introduce the following set of inter-cellular parameters (Fig 5B and Materials and Methods): (i) *inter-cellular time delay*, representing the time required to transfer the signal from one cell to another, that is, the time required for a change in HES5 protein in one cell to affect Hes5 transcription in a neighbouring cell through Notch–Delta; (ii) *the inter-cellular repression threshold,* representing the amount of HES5 protein required to reduce Hes5 transcription in a neighbouring cell by half; the inter-cellular repression threshold is inversely proportional to coupling strength where higher coupling strength (or low inter-cellular repression threshold) indicates that less protein is needed to repress the neighbour's Hes5 transcription by 50%; and (iii) *inter-cellular Hill coefficient* indicating how steep the response curve of Hes5 transcription is in response to a change in HES5 protein in the neighbouring cell, with higher values corresponding to increased nonlinearity. Interactions between cells are considered in a hexagonal grid whereby each cell can interact with its immediate six neighbours and repression between cells is calculated through the inter-cellular Hill function by averaging HES5 protein abundance over six neighbours (Fig 5B and C and Materials and Methods). Thus, we generated a comprehensive, multiscale and stochastic model with time delays, representative of the Delta–Notch–Hes interactions in the multicellular tissue environment.

We parameterised this multiscale HES5 model with previously determined experimental measures of HES5 protein and mRNA stability and with parameter values of the single-cell HES5 self-repression loop that can reproduce single neural progenitor HES5 dynamics (see Materials and Methods and Appendix Table S4 Main), as identified through Bayesian inference in our previous work (Manning *et al*, 2019). We then investigated the parameter space of unknown model parameters that are characteristic of cell-to-cell interactions, namely the repression threshold (inverse of coupling strength) and time delay, to identify values that are compatible with the temporal period and phase synchronisation level of single-cell Venus::HES5 expression dynamics (Fig 5D and E). The mean temporal period of Venus::

---

**Figure 5. Multicellular cell–cell coupling model explains the emergence of microclusters.**

A  Schematic of repressive interactions via Notch–Delta between neighbouring cells whereby the effects of HES5 protein in Cell 1 (marked as $P_1$) on transcription in Cell 2 and vice versa are represented using an inter-cellular Hill function $J(P_{1,2}(t - \tau_{ND}))$ where $t$ denotes time and $\tau_{ND}$ represents the inter-cellular time delay, the time interval required to synthesise the intermediate molecular species (detailed in Fig EV4A); HES5 auto-repression is represented using an intra-cellular Hill function $G(P_{1,2}(t - \tau_H))$ where $\tau_H$ represents the inter-cellular time delay, the time interval required for protein to be produced and repress its own transcription.

B  Mathematical description of the inter-cellular Hill function and its parameters: time delay ($\tau_{ND}$), repression threshold ($P_0$) and Hill coefficient ($n$); (bottom left panel) higher $P_0$ corresponds to reduced inter-cellular repression (i.e. decreased coupling strength) and conversely lower $P_0$ corresponds to higher coupling strength; (bottom right panel) increasing values of $n$ correspond to increased steepness of the inter-cellular response.

C  Multiscale coupled mathematical model of the tissue environment consisting of a 2D hexagonal grid of cells expressing HES5 protein with corresponding auto-repression (described in (A)) coupled together by repressive interactions between its six immediate neighbours (see Materials and Methods); single-cell inter-cellular repression is a Hill function (with parameters described in (B)) dependent on mean protein abundance in the neighbouring cells.

D  Parameter exploration of single-cell temporal period emerging from the model at different repression threshold and time delay values.

E  Parameter exploration of phase synchronisation quantified using the Kuramoto Order Parameter (see Materials and Methods) where 1 indicates global in-phase activity and 0 indicates no coordination of phase between cells.

F  Parameter selection strategy combining experimentally determined temporal phase (insert left panel) and KOP (insert right panel) values in spinal cord tissue (see (Manning *et al*, 2019) and Materials and Methods) to indicate areas where model statistics (i.e. mean temporal period and KOP of synthetic data) resemble real tissue; values within ±1 SD and 2±SD from the mean of the tissue are identified and values found outside of ±2.4 SD from the mean of tissue are excluded.

G  Representative examples of synthetic kymograph data obtained at specific levels of repression threshold: *Alternating high–low* ($P_0 = 400$), *Global in phase* ($P_0 = 15,000$) and *Local in phase* ($P_0 = 21,000$) and corresponding KOP values; the presence of microclusters at weak coupling is indicated with red arrowheads; time delay 150 min, $n = 4$.

H  Kymograph data obtained in the absence of coupling between cells; phase relationships are un-coordinated resulting in a KOP≈0.

I, J  Synthetic data timeseries corresponding to simulations in (G).

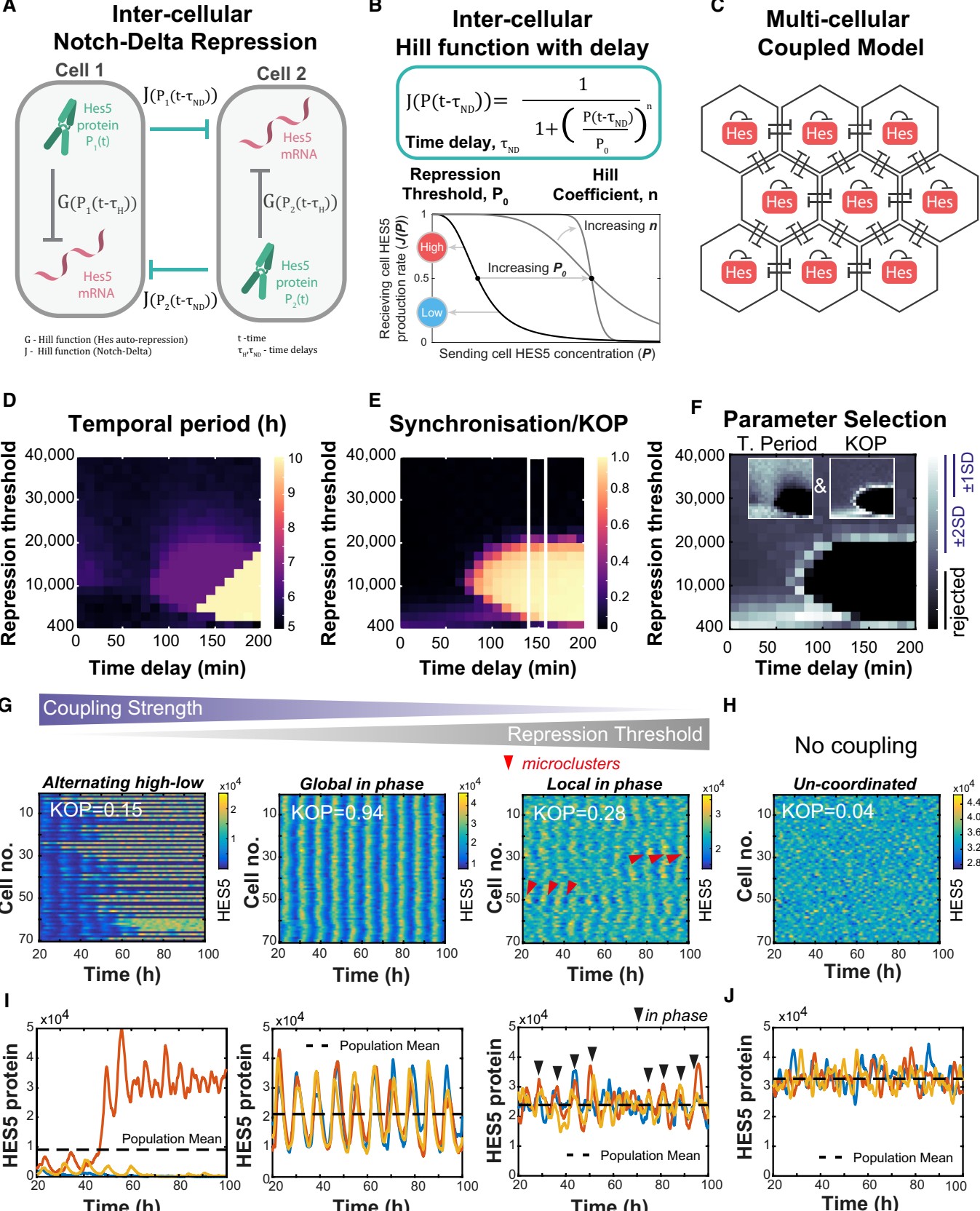

**Figure 5.**

HES5 in *ex vivo* spinal cord tissue is approx. 2–6 h (mean 3.3 h) (Manning *et al,* 2019), which could be reproduced by the model in a wide range of coupling strength and inter-cellular time delay values (Fig 5D and F). We measured the temporal phase synchronisation (KOP) between single Venus::HES5 expressing cells in the apical region and we found that the KOP was between 0.15 and 0.4 (Appendix Fig S4I, mean 0.3) consistent with KOP in the DMSO data (Appendix Fig S4H). This measure aided us in further reducing the parameter space of repression threshold and inter-cellular time delay that could fit the observed data (Fig 5E and F). The accepted parameter values for inter-cellular time delay were consistent with a Delta to Hes transmission time of 128 min measured experimentally (Isomura *et al,* 2017). A Hill coefficient value larger than 2 was required for notable synchrony to emerge (KOP>0), and only minor differences in terms of parameter selection were observed for values between 3 and 6 (Appendix Fig S5A).

This parameter exploration allowed us to optimise the search for spatial patterns that emerge at different coupling strengths using kymograph analysis (Fig 5G and H). We set the inter-cellular time delay to 150 min and Hill Coefficient to 4 (Materials and Methods) and then compared the synthetic HES5 spatiotemporal characteristics at specific coupling strength levels (parameter space indicated by the white box in Fig 5E). Our comparison showed that strong coupling (i.e. high coupling strength or low inter-cellular repression threshold) induces *Alternating high–low* dynamics whereby single neighbouring cells adopt either high oscillatory HES5 or stable low HES5 in an alternating spatial pattern that does not evolve over time (Fig 5G, *Alternating high–low*, Movie EV2 first panel). Meanwhile at mid-level coupling, the multiscale model induces globally synchronised oscillations in all cells (Fig 5G, *Global in phase* and Movie EV2 second panel). At weak coupling strength, the spatial patterns show areas of local synchronisation emerging between neighbouring cells (Fig 5G, *Local in phase* and Movie EV2 third panel) resembling activity observed in tracked single-cell pairs in experimental data (Fig 3 K). Under no coupling conditions, we observed autonomous non-synchronised stochastic oscillations and fluctuations across the tissue (Fig 5H and Movie EV2 fourth panel). These observed changes in synchronisation are indicated by population KOP values (Fig 5G), and we further confirmed that the KOPs correspond to changes in synchrony in terms of single-cell expression dynamics between neighbouring cells (Fig 5I). As expected, in the uncoupled cells we observed no synchrony (KOP≈0) and activity in neighbouring cells was un-coordinated over time (Fig 5H and J). Therefore, the model can recapitulate the local in-phase behaviour in Venus::HES5 observed between single-cell pairs in a microcluster.

Our explorations of synthetic data show that at weak coupling strength microclusters consisting of in-phase cells can be generated in the model with a diameter of 2–6 cells (Appendix Fig S5B and Materials and Methods), consistent with cluster size in spinal cord tissue. However, the occurrence rate of microclusters was low, as these were observed around 20–30% of the time, although still higher than in the uncoupled situation (Appendix Figs S5C and S6A). Thus, weak coupling conditions generate microclusters by promoting in-phase activity between neighbouring cells; however, these appear transiently and with low probability. In addition, the microclusters of locally synchronised cells were not spatially periodic (Appendix Figs S5D and S6B). As expected, at high coupling (low repression threshold) we detected an alternating pattern of

HES5 with a spatial periodicity of two cells, which is a characteristic of the classic lateral inhibition alternating high–low pattern (Appendix Figs S5D and S6B).

In conclusion, our multicellular coupled model shows that spinal cord progenitors can locally synchronise at weak coupling strength to generate microclusters of 2–6 cells in diameter, a similar size to those seen in tissue, (Figs 1D and EV1D) with single-cell Venus:: HES5 expression dynamics consistent with previous reports (Manning *et al*, 2019). However, the model cannot recapitulate the repeated spatial coordination and continuous presence of dynamic microclusters, suggesting that additional mechanisms may act in the tissue environment to stabilise their presence and promote spatially periodic emergence.

### The model predicts that probability of differentiation is regulated by the coupling strength between cells

To understand how the spatial pattern of HES5 and dynamic micro-patterns in particular may affect properties of neurogenesis, we made the assumption that when HES5 is low, there is increased probability that the cell would differentiate consistent with findings that differentiation is accompanied by switching off of HES5, a repressor of neurogenesis (Bansod *et al,* 2017; Sagner *et al,* 2018; Manning *et al,* 2019). We introduced a "differentiation threshold", which was set at the level of the HES5 population mean for each simulation (Fig 5I, *Population Mean*) and we reasoned that if expression level in a cell dropped below this threshold there was an increasing probability to switch off HES5 and differentiate (Fig 6A). We found that at high coupling strength (*Alternating high–low* conditions) the probability to differentiate is the highest, whereas medium and weak coupling strength (corresponding to *Global* and *Local in phase* synchronisation, respectively) had progressively lower probability of differentiation (Fig 6B).

To understand why this is happening, we looked at the Coefficient of Variation (CoV, Fig 6C), a measure of variability denoting standard deviation over the mean. We investigated both the temporal (Temporal CoV) and spatial variation (Spatial CoV) in simulated HES5 expression. Indeed, both temporal (indicative of single-cell amplitude) and spatial CoV (indicative of variation between HES5 high and low regions in space) appear highest in *Alternating high–low* conditions and lowest for *Local in phase* micro-patterns (Fig 6 C). However, we found that changes in spatial CoV correlated better with changes in rate of differentiation, especially at low repression threshold/high coupling strength (Fig 6C vs B *Alternating high–low*). Thus, our model predicts that the strength of cell:cell coupling may increase the probability of differentiation through amplifying cell:cell differences in abundance which in turn affects how far the cells dip below the threshold of differentiation.

### In tissue, HES5 spatial pattern varies predictably with the rate of differentiation

To test the computational prediction that the spatial pattern of HES5 (determined by the coupling strength) regulates the probability of differentiation, we compared the pattern in motorneuron and interneuron progenitor domains. We chose this comparison because at E10.5 the motorneuron domain is known to have a higher differentiation rate than the interneuron domain (Kicheva *et al,* 2014);

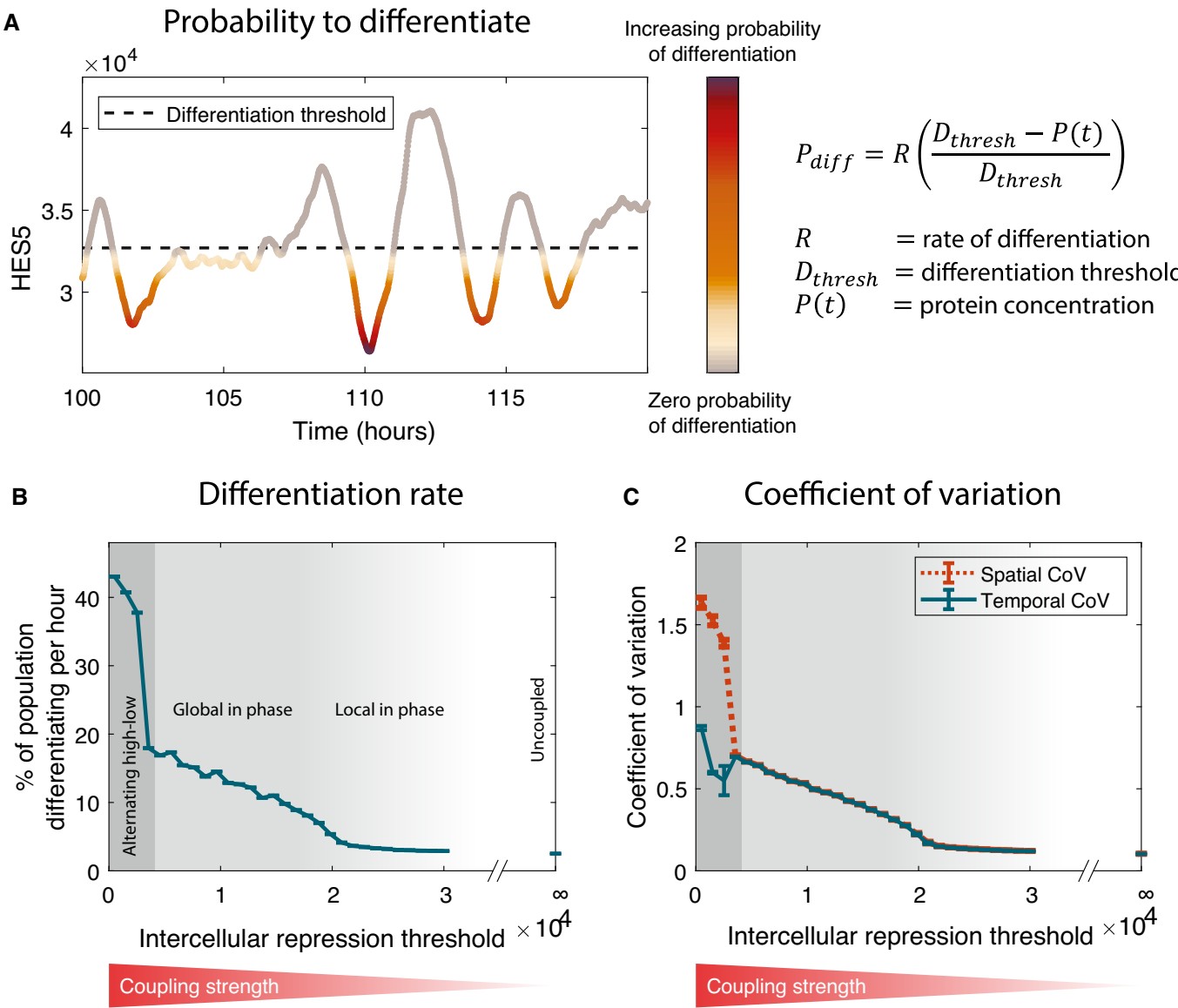

**Figure 6. Cell–cell coupling strength can regulate probability of differentiation in a multicellular environment.**

A Representative synthetic timeseries example and mathematical description of probability of differentiation ($P_{diff}$) in relation to population mean HES5 protein levels (referred to as "differentiation threshold", $D_{thresh}$) whereby HES5 protein abundance ($P(t)$) dropping below the threshold increases the rate at which cells differentiate.
B Differentiation rates estimated from the multicellular coupled model (detailed in Fig 5) over a wide range of repression threshold values corresponding to decreasing coupling strength; three dynamic regimes are labelled as *Alternating high–low*, *Global in phase* and *Local in phase* mirroring examples shown in Fig 5G and I.
C Analysis of temporal CoV and spatial CoV from synthetic data corresponding to differentiation rates shown in (B); these statistics indicate that spatial variability correlates better with differentiation rates meanwhile temporal variability shows only a moderate quasi-linear increase in *Alternating high–low* conditions compared with the rest.

Data information: Single-cell parameters used to generate (B) and (C) are shown in Appendix Table S4 Main, and the multicellular parameters used in (B) and (C) were $n_{ND} = 4$, $\tau_{ND} = 150$ min. Each value plotted in (B) and (C) shows the mean and SD from 10 simulations at each repression threshold value.
Source data are available online for this figure.

therefore, one would expect a different HES5 spatial pattern. We stained for the motorneuron progenitor marker OLIG2 (Figs 7A and EV4B and C) and analysed expression levels of Venus::HES5 and Neurogenin 2 (NGN2) in the two domains. The motorneuron domain had lower HES5 levels and higher NGN2 levels than the interneuron domain (Fig 7B) consistent with the opposing activity of these genes on cell differentiation (Imayoshi & Kageyama, 2014).

We then used nuclear segmentation and pseudo-color analysis of mean Venus::HES5 intensity per nucleus (Fig 7C) and found that the interneuron domain shows the presence of microclusters mainly consisting of 2–3 cells wide in the dorsal to ventral axis whereas in the motoneuron domain high Venus::HES5 cells were mainly found as single cells, alternating with cells expressing lower Venus::HES5 (Fig 7D). We validated this finding further by investigating spatial

periodicity by domain in live tissue slices. The domain border between motorneuron and interneuron progenitors was 35 μm ventral to the peak of HES5 expression (Fig EV4D) allowing us to correctly identify the two domains without the need for an OLIG2 reporter in the same tissue. We found that spatial periodicity was reduced in the motorneuron compared with the interneuron domain when analysed using both peak to peak distance in auto-correlation (Fig 7E, MN mean 31 μm vs IN mean 41 μm and Fig EV4E and F) and dominant spatial period by Lomb-Scargle periodogram (Fig EV4G, MN mean 25 μm vs IN mean 40 μm). Thus, both nuclear segmentation analysis and spatial periodicity indicated that, in the interneuron domain, microclusters of cells are found in a spatially periodic pattern repeated every four cells. Meanwhile, the motorneuron domain shows alternating high and low HES5 levels between neighbouring cells and a significant reduction in spatial periodicity, both of which are pointing to the motorneuron domain more closely resembling *Alternating high–low* conditions.

The model predicts that the coupling strength regulates the type of spatial micro-patterning hence, we hypothesised that the interneuron and motorneuron domains have different coupling strength. The model indicates that weak coupling, likely to be characteristic of the interneuron domain, would generate smaller cell–cell concentration differences compared with strong coupling (Appendix Fig S7A). This is because weakly coupled cells have less ability to repress the transcription of their neighbours and so are more similar in levels. This relationship should persist even after correcting for mean level in each condition. We have previously used fluorescence correlation spectroscopy (FCS) to generate a spatial map of nuclear Venus::HES5 concentration in the E10.5 spinal cord (Manning *et al*, 2019). Using this data, we calculated the difference in Venus::HES5 concentration between neighbouring cell pairs relative to the mean by domain and indeed found that it is lower in the interneuron domain compared with the motorneuron domain (Fig 7F). The correction by mean was important as variability in expression is expected to scale with the mean. This finding was confirmed by measuring the spatial amplitude of Venus::HES5, which was also higher in the motorneuron domain (Fig EV4H). These findings are consistent with the notion that the coupling strength in the IN domain is lower than in MN one. Taken together, these results show that interneuron progenitors are more likely to

be found in a locally synchronised state through weak coupling which correlates with a lower rate of differentiation. By comparison, progenitors in the motorneuron domain are mostly found in alternating high–low pattern and show a higher rate of differentiation, as predicted computationally by a higher coupling strength.

### NGN2 expression is spatially periodic and coordinates with the HES5 pattern

Given that the spatial pattern of HES5 is relevant to the rate of neurogenesis, we investigated the wider applicability of our findings by characterising the spatial patterns of other genes in the Notch–Delta gene network. Chromogenic in situ hybridisation of *Dll1* and *Jag1* mRNA shows that *Dll1* has a broad expression domain that covers the motor neuron domain and the ventral-most part of the interneuron domain (Fig EV5A) (Marklund *et al*, 2010). Alternate stripes of *Jag1* and *Dll1* are observed in the intermediate spinal cord, which covers the remaining part of the interneuron domain (Fig EV5A) (Marklund *et al*, 2010). We performed smiFISH for *Dll1* to get a high-resolution understanding of *Dll1* expression pattern in the interneuron domain where HES5 is expressed in microclusters. We found that *Dll1* expression is non-uniform and appeared in microstripes of a few cells (Fig EV5B and C, Materials and Methods, Appendix Table S5), suggesting that other genes show similarities in local spatial patterning.

We next analysed the spatial expression pattern of the proneural factor NGN2. Using both NGN2 antibody staining and a NGN2::mScarlet fusion reporter mouse (Appendix Fig S8A–C and Materials and Methods), we found that NGN2 also has a spatially periodic expression pattern, with around half the spatial period of Venus::HES5 (Fig 8A–C). The spatial period of NGN2 is smaller in the motorneuron domain with a mean period of 21 μm supporting the conclusion that NGN2 spatial expression patterns are different between motorneuron and interneuron domains (Fig 8D). To understand how the NGN2 and Venus::HES5 periodic patterns map on to each other, we used the cross-correlation function of the NGN2 and Venus::HES5 spatial profile from the same tissue (Fig 8E and F). The cross-correlation analysis showed the presence of multiple peaks indicating coordination between the two signals that was not reflected in the brightfield control (Fig 8F). As expected for signals

---

**Figure 7.   Type of HES5 spatial pattern and coupling strength correlates with rate of differentiation in motorneuron and ventral interneuron domains.** ▶

A   Transverse cryosection of E10.5 Venus::HES5 spinal cord. Venus::HES5 endogenous signal, OLIG2—motorneuron progenitor marker, NGN2—early marker of neuronal commitment; scale bar 20 μm.

B   Relative nuclear intensities of Venus::HES5 and NGN2 in motorneuron and interneuron progenitors; bars show mean and SD of at least 494 cells per domain from five slices in two experiments; Kruskal–Wallis with Dunn's multiple comparison test adjusted P-values **P = 0.0032, ***P < 0.001.

C   Pseudo-color look-up table applied to mean nuclear Venus::HES5 intensity within motorneuron (MN) and interneuron (IN) domains, corresponding to segmented image in (A).

D   Dimension of microclusters in DV axis for MN and IN domains; microclusters counted contained cells with high and similar levels of HES5 (Materials and Methods); bars show mean ±SD; data consists of 34 microclusters measured from five sections and three independent experiments; 2-tailed Mann–Whitney test. ****P < 0.0001.

E   Peak to peak distance in auto-correlation plots of detrended Venus::HES5 spatial profile in MN and IN domains; this is a measure of spatial period in Venus::HES5 profile along dorsal–ventral axis of spinal cord; grey data points represent mean peak to peak distance of at least three slices with left and right ventricle analysed separately in six experiments; black dots show median per experiment and line shows overall median; 2-tailed Mann–Whitney test P-values ****P < 0.00001.

F   Cell–cell concentration differences in HES5 between neighbours, normalised to mean concentration of HES5 in that domain; grey data points represent normalised concentration difference between a pair of neighbours, bars shows mean and SD; two independent experiments; 2-tailed Mann–Whitney test with P-values ***P = 0.003, ****P < 0.00001.

Source data are available online for this figure.

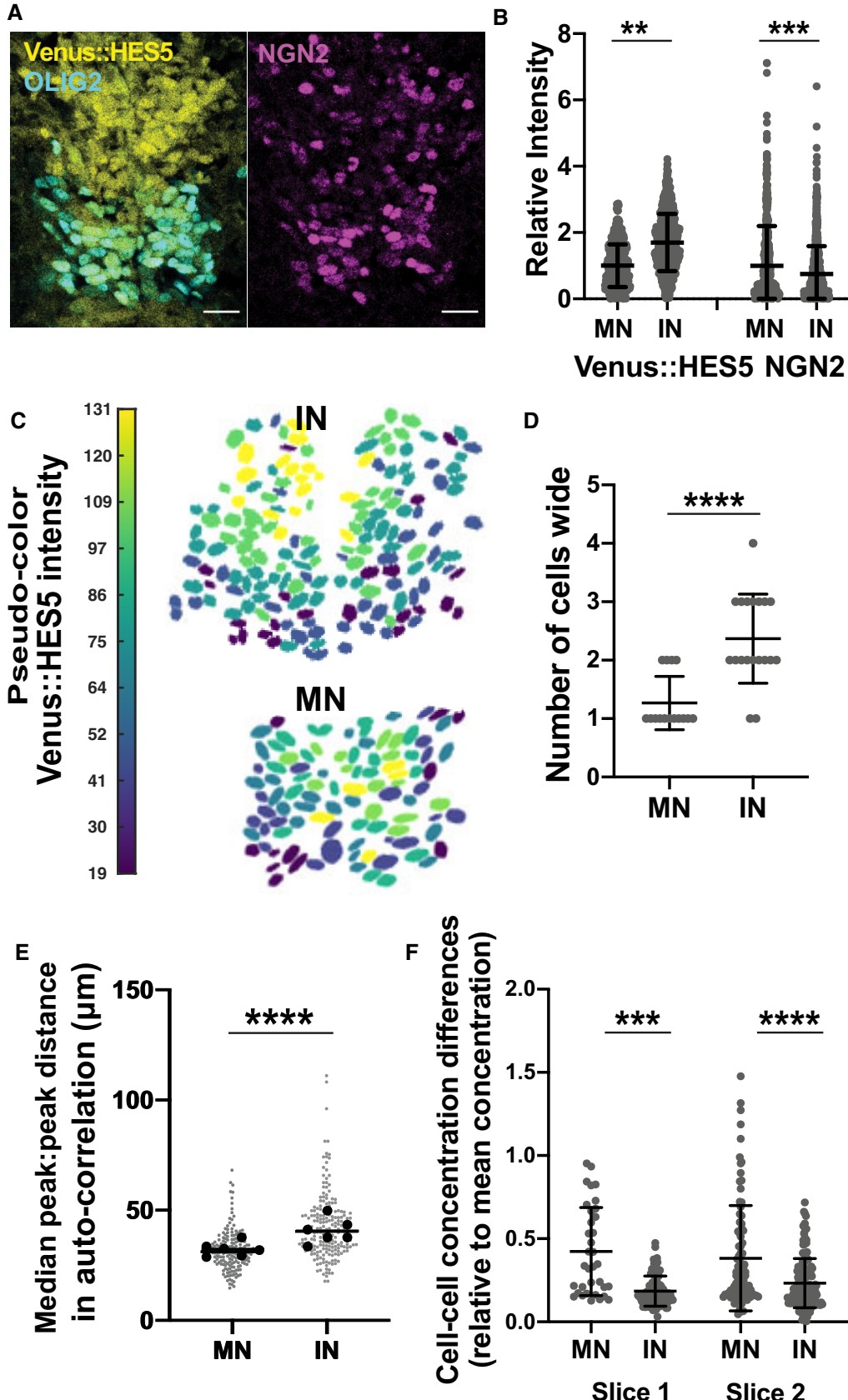

**Figure 7.**

of different periodicities, we observed primary peaks indicative of positively correlated activity (Fig 8F, red arrowheads) as well as secondary peaks indicative of negatively correlated activity (Fig 8F, black arrowheads). The cross-correlation is also indicative of whether peaks of activity are present in the same area. To ascertain this, we measured phase shift as the absolute lag corresponding to the primary cross-correlation peak closest to lag 0. In Fig 8F, such a peak falls close to lag 0 thus indicating that NGN2 and Venus::HES5 patterns coordinate in the same region. Thus, we concluded that NGN2 shows a spatial periodic pattern of half the period of Venus::HES5 resulting in half of the NGN2 high cells occurring in HES5 high microclusters and half in HES5 low.

Furthermore when we performed phase shift analysis in multiple cross-correlation examples (Materials and Methods), the shift was minimal and consistently less than a single-cell width (Fig 8G). This strongly pointed to coordination not only in the same region but also in the same cells. We subsequently investigated this by using single nuclear segmentation of high-resolution images to visualise the NGN2-HES5 spatial relationship. Indeed, we found that within a HES5 microcluster in the interneuron domain, only 1–2 cells (and in the MN domain only one cell per cluster) show high NGN2 expression levels (Fig 8H). As high NGN2 is an early marker of differentiation, this suggests that similar to the mathematical model (Appendix Fig S7B and C) cells in a cluster do not differentiate in unison; instead, microclusters may act to select a cell for differentiation, hence regulating spatial and temporal aspects of neurogenesis.

## Discussion

In this paper, we have addressed how cells coordinate their decisions with that of their neighbours so that neurogenesis takes place at a pace appropriate for the anatomical location. We have investigated the fine-grained pattern of neurogenesis in the spinal cord by monitoring the spatiotemporal patterning of key progenitor TF HES5 using live imaging analysis that is optimised towards revealing coordinated tissue-level behaviour that would not otherwise be evident. In combination with computational modelling it enabled a multi-scale synthesis of the data with predictive power. We have uncovered an unexpected 3-tiered spatial and temporal organisation, which we discuss below in an ascending order of complexity.

First, within the ventral HES5 expression domain, which encompasses distinct MN and IN domains, we have discovered clusters of cells with positively correlated HES5 expression levels. These clusters, described for the first time here, are 2–3 cells wide in D-V and 3–4 cells wide in A-B axes, hence termed microclusters. To detect microclusters, we removed longer-range spatial trends such as overall gradients of intensity in HES5 expression (which have not been dealt with further here) allowing us to concentrate on local correlations of expression. By following Venus::HES5 in pairs of single cells in proximity, we find that microclusters are a composite of positive correlations in slow-varying mean levels of Venus::HES5 and locally synchronised (in-phase) ultradian HES5 dynamics. This type of composite spectral activity or nested oscillations have been previously described in circadian rhythms containing an ultradian periodicity as well as neuronal firing patterns (Lopes-Dos-Santos *et al*, 2018; Wu *et al*, 2018). We propose that the local synchronisation in ultradian HES5 dynamics comes from coupling through Notch–Delta, although we cannot rule out the possibility that sister cells have synchronous HES5 expression after division. In the latter case, Notch–Delta coupling may act to re-inforce or help maintain local coordination over time. We also found that the microcluster organisation extends to DLL1 although we have not been able to study it with live imaging in this work. The clustering organisation was surprising because previous studies have suggested that in neurogenesis oscillators are in anti-phase in neighbouring cells (Kageyama *et al*, 2008; Shimojo & Kageyama, 2016; Shimojo *et al*, 2016). DLL1 oscillations were observed with live imaging in tissue but only a

---

**Figure 8. NGN2 expression is spatially periodic and positively correlates with the HES5 pattern.**

A  Detrended spatial profile of NGN2::mScarlet-I intensity from transverse slice of E10.5 spinal cord from heterozygous knock-in mouse in ventral–dorsal direction; red indicates motorneuron (MN) domain, blue interneuron domain (IN).

B  Auto-correlation analysis of detrended NGN2::mScarlet-I intensity spatial profiles from motorneuron and interneuron domains; multiple peaks indicating spatial periodicity; significant peaks (red triangle) lie outside black dotted lines indicating 95% significance based on bootstrap approach (see Materials and Methods) and non-significant peaks (black triangle).

C  Ratio of NGN2:HES5 spatial period in the same tissue; grey dots show ratio for single image from four experiments; line shows overall median and error bars 95% confidence limits.

D  Peak to peak distance in auto-correlation plots of detrended NGN2::mScarlet-I spatial profile in motorneuron (MN) and interneuron (IN) domains as a measure of spatial period in NGN2 expression along dorsal-ventral axis of spinal cord; Grey data points represent mean peak to peak distance in a single slice, $n = 33$, left and right ventricle analysed separately in four experiments; black line shows overall mean, error bars show SD; 2-tailed Mann–Whitney test with exact *P*-value *** $P = 0.0003$.

E  Detrended spatial profile of Venus::HES5 (black) and NGN2::mScarlet-I (red) intensity from the same transverse slice of E10.5 spinal cord in ventral–dorsal direction.

F  Example cross-correlation function of Venus::HES5 with NGN2::mScarlet-I (thick black), Venus::HES5 with brightfield signal (black), and NGN2::mScarlet-I with brightfield signal (red) from the same transverse slice of E10.5 spinal cord; markers indicate the presence of two types of coordination namely in-phase (red arrowhead) and out-of-phase (black arrowhead).

G  Phase shift showing absolute lag distance corresponding to in-phase peak in Venus::HES5 vs NGN2::mScarlet-I cross-correlation function of spatial intensity profiles from the same slice. 34 individual data points from six slices, two experiments; red line indicates average inter-nuclear distance in D-V; bars show mean±SD; 2-tailed Mann–Whitney test not significant, $P = 0.32$.

H  Pseudo-color look-up tables applied to mean nuclear Venus::HES5 and NGN2 staining intensity in motorneuron (MN) and interneuron (IN) domains. Venus::HES5 microcluster and single NGN2 high cell (red arrow) in IN domain; Alternating high–low expression of Venus::HES5 in MN, red arrows show high cells.

I  Graphical summary: Through a combination of experimental and computational work we characterised the HES5 dynamic expression in the mouse E10.5 ventral spinal cord. We found evidence that progenitors located in two domains (motorneuron, MN and interneuron, IN) give rise to distinct spatiotemporal characteristics that are indicative of differences in coupling strength and can explain increased differentiation rates observed in MN.

Source data are available online for this figure.

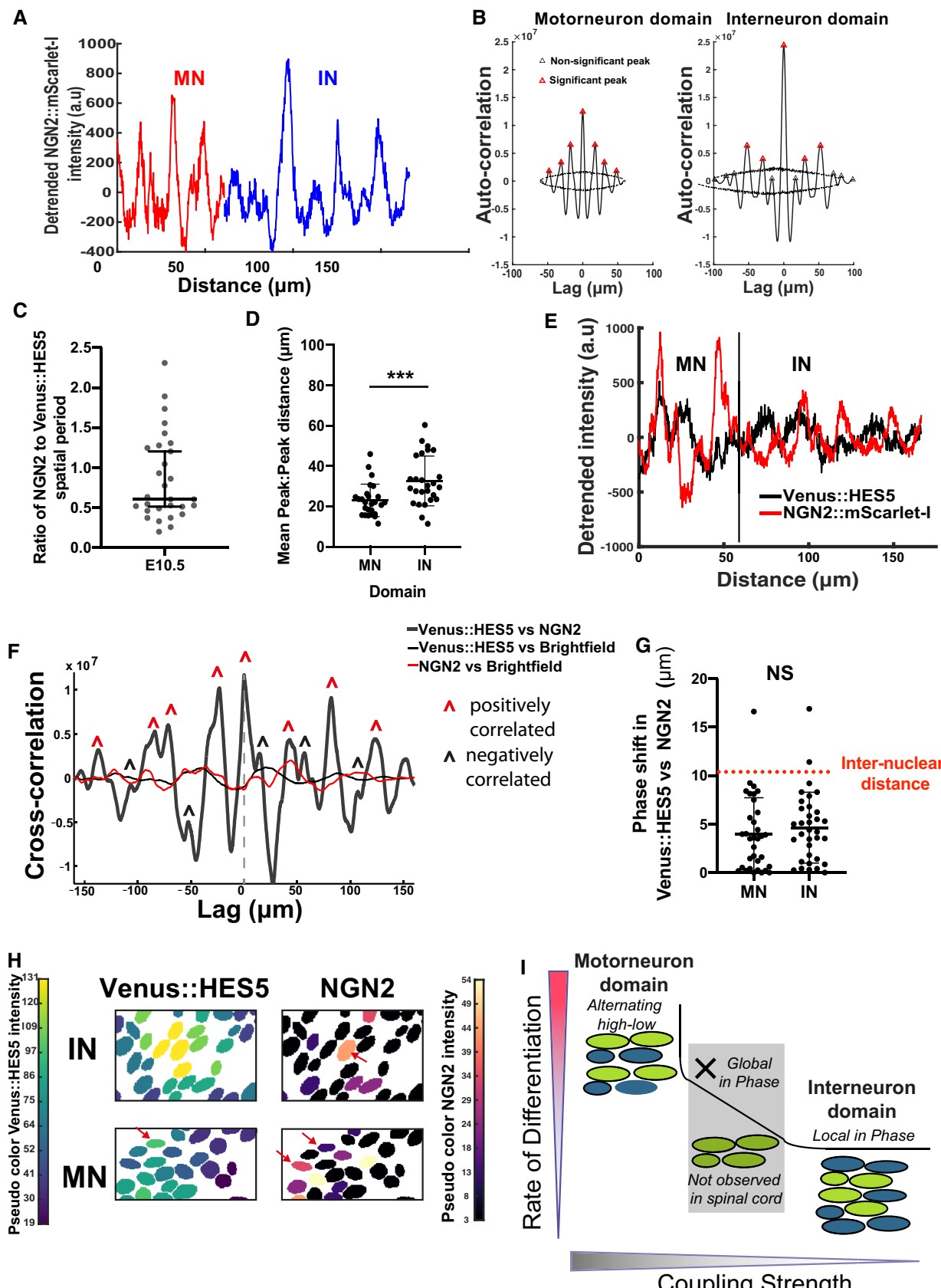

Figure 8.

single example shown for anti-phase oscillations. Thus, the discrepancy could be down to a difference in scale of analysis or perhaps to the different molecules studied. Interestingly single-cell resolution snapshot data from chick embryos appears to be consistent with the presence of microclusters (Baek *et al*, 2018).

Second, we have found that HES5 microclusters are arrayed in a spatially periodic pattern along the D-V axis of the ventral HES5 domain, meaning that high and low HES5 expression clusters alternate regularly in space. We also found that NGN2 is expressed periodically along the D-V axis with half the periodicity of HES5 such that NGN2 high cells are found both within HES5 high and low microclusters. SmiFISH showed *Dll1* expressed in microstripes but the images of single Dll1 mRNA molecules were not amenable to auto-correlation; thus, it is not known whether they occur on the same spatial scale as HES5. Multiple stripes of *Dll1* and *Jag1* and *Lfng* have been observed, but at the larger progenitor domain scale (Marklund *et al*, 2010; Ramos *et al*, 2010). Such spatial periodicity at a fine level within the ventral HES5 domain contrasts with the large-scale organisation of HES5 in 2 separate broad domains along the D-V axis (Sagner *et al*, 2018).

Thirdly, the HES5 spatial pattern of microclusters was not static but appeared dynamic over time; High HES5 expressing microclusters persisted for 6–8 h and then switched to low expression, while low expressing microclusters showed the opposite behaviour. In other words, high and low expressing microclusters alternated and sometimes created phase waves that travelled through the tissue over time. These waves are somewhat reminiscent of phase waves of LFNG and AXIN2 expression that are observed in somitogenesis (Tsiairis & Aulehla, 2016; Sonnen *et al*, 2018; Baron & Galla, 2019); however, in the spinal cord such phase waves were incoherent. This analysis was performed with a static ROI and it is possible that random movement of nuclei out of the ROI somewhat complicates the analysis of dynamic switching between high and low microcluster states. However, it is unlikely that such random behaviour could generate any of the reproducible phenomena we report in the paper. This complex spatial and temporal dynamic pattern of HES5 in spinal cord generated two important questions: how might it be generated and what might it mean for neurogenesis? Knowing that Hes genes and HES5 in particular, are activated by Notch signalling, we treated *ex vivo* spinal cord tissue with DBZ to disrupt Notch signalling (Falo-Sanjuan & Bray, 2020). We observed that the Notch inhibitor treatment extinguished spatial periodicity gradually and slowly, over a period of 10–12 h, concurrent with HES5 level downregulation. This is consistent with the amplitude death that we observed in single-cell data under the same treatment (Manning *et al*, 2019). The effect of Notch inhibition was far more pronounced in the temporal nature of the pattern; in the absence of Notch signalling, the HES5 spatially periodic pattern of low and high expressing microclusters became "frozen" in time. These findings suggest that Notch signalling plays a part in making the pattern dynamic over time but cannot account for the entire spatiotemporal complexity of HES5 expression that we see *ex vivo*.

Computational modelling helped us to explore further the role of Notch in generating the spatiotemporal pattern of HES5 expression. We have used a simplified multiscale stochastic model of HES5 self-repression and inter-cellular coupling with delay, parameterised on our own experimental data, namely the single-cell HES5 temporal period and extent of HES5 expression synchronicity between cells using the KOP. With this model, we were able to explore the influence of the coupling strength between cells in producing spatiotemporal HES5 expression patterns. We found multiple spatiotemporal patterns, namely; an alternating high and low pattern (at high coupling strength), global tissue synchronisation (at mid coupling strength) and un-coordinated pattern (at no coupling), see Movie EV2. Importantly, at weak coupling strength and inter-cellular time delay that is consistent with previous reports, we observed the emergence of dynamic microclusters that matched our experimental observations. The emergence of dynamic patterns that do not resolve into steady HES "on" or "off" static patterns has been previously observed in a stochastic multicellular tissue model combining Notch–Delta and Hes auto-repression but not confirmed in tissue (Tiedemann *et al*, 2017). However, the dynamic microclusters in our model occurred infrequently (with a probability of 20–30%) even though the model takes into consideration stochasticity and time delays; two features that represent the tissue context well. The low frequency of clusters did not improve after detailed optimised exploration of parameter space, which led us to conclude that a Notch-based cell-to-cell signalling with the assumptions we have made, recapitulates only part of the observed pattern *in vivo*. Extension of the model to include (i) longer-range cell–cell interaction via cytonemes, or due to the elongated shape of the progenitor cells, and (ii) increased complexity of the gene network such as cis inhibition between Delta–Notch or differences in signalling between different Notch ligands, may be able to increase the fidelity of microcluster emergence. Indeed, it has been shown that such modifications increase the range of spatial patterns that can be obtained (De Joussineau *et al*, 2003; Cohen *et al*, 2010; Sprinzak *et al*, 2010; Petrovic *et al*, 2014; Boareto *et al*, 2015; Hadjivasiliou *et al*, 2019). Other ways in which the model can be extended is to incorporate the influence of morphogen signalling gradients along the D-V axis or differentiation gradients along the A-B axis, as these are known to exist in the tissue.

Nevertheless, the computational model we developed, allowed us to explore the advantages that organisation in dynamic microclusters may offer as a developmental strategy for neurogenesis in the embryonic spinal cord. Overall, we found that the spatiotemporal HES5 pattern was affected by the coupling strength between cells and in turn, affected the rate of differentiation. Based on our findings, we propose that a classic lateral inhibition alternating high–low HES5 pattern (achieved at high coupling strength) shows the highest rate of differentiation because it generates two HES5 states ("on" and "off") in a spatially alternating pattern and this is likely to result in tipping of more cells towards differentiation. Global synchronisation (medium coupling strength) shows a medium rate of differentiation; however, this regime is not observed in spinal cord data perhaps because the synchronous differentiation in "blocks" of cells found close by in tissue, although an appropriate developmental strategy for somitogenesis, may be incompatible with the structural integrity of the neural tissue or the finer diversification of neuronal fates within each domain. The un-coordinated pattern (no coupling between cells) has similar rates of differentiation as weak coupling; however, weak coupling strength is advantageous because it allows local in phase synchronisation, which by analogy to global synchronisation (Fig 6C, *Global vs Local in phase*), appears to transiently increase the amplitude of temporal oscillations in HES5 expression (Fig 5I, *panel 3-Local in phase*). This is

important because a transient amplitude increase (due to the presence of microclusters at *Local in phase* conditions) could facilitate the progression to differentiation. Indeed, we have previously shown that HES5 oscillations in proliferating spinal cord progenitors have low amplitude and show mainly aperiodic fluctuations (noisy dynamics) but the propensity to oscillate as well as the peak-to-trough amplitude increases as cells enter the differentiation pathway (Manning *et al*, 2019). We have also shown that when the transition from noisy dynamic expression to oscillatory expression does not take place, progenitor cells are unable to downregulate HES levels and differentiate (Soto *et al*, 2020). We speculate that microclusters may act to reliably select one or two cells that go on to express NGN2 and differentiate and that the spatial periodicity of microclusters may space out differentiating cells to maintain tissue organisation.

We tested the model hypothesis that by changing the HES5 spatiotemporal pattern through tuning the coupling strength, the tissue is able to fine tune the rate of neurogenesis. We compared the motorneuron and interneuron progenitor domains as these two neighbouring domains in the D-V axis are known to have different rates of differentiation (Kicheva *et al*, 2014; Kuzmicz-Kowalska & Kicheva, 2020). Indeed, we find that that in the MN domain where the rate of differentiation is highest at E10.5, the HES5 and NGN2 pattern most closely matches the alternating high–low pattern (Fig 8 I, MN). In the ventral interneuron domain, we propose that the local in phase synchronisation pattern (predicted to occur at weak coupling strength) is the closest match to the *ex vivo* situation (Fig 8 I, IN). We propose it represents a strategy to balance prolonged neurogenesis, with a reasonable rate of differentiation and a transient increase in oscillation amplitude that is suitable for decoding by downstream genes. There may be additional molecular differences between the motorneuron and interneuron domains that regulate the rate of differentiation. Indeed, the transcription factor OLIG2 is expressed in the motorneuron domain and has been shown to promote differentiation by directly inhibiting HES5 (Sagner *et al*, 2018). We speculate that this mechanism could interplay or directly affect the cell–cell coupling strength by changing HES5 levels or binding partners.

In conclusion, our findings show HES5 spatially periodic micro-patterns exist in the developing spinal cord, they underlie the rate of neurogenesis and are an emergent property of the multiscale synthesis of dynamical gene expression and Notch coupling. The characterisation of this temporally dynamic expression is a testament to the power of live tissue imaging in providing mechanistic insights of complex phenomena as they unfold in real time.

# Materials and Methods

### Animals

Animal experiments were performed by personal licence holders under UK Home Office project licence PPL70/8858 and within the conditions of the Animal (Scientific Procedures) Act 1986. Venus:: HES5 knock-in mice (ICR. Cg-Hes5<tm1(venus)Imayo>) were obtained from Riken Biological Resource Centre, Japan and maintained as a homozygous line. In these mice, the mVenus fluorescent protein is fused to the N terminus of endogenous HES5. Sox1Cre:

ERT2 mice (Sox1tm3(cre/ERT2)Vep) were obtained from James Briscoe with the permission of Robin Lovell-Badge. R26R-H2B:: mCherry mice were obtained as frozen embryos from Riken Centre for Life Science Technologies, Japan and C57Bl6 mice were used as surrogates. NGN2::mScarlet-I mouse was generated by the University of Manchester Genome Editing Unit (see Appendix Supplementary Methods 1 and Appendix Fig S8). The mScarlet-I fluorescent protein is fused to the C terminus of endogenous NGN2.

### Embryo slicing and live imaging

E0.5 was considered as midday on the day a plug was detected. For matings with R26R-H2B::mCherry Sox1Cre:ERT2, intra-peritoneal injection of pregnant females with 2.5 mg Tamoxifen (Sigma) was performed 18 h prior to embryo dissection. This enables single-cell tracking through mosaic labelling of nuclei with H2B::mCherry. Whole embryos were screened for H2B::mCherry expression using Fluar 10×/0.5 objective on a Zeiss LSM880 confocal microscope. After decapitation, embryo bodies were embedded in 4% low-gelling temperature agarose (Sigma) containing 5 mg/ml glucose (Sigma). 200 µm transverse slices of the trunk containing the spinal cord around the forelimb region were obtained with the Leica VT1000S vibratome and released from the agarose. Embryo and slice manipulation were performed in phenol-red free L-15 media (Thermo Fisher Scientific) on ice and the vibratome slicing was performed in chilled 1×PBS (Thermo Fisher Scientific).

For snapshot imaging of live E10.5 spinal cord, slices were stained with 50 µM Draq5 (Abcam—ab108410) in 1×PBS (Thermo Fisher Scientific) for 1.5 h on ice if required and then placed directly on to a 35 mm glass-bottomed dish (Greiner BioOne). Images were acquired with a Zeiss LSM880 microscope and C-Apochromat 40× 1.2 NA water objective. E10.5 spinal cord slices for live timelapse microscopy were placed on a 12 mm Millicell cell culture insert (MerckMillipore) in a 35 mm glass-bottomed dish (Greiner BioOne) incubated at 37°C and 5% $CO_2$. The legs of the cell culture insert were sanded down to decrease the distance from the glass to the tissue. 1.5 ml of DMEM F-12 (Thermo Fisher Scientific) media containing 4.5 mg/ml glucose, 1× MEM non-essential amino acids (Thermo Fisher Scientific), 120 µg/ml Bovine Album Fraction V (Thermo Fisher Scientific), 55 µM 2-mercaptoethanol, 1× GlutaMAX (Thermo Fisher Scientific), 0.5× B27 and 0.5× N2 was added. Movies were acquired using Zeiss LSM880 microscope and GaAsP detectors. A Plan-Apochromat 20× 0.8 NA objective with a pinhole of 5AU was used. 10 z-sections with 7.5 µm interval were acquired every 15 min for 18–24 h. DMSO (Sigma) or 2 µM DBZ (Tocris) was added to media immediately before imaging.

### Single-cell tracking over time

Single neural progenitor cells in E10.5 spinal cord slices were tracked in Imaris on the H2B::mCherry channel using the "Spots" function with background subtraction and the Brownian motion algorithm. Tracking on the H2B::mCherry signal ensured no bias in the levels of Venus::HES5 in tracked cells. All tracks were manually curated to ensure accurate single-cell tracking. Background fluorescence was measured via an ROI drawn on a non-Venus::HES5 expressing region on the tissue and subtracted from spot intensity. To account for any photobleaching and allow comparison of

intensities between movies, the mean intensity of mCherry and Venus in each spot was normalised to the mean intensity of mCherry or Venus in the whole tissue. The whole tissue volume was tracked using the "Surfaces" and "Track over time" function.

## Immunofluorescent staining

Trunks of E10.5 embryos for cryosectioning were fixed in 4% PFA for 1 h at 4°C, followed by three quick washes with 1×PBS and 1 longer wash for 1 h at 4°C. Embryos were equilibrated overnight in 30% sucrose (Sigma) at 4°C before mounting in Tissue-Tek OCT (Sakura) in cryomoulds and freezing at −80°C. 12 µm sections were cut on Leica CM3050S cryostat. E10.5 spinal cord slices cultured on Millicell inserts were fixed in 4% PFA for 4 h. For staining, tissue and sections were washed in PBS followed by permeabilisation in PBS 0.2% Triton X-100 (Sigma) and blocking with PBS 0.05% Tween20 (Sigma) + 5% BSA (Sigma). Primary and secondary antibodies were diluted in PBS 0.05% Tween20 + 5% BSA. Tissue was incubated with primary antibodies overnight at 4°C, then washed three times for 5–10 min in PBS 0.05% Tween20, incubated with secondary antibodies and DAPI (Sigma) for 6 h at room temperature, and washed again three times in PBS-T. Sections were mounted using mowiol 4–88 (Sigma). Primary antibodies used were rabbit anti-SOX2 (ab97959, 1:200), rabbit anti-OLIG2 (EMD Millipore AB9610, 1:200) and goat anti-NGN2 (Santa Cruz Biotechnology sc-19233, 1:200).

## smiFISH probe design and synthesis

The smiFISH probes were designed using the probe design tool at http://www.biosearchtech.com/stellarisdesigner/. Depending on the GC content of the input sequence, the software can return varied size of probes, 18 and 22 nt, hence giving the largest number of probes at the maximum masking level. It also uses genome information for the given organism to avoid probes with potential off-target binding sites. Using the respective gene mature mRNA sequence, we designed 36 probes for Hes5 and 48 probes for Dll1 (Appendix Table S5) and added a FLAP sequence (5'-CCTCCTAAGTTTCGAGCTGGACT CAGTG-3') to the 5' of each gene-specific sequence (IDT). The designed set of probes were labelled with Quasar 670 (Biosearch Technologies) for Hes5 and CalFluor 610 (Biosearch Technologies) for Dll1 following the protocol from Marra *et al*, 2019.

## smiFISH on mouse sections

smiFISH protocol for mouse section embryos was developed by adapting smiFISH protocol from (Marra *et al*, 2019) and (Lyubimova *et al*, 2013). 50-µm-thick sections of E10.5 spinal cord were collected and transferred onto superfrost glass slides (VWR 631-0448) and kept at −80°C. Sections were left at room temperature to dry for 5–10 min and then fixed in 4% formaldehyde in 1× PBS followed by two quick washes in 1×PBS. 1:2,000 dilution of proteinase K (20 mg/ml stock) in 1× PBS was pipetted onto each slide and left for 5–10 min followed by two washes in 2× SCC. Sections were then incubated at 37°C twice in wash buffer (5 ml of 20× SSC, 5 ml of formamide and 45 ml of deionised, nuclease-free water). 250 µl of hybridisation buffer (1 g dextran sulphate, 1 ml 20× SSC, 1 ml deionised formamide, 7.5 ml nuclease-free water) with 100–240 nM the

fluorescent smiFISH probes was pipetted onto each slide and incubated overnight at 37°C in a humid container shielded from light. Samples were then washed as follows: twice in wash buffer at 37°C for 3 min, twice in wash buffer at 37°C for 30 min and one wash in 1× PBS at room temperature for 5 min. After smiFISH staining, sections were washed for 2 min in PBS and mounted using Prolong Diamond Antifade Mountant with DAPI (Thermo Fisher P36962).

## smiFISH microscopy and deconvolution

smiFISH images were collected with Leica TCS SP8-inverted confocal microscope using objective HC PL APO CS2 40×/1.30 oil. We acquired three-dimensional stacks 2,048 × 1,024 pixels and z size 0.4 µm. The voxel size was 0.19 × 0.19 × 0.4 µm. Quasar 670 and CalFluor 610 were imaged with pinhole 1 Airy Unit. Channels were sequentially imaged. Deconvolution of confocal images was performed using Huygens Professional Software. As pre-processing steps, the images were adjusted for the "microscopic parameters" and for additional restoration such as "object stabiliser"; the latter was used to adjust for any drift during imaging. Following, we used the deconvolution Wizard tool, the two main factors to adjust during deconvolution were the background values and the signal-to-noise ratio. Background was manually measured for every image and channel, while the optimal signal-to-noise ratio identified for the images was value 3. After deconvolution, the images were generated with Imaris 9.3

## Microcluster quantification

The number of cells in HES5 microclusters were automatically determined from images of Venus::HES5 spinal cord tissue stained with the live nuclear marker Draq5. First individual Draq5[+] nuclei were manually segmented as ellipses using ImageJ, converted to a mask and subsequently eroded using the ImageJ function "erode" to ensure no overlap between nuclei. The mask was applied to the Venus::HES5 channel generating images of nuclei with the raw Venus::HES5 intensities. Next, these segmented images were imported into MATLAB and analysed using custom scripts (available on GitHub see "Data availability") with the following steps. (i) Dead cells were excluded by removing nuclei with outlying high Draq5 intensity (>top 4% of intensity per slice) indicative of increased membrane permeability and condensed chromatin. (ii) Mean Venus::HES5 intensity was calculated per segmented nuclei. (iii) Intensity distributions of mean Venus::HES5 nuclei intensity were quantile normalised between experiments using the "quantilenorm" function in MATLAB. This ensured that the intensity in each experiment was adjusted to the same range and thus allowing consistent colormaping. (iv) Normalised mean Venus::HES5 intensities were displayed using the "viridis" (Venus::HES5) or "magma" (NGN2) colormap. The colormap was split in to six colour levels, such that nuclei within 80–120% intensity range of each other were given the same colour. This range was chosen because it matches the amplitude of Venus::HES5 ultradian oscillations (see Manning *et al*, 2019). (v) Microclusters were segmented separately for the top two intensity bins. The automated clustering approach emulated manual clustering by grouping together cells with similar intensity into a microcluster. We defined a microcluster as a minimum of two cells with the binned intensity for which there is a direct path

between the centre of the nucleus that does not intersect cells of different binned intensities (interceding cells). In the automated approach, for a specific binned intensity level, the nuclei found within less than 2.5 of average inter-nuclear distance (dmax) of each other were assigned to a microcluster. To achieve this, nuclear regions were dilated using the MATLAB routine imdilate.m with a disc structural element (generated using strel.m) of radius dmax in every direction until they merged with neighbouring nuclei forming a microcluster region. Separation between microcluster regions bounded by interceding cells of different intensity values was maintained by subtracting top 1 and top 2 nuclear regions, respectively, using the imsubtract.m routine followed by detection of connected regions using bwlabeln.m. (vi) The number of cells within a cluster was counted by testing overlap between the microcluster mask and the nuclear regions corresponding to individual nuclei to produce a nucleus-to-microcluster labelling and this is reported in Fig EV1D. (vii) Diameters in DV and AB were computed as the maximum number of nuclei observed in the x- and y-axis per microcluster. 8. Inter-cluster distances between microclusters of the same intensity level were computed in the y-axis between two or more microclusters observed along the DV axis in the same image section; specifically, we used the microcluster regions detected in step 5 and computed the centre of mass per microcluster using the routine regionprops.m with option "Centroid"; we then sorted the centroids per slice based on distance in DV and computed the distance between successive centroids; in Fig EV2F, we report the centroid to centroid distance in DV divided by inter-nuclear distance per slice.

### Microcluster detection in randomised segmented images

Using the automated microcluster detection method, we performed tests in control synthetic data (Fig EV1D). In Randomisation 1 (Rnd1), we randomly shuffled the existing intensities assigned to each nucleus, and in Randomisation 2 (Rnd2), we randomly sampled from a distribution of intensities with the same mean and standard deviation as the data. For each segmented image, we generated 20 Rnd1 and 20 Rnd2 synthetic images and performed automated counting as described in Microcluster quantification. As expected, randomised images showed dublets with only rare instances of values of three cells or above.

### Correlation of nuclear Venus::HES5 intensity with distance and neighbours

The centroids of the manually segmented nuclei were used to measure distance, and hence, rank between neighbours and a correlation of the distance and mean nuclear Venus::HES5 intensity was calculated using the "corr" function in MATLAB. Mean nuclear Venus::HES5 intensity was also randomised between nuclei before undergoing the same distance vs mean intensity correlation; randomisations were repeated five times per image.

### Centre of intensity detection and radial gradient removal

The centre of intensity (COI) was calculated using a centre of mass approach. The intensity of each nuclei was multiplied by their position. These were then summed and divided by the sum of all nuclear intensities. The COI was used to sort cells in to five equally

spaced radial zones with increasing distance from the COI. The mean Venus::HES5 intensity of nuclei in these zones was calculated and plotted against distance from the COI. For radial gradient removal, a polynomial of degree 3 was fitted to the mean zone intensity vs distance plot and the intensity subtracted from each nucleus in that zone to remove the radial gradient.

A simulated radial gradient from a single focal point in the image was generated using

$$I_r = Z + \alpha x_r,$$

where $I_r$ is the new intensity of the cells, $Z$ is simulated intensities with the mean and variance similar to that of real data, $\alpha$ is the gradient strength parameter and $x_r$ is a function of the distance from the centre of intensity. As $\alpha$ increases, the radial gradient is less affected by random deviation in HES5 expression.

### Quality controls and movie pre-processing

To remove the possibility that changes in cell positions lead to shifts in the kymograph stripes and artefacts in the dynamic analysis, movies underwent image registration to account for global tissue drift and were subject to strict quality controls for local tissue deformation. Image registration was performed in Imaris by tracking a static landmark of the tissue. Furthermore, to avoid artefacts due to local tissue deformation the average motility of tracked single cells over time in the D-V axis was compared with patterns/waves of Venus::HES5 intensity in the kymograph. A maximum threshold of 20 μm for the averaged single nuclear displacement was applied. 1 movie failed this threshold and was not used for analysis of microcluster persistence (see Appendix Table S2). Finally, bleach correction was performed using a ratiometric method in ImageJ.

### Generation of spatial expression profiles and kymographs

Spatial expression profiles and kymographs were generated in Zen Blue (Carl Zeiss Microscopy) by drawing a spline 15 μm wide starting ventrally and extending parallel to the ventricle in the dorsal direction, then using the "Line profile" or "Kymograph" function. To understand how much movement individual nuclei undergo during imaging and to help choose the width (apico-basal) of kymographs, single nuclear displacements were measured. A total of 188 individually tracked cells were obtained from three experiments (Exp1 56 cells, Exp2 54 cells, Exp3 78 cells). Tracks were 12 h long with a sampling time of 15 min (total of 49 time points). A subset of these cells were selected such that only apically located cells were included (Exp1 16 cells, Exp2 22 cells, and Exp3 27 cells). For each cell track, positional data values that were 2.5 h apart were used to determine how far a cell moves in this time window. This resulted in 39 displacement values per track, all of which the absolute value was taken and averaged across all cell tracks to give an effective root mean square (RMS) value of 7.9 μm (inter-quartile range 10.9) in apical–basal direction (summarised by experiment in Appendix Table S3).

A 15 μm width was chosen as this was larger than both a cell width and the effective root mean square displacement in 2.5 h. 0 distance corresponded to the ventral-most end of the spline. Apical, medium and basal expression profiles and kymographs were

generated from splines around 10, 30 and 60 μm from the ventricle, respectively, and analysing each side of the ventricle separately. 2–3 non-overlapping z-sections were used to generate kymographs per movie. Expression profile data for Draq5 and NGN2 from single snapshot images of live slices were generated in ImageJ using a rectangular ROI of width 15 μm and the "Plot profile" function.

**Detection and periodicity analysis of spatial expression patterns**

Kymographs were analysed using custom scripts in MATLAB and averaged along the time axis in 2 h windows. Spatial Venus::HES5 intensity in the ventral–dorsal direction was detrended by fitting a polynomial (order 4–6) and subtracting this from the raw data. This removed the larger trend due to the profile of the HES5 expression domain.

Auto-correlation and Lomb-Scargle periodograms were used to analyse periodicity of the detrended spatial intensity plots. Lomb-Scargle periodograms were generated with the MATLAB "plomb" function and plotting power level thresholds as specified in figure legends. Auto-correlation was performed with the MATLAB "xcorr" function. Auto-correlation functions were smoothed using Savitzky–Golay filter and then peaks identified using the "find peaks" function. Significant peaks were identified using a bootstrap method with 100 randomisations. Auto-correlations were randomised and then re-subjected to auto-correlation. 2 standard deviations of the auto-correlations of randomised data were used as a threshold and peaks were designated as significant if they exceeded this threshold. The mean distance between significant peaks was calculated per kymograph timepoint. Fold changes of spatial intensities were calculated between significant peaks and troughs in the signal identified using "find peaks" on the negative signal.

Splitting Venus::HES5 kymographs in to motorneuron and interneuron domains was based on staining of cryosectioned E10.5 spinal cord with motorneuron progenitor domain marker OLIG2. The peak of the trend in Venus::HES5 was found to occur on average at 35 μm dorsally from the edge of the OLIG2[+] domain. This criterion was used to split kymographs from movies of Venus::HES5 spinal cords that had not been immuno-stained.

**Correlation coefficient analysis in the anterior to posterior (A-P) axis**

We produced kymographs from multiple non-overlapping stacks extending in the AP direction using the same region of interest (ROI) which meant that Venus:HES5 intensity was comparable at the same position in DV. We used detrended Venus::HES5 averaged over 2 h per z and compared the detrended coefficients pairwise across subsequent z-stacks. Using the confocal magnification in the AP axis per experiment, we reconstructed the absolute distance between subsequent z-stacks. Data from untreated and tissue treated with DMSO were analysed in the same way.

**Hierarchical clustering of local HES5 expression and microcluster persistence time**

Kymographs of HES5 expression were split into adjacent 20 μm regions along the D-V axis and the HES5 intensity averaged in these regions to give a timeseries per region. To account for any single-

cell movement in DV, we applied a 2 μm Gaussian blur filter onto the kymograph data using the MATLAB routine *imgaussfilt.m* prior to extracting timeseries per region. These timeseries were normalised to the mean and standard deviation of each region over time (z-scoring) and subject to hierarchical clustering using the *clustergram.m* routine in MATLAB with Euclidean distance and average linkage. The persistence time was calculated as continuous time when the signal in the region was above (high) or below (low) its mean level. The persistence ratio was calculated as the time interval spent in a high state divided by the time interval spent in a low state within the same 20 μm region. Where only high or low persistence time intervals were detected in a region, these observations were excluded from the ratio. We also used an alternative method to compute persistence time relying on zero-crossing of the detrended Venus::HES5 signal averaged over 0 to 2 h timepoints; in this approach, we identified specific areas containing a microcluster with high expression (above the mean) and low expression (below the mean) and repeated the persistence time calculation as described above.

**Phase mapping of kymograph Data**

We used kymograph data (see Generation of spatial expression profiles and kymographs) to produce spatiotemporal phase mapping from Untreated tissue (Fig EV2L and Appendix Fig S3) as well as DMSO vs DBZ (Figs 4 and EV3). Firstly, kymograph data were averaged over 2 h to produce low temporal resolution information in the dorsal–ventral direction. The resulting spatial signal was detrended in the DV direction using a polynomial order 4 and smoothed using a Savitzky–Golay filter. Phase reconstructions were obtained from DV signal for every 2 h timeblock using the Hilbert transform, and these were presented as a colormap indicating time on the x-axis and space on the *y*-axis. We refer to this as phase mapping and it enables detection of phase resets (indicative of changes from high to low) in the same region over time.

**Phase–phase mapping and phase shift analysis in cell pairs**

We analysed Venus::HES5 ultradian dynamics using the approach in Manning *et al,* 2019, Phillips *et al.* 2017. Specifically, we used a Gaussian Processes pipeline to fit the single-cell trend of Venus::HES5 expression (examples shown in Appendix Fig S4C). We performed detrending of Venus::HES5, followed by z-scoring and estimated a periodic Ornstein–Uhlenbeck covariance model. This procedure produces a smooth detrended curve (examples shown in Appendix Fig S4C). Using the detrended smoothed curves, we extracted the phase shift using cross-correlation analysis of pairs of timeseries using the *xcorr.m* MATLAB routine. The phase shift corresponded to the lag time interval closest to 0 at which the cross-correlation function shows a peak. From detrended smooth curves, we then performed Hilbert reconstruction of instantaneous phase using the *hilbert.m* MATLAB routine. We used the phase angles corresponding to neighbouring cell pairs at multiple timepoints to produce a phase–phase mapping. We plotted the density of the phase map using the *dscatter.m* routine with 24 × 24 binning of phase values (Eilers & Goeman, 2004). This approach (Hilbert and dscatter) has been previously described in Sonnen *et al,* 2018. Cells pairs were identified based on the median 3-dimensional Euclidean

distance <20 µm across the whole timeseries. Hence, we also performed a phase–phase analysis using cross-pairing whereby cells in the same experiment were paired with cells found further than 20 µm away (cell1:pair1 versus cell1:pair 2; cell2:pair1 versus cell 2:pair 2 etc.). Phase distributions of proximal pairs (Fig 3J) and those obtained by cross-pairing (Appendix Fig S4F) were compared in likelihood of observations in-phase versus observations out-of-phase at all phase angles. Regions of phase–phase mapping corresponding to in-phase and out-of-phase are outlined in Appendix Fig S4F. Likelihood of cross-paired tests showed values close to 1 indicating no predominant in-phase activity whereas values for paired data were significantly higher.

### Stochastic multicellular HES5 model with time delay

The core unit of the multicellular model is a single-cell unit that explicitly models Hes5 protein and mRNA abundance and is adapted from the work done in Manning *et al*, 2019. The single-cell model makes use of a Langevin approach to include stochastic fluctuations in both protein and mRNA as well as the inclusion of a time delay associated with the inhibitory Hill function used to describe the repressive action of Hes protein on its own mRNA production. This implementation, along with the parameter inference (Manning *et al*, 2019), results in a single-cell model capable of reproducing stochastic oscillations closely matched with the single-cell dynamics observed in the developing neural tube. The multicellular approach extends the single-cell model by introducing an inhibitory Hill function to couple nearest-neighbour cells (in a fixed, no cell movement, hexagonal geometry) whereby high Hes5 protein in one cell is able to repress Hes5 mRNA production in a neighbouring cell (see Appendix Supplementary Methods 2). This inhibitory Hill function (the coupling function) is representative of the overall behaviour of the Notch Delta pathway and its interaction with Hes5, allowing for the bidirectional interaction of Hes5 dynamics between neighbouring cells. Three parameters are associated with this Hill function that make it flexible enough to explore different possible coupling realisations of the Notch–Delta pathway, the effects of which are illustrated in Fig 5B. The main parameter modulated for the analysis in this paper is the repression threshold which defines the number of protein molecules that is required to repress mRNA in a neighbouring cell.

### Cell-to-cell HES5 differences by domain and by coupling strength

We used raw Venus::HES5 data, absolute HES5 quantitation by Fluorescence Correlation Spectroscopy (FCS) and manually segmented nuclear maps made available in (Manning *et al*, 2019). We obtained average HES5 concentration per nuclei by quantile–quantile matching the Venus distribution to the reference FCS distribution of HES5 levels across the tissue. Using nuclear centroid location, we produced absolute cell-to-cell concentration differences between every cell and its closest neighbour. We performed a by domain analysis by dividing the cell-to-cell concentration differences by the average HES5 concentration by domain. In the synthetic examples, HES5 molecular abundance data obtained from the multicellular model were used to produce absolute cell-to-cell abundance differences over a range of coupling strength values. We also produced synthetic cell-to-cell abundance differences relative to the mean HES5 abundance per simulation over a range of coupling strength values.

### Phase reconstruction and Kuramoto order value as a measure of synchrony

To determine the synchronisation of real signals both in the model and experimental data, the phase of each oscillator was first reconstructed in complex space. This reconstruction was achieved by using the Hilbert transform, which shifts the phase of each frequency component in a signal by 90 degrees (Benedetto, 1996). The Hilbert transform of a function $u(t)$ is defined as

$$H(u)(t) = \frac{1}{\pi} \int_{-\infty}^{\infty} \frac{u(\tau)}{t - \tau} d\tau. \tag{1}$$

To obtain a rotating vector that contains both the amplitude and phase information of the signal at a given time $t$, the original signal and the 90 degrees shifted Hilbert transform can be combined in complex space to give

$$u_a(t) = u(t) + i \cdot H(u)(t). \tag{2}$$

By comparing $u_a(t)$ of two or more cells, a measure of how synchronised a population of cells is can be determined by first calculating what is known as the complex order parameter

$$\psi = \frac{1}{N} \sum_{j=1}^{N} e^{i\phi_j}, \tag{3}$$

where $N$ is the number of oscillators and $\phi_j$ is the phase of oscillator $j$. From this, the Kuramoto order parameter is defined as the absolute value of the complex order parameter $\psi$, which is the magnitude of the vector and has a value between 0 and 1 (Choi *et al*, 2000). A value of 1 indicates perfect synchrony and matching phase, meaning that in complex space the phases of each oscillator would be at the same angle and would rotate at the same frequency. A value of 0 indicates no synchronisation, and in complex space would appear as a distribution of phases that average to a point at the origin.

### Phase synchronisation index

In addition to calculating KOPs, we also used the Hilbert transform to extract phase from spatial data to determine how dynamic the positions of peak and trough were over time. This involved extracting and plotting the phase from time-averaged spatial signals. The phase synchronisation index for DMSO and DBZ conditions (Fig 4E) was obtained by calculating KOP per position in D-V axis and averaging per z-slice (with left and right of the ventricle analysed separately). To account for the loss of spatial periodicity in DBZ at later timepoints, only data passing significance for an auto-correlation test has been included resulting in an analysis restricted to periodic spatial expression observed in both DMSO and DBZ up to 10 h.

### Detrending methods

Multiple detrending methods are used depending on the type of data. Detrending and removal of the radial gradient in images of segmented Venus::HES5 nuclei (as in Fig 1) are covered in Materials and Methods section entitled "Centre of intensity detection and radial gradient

removal". The images are 2-dimensional data and so require removing trends in both the apical–basal and dorsoventral direction. Detrending of spatial profiles of Venus::HES5 and NGN2::mScarlet-I is covered in Materials and Methods "Detection and periodicity analysis of spatial expression patterns". Spatial profiles are generated from a ROI 15 μm wide in apicobasal axis and extending up to 250 μm + in dorsoventral direction. The Venus::HES5 intensity is averaged in the apicobasal axis by the image analysis software (either Zen Blue or ImageJ). This generates the 1-dimensional spatial profile and detrending is applied along the dorsoventral axis. Detrending of single-cell timeseries of Venus::HES5 expression is outlined in "Phase-phase Mapping and Phase Shift Analysis in Cell Pairs".

### Statistical testing

Statistical tests were performed in GraphPad Prism 8. Data were tested for normality with D'Agostino–Pearson test. The relevant parametric or non-parametric test was then performed. Bar plots and discrete scatter plots show mean mean±SD where multiple independent experiments are analysed. Statistical significance between 2 datasets was tested with either *t*-test (parametric) or Mann–Whitney test (non-parametric). Statistical significance ($P < 0.05$) for 2+ datasets was tested by Kruskal–Wallis with Dunn's multiple comparison correction. All tests were 2-sided. Multiple comparison testing involved comparing all pairs of data columns. Correlations were analysed using Pearson correlation coefficient. Sample sizes, experiment numbers, *P* values<0.05 and correlation coefficients are reported in each figure legend.

## Data availability

All code is written in MATLAB and is available on GitHub: https://github.com/Papalopulu-Lab/Biga2020

**Expanded View** for this article is available online.

### Acknowledgements
We are grateful to members of the Papalopulu lab, Andrew Hazel, James Briscoe and Andy Oates for advice and discussions. The authors would also like to thank Robert Lea, the Biological Services Facility and the Bioimaging Facilities of the University of Manchester for technical support. CM was supported by a Sir Henry Wellcome Fellowship (103986/Z/14/Z) and University of Manchester Presidential Fellowship. VB was supported by a Wellcome Trust Senior Research Fellowship to NP (106185/Z/14/Z). JH (220001/Z/19/Z), EJ (204057/16/Z) and DH (Wellcome Trust Grant No. 215189/Z/19/Z) were supported by Wellcome Trust PhD studentships. JK was supported by Wellcome Trust Senior Research Fellowship to NP and a University of St Andrews Lectureship. The funders had no role in study design, data collection and analysis, decision to publish, or preparation of the manuscript.

### Author contributions
Study conception and experiment design: CM and NP. Wet lab experiments, supervision, data analysis, data interpretation and manuscript writing: CM. Supervision and development of method to analyse spatial micro-patterns of HES5 expression both in data and from the model, method for Hilbert phase persistence analysis, data analysis, data interpretation and manuscript writing: VB. Design and implementation of stochastic coupled HES5 model, parameterisation, analysis and model interpretation: JH. smiFISH and imaging: XS. Data collection and development of method to analyse correlations of HES5 nuclear intensity: EJ. Method to analyse periodic spatial micro-patterns of HES5 expression: DH. Design and generation of Neurog2::mScarlet-I knock-in mouse: HB. Design and generation of Neurog2::mScarlet-I knock-in mouse: ADA. Supervision and assistance in design, analysis and interpretation of the mathematical model: JK. Supervision and assistance in analysis and interpretation of the mathematical model: PG. Supervision and work, data interpretation, and manuscript writing: CM, VB, NP, JH, JK, EJ, DH and PG.

### Conflict of interest
The authors declare that they have no conflict of interest.

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
