## [Review Process File · Molecular Systems Biology]

A dynamic, spatially periodic, micro-pattern of HES5 underlies neurogenesis in the mouse spinal cord

Veronica Biga, Joshua Hawley, Ximena Soto, Emma Johns, Daniel Han, Hayley Bennett, Antony Adamson, Jochen Kursawe, Paul Glendinning, Cerys Manning, and Nancy Papalopulu

DOI: 10.15252/msb.20209902

Corresponding author(s): Nancy Papalopulu (Nancy.Papalopulu@manchester.ac.uk), Cerys Manning (cerys.manning@manchester.ac.uk)

Review Timeline:

Submission Date:	3rd Aug 20
Editorial Decision:	24th Aug 20
Revision Received:	7th Dec 20
Editorial Decision:	15th Jan 21
Revision Received:	30th Mar 21
Accepted:	6th Apr 21

Editor: Maria Polychronidou

Transaction Report:

Thank you again for submitting your work to Molecular Systems Biology. We have now heard back from the three referees who agreed to evaluate your study. As you will see below, the reviewers acknowledge that the presented findings seem interesting. However, they think that the study remains preliminary and raise a series of concerns, which we would ask you to address in a major revision.

Both reviewers #1 and #2 point out that substantial additional analyses are required to convincingly support the reported conclusions. Specifically, the existence, origin and role of the microclusters as well as the role of Notch in the microcluster dynamics need to be better supported. Moreover, as the reviewers recommend, further experiments need to be performed to challenge and better substantiate the model's predictions. All other issues raised by the referees would need to be satisfactorily addressed. Overall, the reviewers provide constructive suggestions on how to improve the study. Please let me know in case you would like to discuss in further detail any of the issues raised.

On a more editorial level, we would ask you to address the following issues.

Reviewer #1:

A key fate transition during mammalian neurogenesis is thought to rely on the temporal dynamics (oscillations) of the transcription factor HES5. The present study reports on the spatial and temporal periodic expression HES5 in the mouse spinal cord. By combining live imaging, sophisticated image analysis and mathematical modeling, the authors suggest that HES5 is expressed in small clusters of cells that are spatially periodic, propose that this periodicity arises from the strength of local cell-cell coupling between single cell oscillators and provide correlative evidence that the rate of differentiation may depend on this coupling strength. I review below the data provided in support of these original findings.

A previous study from the Papalopulu lab indicated that a subset of the neural progenitors express HES5 with a 3-4h period in the mouse spinal cord (while other cells only undergo aperiodic fluctuations). This study, however, did not examine the relative positions of oscillating and non-oscillating cells. Here, the authors address whether these behaviors are coordinated at the tissue-level. This is an interesting and timely question. To do so, they examine the spatial distribution of HES5 expressing cells within the embryonic spinal cord using a Venus-HES5 knock-in mouse

Analysis of snapshot images first suggested the existence of small clusters of cells expressing slightly higher HES5 levels (Fig 1). These clusters of 5-8 cells were identified following manual segmentation of nuclei by visual inspection, not automatically through image processing. This is a significant caveat. Also, nuclei close to each other display correlated levels of HES5 (Fig 1f). This observation is interpreted to support the existence of microclusters. However, data shown in Fig 2 indicate that HES5 is not uniformly expressed in the tissue. Thus, one would expect positive correlation at the short range and negative correlation at the long-range, which is what is observed in Fig 1f. Thus, the analysis shown in Fig 1f needs to be performed on detrended HES5 values. Whether these clusters comprise progenitors that are oscillating in phase and/or express persistent levels of HES5 remains at this point unknown. Of note, the paragraph title, 'microclusters ... have positively correlated HES5 levels', is too obvious to be a conclusion given that microclusters were presumably visually identified on this basis. In sum, the existence of microclusters is not convincingly demonstrated. The authors need to perform a rigorous unbiased approach aiming at identifying nuclei with positively correlated HES5 levels before examining the spatial distribution of these clusters.

The authors then switched methods and performed low-resolution live imaging (Fig 2). In these conditions, nuclei were not identified and microclusters were identified as peaks in detrended HES5 intensity profiles. The median spatial period corresponds to approx 4 cells. The authors therefore suggest that clusters of 2-3 cells in a high HES5 state in synchrony are periodically found every 4 cells (Fig 2). However, while the periodicity is clear in the sample shown in Fig 2c, it is much less obvious in the other samples (Fig 3f and S3b). Also, the extent to which the automatically-defined peaks (Fig 2) correspond to the visually-defined microclusters (Fig 1) remains to be examined. Interestingly, this analysis may have been performed in Fig 7: 34 microclusters have been identified following nuclear segmentation and detrended intensity profiles were studied. How peaks and clusters compare is, however, not presented.

Next, the authors examined the time evolution of the non-detrended HES5 intensity obtained from low-resolution imaging (Fig 3a-d). The kymograph data are interpreted to suggest that groups of 2-3 cells (defined arbitrarily using 20 micron-wide ROIs) can interconvert between low and high HES5 states every 6-8 hours (Fig 3a-d). Second, the authors extracted the mean phase values of the underlying single-cell oscillators from the detrended HES5 intensity values (Fig 3f-h). The latter analysis was performed at higher spatial resolution but lower temporal resolution. Again, this

analysis supports the notion that clusters of cells can interconvert between low and high HES5 states. The authors therefore conclude that the spatial pattern of HES5 is dynamic over time. Two observations need, however, to be clarified. First, it is unclear whether clusters exhibit periodic expression in time. Given the temporal persistence of the high and low states, a 12-16h period may be expected. How do the authors relate this periodicity to the 3-4h period of HES5 expression seen at the single-cell level? Also, the authors assume that the x,y coordinates of the recorded cells/clusters do not change over the time course of the live imaging experiment. In other words, the living tissue must not locally expand, contract, deform etc... over the 12-24h period of recording: is this really the case? Actually, the kymograph shown in Fig 3b may suggest that the shape of the tissue, hence cell positions, change over time, hence generating these non-reproducible traveling waves of HES5 expression. In sum, the authors need to exclude this possibility to convincingly show that the HES5 spatial pattern is dynamic over time.

Next, the authors looked at the role of Notch using drugs. As expected, the expression levels of the Notch target HES5 are down (Fig 4a) and this effect is associated with reduced amplitude of the peaks upon detrending (Fig 4b). However, whether Notch contributes to the dynamic high-to-low and low-to-high switches at the cluster level may be hard to judge when HES5 levels are overall reduced. Thus, the contribution of Notch to the dynamic switching remains to be clarified.

Then, the authors switched to computational modelling to test whether varying the strength of cell-cell coupling between oscillators is sufficient to produce spatial and temporal periodicity (Fig 5). Exploration of the parameter space identified conditions that allow for the emergence of in-phase HES5 expression in small clusters of cells (Fig 5g). This analysis is nice and clear. Nevertheless, as the authors note, this modelling does not produce the kind of periodic spatial patterns that are reported in Fig 2. The authors further observed that both the amplitude of simulated HES5 variations in time and space depends on the coupling strength (Fig 6c). Assuming that the probability to differentiate increases when HES5 levels are below a threshold, the rate of differentiation is then expected to vary with the coupling strength (Fig 6b). To test whether this idea may apply in vivo, the authors correlated the differentiation rate observed in two domains of the spinal cord with the 'spatial amplitude' measured in these two domains in snapshot views of HES5. The observed correlation is consistent with the idea that the rate of differentiation is regulated by the coupling strength. Obviously, other regional differences may account for the observed differences in the rate of differentiation.

In sum, the ideas explored in this study are very interesting but the data do not support well enough the conclusions.

Reviewer #2:

In the current manuscript, the authors build on the previous finding that Hes5 expression is oscillatory in neuronal progenitors (Manning et al., 2019) and try to address how these oscillations unfold in the tissue level. Examining the spatial expression of Hes5 they uncover so-called microclusters of correlated expression that appear regularly arranged along the D/V axis of the developing neural tube. Moreover, these patterns do not appear static in time, as the levels of Hes5 follow different dynamics. Pharmacological manipulation with Notch signalling inhibitors reveal that communication through Notch is critical for such temporal dynamics of the microclusters while the spatial arrangement appear less affected. The authors employ a mathematical model to formalize

assumptions and hypothesis and test the potential origin and impact of microclusters on neuronal differentiation. According to this analysis the coupling strength between cells can partially explain the observed dynamics and the authors show that different spatial patterns correlate with differentiation rates.

The authors attempt to address an important and timely question, and towards these they employ an impressive toolbox applying mathematical and live imaging analysis. Their main findings are the existence of microclusters and their role in differentiation. The presence of microclusters is quite clear, although their origin is not investigated deeply enough. How much of the pattern can be attributed to clonal expansion of some cells, and what role does the cell migration play? While the authors emphasize Notch signalling to explain their formation, it is clear from their data that the spatial pattern cannot be attributed to this. Seeing the expression pattern of *olig2*, and *ngn* in Suppl fig 7 it is clear that some form of spatial pattern is definitely not specific to *Hes5*. The authors skip critical questions on the microclusters, i.e. what other genes show similar pattern? Are they correlating/anticorrelating with *Hes5*? Some of the interesting candidates would be *Delta*. If *Delta* expression doesn't follow the *Hes5* expression, as they imply in the discussion, it will be clear that the pattern is not emerging through Notch signalling interaction.

The role that the micropatterns play is even more blurry and safe conclusions are difficult to draw. Important questions on the fate of one microcluster's cells remain unanswered. The implication from the authors' model is that the cells of the microcluster, having coherent *Hes5* levels, would differentiate in unison. It would be important to examine if this is the case. The mathematical model is also sophisticated and is built on reasonable assumptions but the comparison to experimental results is problematic. In particular, the authors see that different patterns correlate with different differentiation rates. They are then too fast to assign a causal link through the model, without any experimental validation. According to their hypothesis the pattern depends on the coupling strength, the ultimate cause of the pattern and the variable differentiation rate. However, the coupling strength is not probed. Since the coupling is expected to be mediated by notch signalling, one would expect that its manipulation can change the pattern and differentiation rate. Such experiments are critical in order to substantiate the links drawn through the model's hypotheses. Overall, I find the attempt very interesting but not very convincing. The authors would benefit from deeper examination of the origin and role of the microclusters and I would recommend publication after serious revisions. Other specific points that need to be addressed are following:

1. The authors remove the signal trend by fitting a polynomial. What is the polynomial's grade?
2. In figure 2d an interesting application of the LS periodogram is exploited to detect the spacing of the microclusters. It may be possible to use this further and examine which, if any, of the significant peaks are detected in multiple samples. This will give further credibility on the uncovered periodicity.
3. In figure 2 g-i there is analysis of the intermediate and basal domains along the D/V axis, in addition to the apical. It may be worth performing LS periodograms for these signals and compare the detected peaks.
4. For figure 3, the analysis is done in blocks of 20 μ m in width. While this is in principle fine, it is not clear why this particular segmentation is the one to perform. What would happen if the windows are shifted by 2 or 3 μ m? I think a better approach would be to apply a gaussian blur filter on the kymograph.
5. In figure 3c there are 2 big groupings of the temporal behaviour, as increasing or decreasing. I am wondering whether this is an oversimplification, as buried in these patterns are oscillatory trends. In fact, such dynamics are commented on, in the main text ("...revealed changes from low to high Venus::HES5, high to low, or fluctuations between the two states...") but are not highlighted in the figure.
6. I would recommend a rainbow LUT (HSV in MATLAB) instead of the one used from blue to yellow. It would have the advantage of continuity between 0 and 2 π .
7. The term "salt and pepper" is not the most informative for the described case in the

mathematical model. The term "checkerboard-like" that the authors first use in the discussion is much more accurate and should be used throughout the manuscript.

8. When explaining the Kuramoto order parameter in the materials and methods section (p.28), the authors state that "A value of 0 indicates no synchronization, and in complex space would appear as a random distribution of phases that average to a point at the origin". This is not correct and I would remove the word "random", because a value of 0 can emerge from properly arranged phases, i.e. dividing the 2π cycle in equal n intervals and placing n oscillators' phases at the interval boundaries. Importantly, the oscillators are synchronized out-of-phase and thus a value 0 indicates no in-phase synchronization.

9. In Suppl fig.1, panel g is not accounted for in the legend or the main text.

Reviewer #3:

It has been shown that HES5 expression oscillates in neural progenitors, but its multiscale pattern is not known in the developing nervous system. The authors sought to determine the HES5 expression patterns in the developing spinal cord by live imaging and found the existence of spatially periodic and temporally dynamic microclusters of HES5 expression. Using the computational modeling and experimental data, the authors showed that Notch signaling-mediated local synchronization is responsible for formation of such microclusters, and that the HES5 spatial patterns correlate with the rate of cell differentiation.

By taking advantage of a powerful live imaging technique, the authors for the first time demonstrated the local synchronization of HES5 expression in the developing nervous system. This is an interesting and elegant work showing the significance of the HES5 spatial patterns in cell differentiation. The results are of general interest but need some clarification. Specific comments are indicated below.

1. In Figure 1, the authors should show whether or not HES5 microclusters continue like stripes along the z axis (anterior-posterior axis). If this is the case, it is interesting to image HES5 expression from the apical side by opening the neural tube.

2. The authors showed that Notch inhibition disrupts HES5 microclusters (Figure 4), but it is not clear how Notch signaling regulates different HES5 spatial patterns. The authors suggested that different coupling strength leads to different patterns like microclusters or salt-and-pepper conditions (Figure 5g), but this could be explained by different levels or types of Notch ligands. The authors should examine this possibility.

3. In the mathematical model in Figure 5, cells are steadily present in the same place. However, in the apical region, cells only in late G2-M-early G1 are present. Thus, it is likely that many cells come and go due to interkinetic nuclear migration every hour or so in each microcluster. The authors should clarify whether this mathematical model represents the *in vivo* condition of the developing spinal cord.

4. In Figure 5, depending on the inter-cellular time delay, $2\pi/3$ phase difference of HES5 oscillations could be stable. Did the authors observe such phase difference between cells, particularly when HES5 microclusters are absent?

Editor's comments

Both reviewers #1 and #2 point out that substantial additional analyses are required to convincingly support the reported conclusions. Specifically, the existence, origin and role of the microclusters as well as the role of Notch in the microcluster dynamics need to be better supported. Moreover, as the reviewers recommend, further experiments need to be performed to challenge and better substantiate the model's predictions. All other issues raised by the referees would need to be satisfactorily addressed. Overall, the reviewers provide constructive suggestions on how to improve the study. Please let me know in case you would like to discuss in further detail any of the issues raised.

We have also found the reviewer's comments constructive and they helped us to improve the paper by additional experimentation, analysis and textual additions/changes resulting in several new figure panels, new figures and new supplementary figures. Thank you for giving us the opportunity to respond to these comments. Our detailed response is shown below. All new data have also been added to the paper in the places indicated.

Reviewer's comments

Reviewer #1:

A key fate transition during mammalian neurogenesis is thought to rely on the temporal dynamics (oscillations) of the transcription factor HES5. The present study reports on the spatial and temporal periodic expression HES5 in the mouse spinal cord. By combining live imaging, sophisticated image analysis and mathematical modeling, the authors suggest that HES5 is expressed in small clusters of cells that are spatially periodic, propose that this periodicity arises from the strength of local cell-cell coupling between single cell oscillators and provide correlative evidence that the rate of differentiation may depend on this coupling strength. I review below the data provided in support of these original findings.

A previous study from the Papalopulu lab indicated that a subset of the neural progenitors express HES5 with a 3-4h period in the mouse spinal cord (while other cells only undergo aperiodic fluctuations). This study, however, did not examine the relative positions of oscillating and non-oscillating cells. Here, the authors address whether these behaviors are coordinated at the tissue-level. This is an interesting and timely question. To do so, they examine the spatial distribution of HES5 expressing cells within the embryonic spinal cord using a Venus-HES5 knock-in mouse

Analysis of snapshot images first suggested the existence of small clusters of cells expressing slightly higher HES5 levels (Fig 1). These clusters of 5-8 cells were identified following manual segmentation of nuclei by visual inspection, not automatically through image processing. This is a significant caveat. Also, nuclei close to each other display correlated levels of HES5 (Fig 1f). This observation is interpreted to support the existence of microclusters. However, data shown in Fig 2 indicate that HES5 is not uniformly expressed in the tissue. Thus, one would expect positive correlation at the short range and negative correlation at the long-range, which is what is observed in Fig 1f. Thus, the analysis shown in Fig 1f needs to be performed on detrended HES5 values.

Yes, the reviewer is correct in saying that given non-uniform expression of Venus::HES5 in the tissue, a positive correlation between Venus::HES5 intensities at short range and negative correlation at the long-range would be expected and that the analysis should be done in de-trended values. In fact, this is exactly what we did in the original manuscript, but we apologise if it was not clear. Specifically, we analysed Venus::HES5 correlation between nuclei on data in which the influence of the trend, generated by the shape of the expression domain, had been excluded. The trend was removed by finding a centre of intensity of nuclei in the image and sorting cells in to 5 equally spaced radial zones. The mean Venus::HES5 intensity of nuclei in these zones was calculated and then subtracted from each nucleus in that zone to remove the radial gradient accounting for gradual differences in intensity both in the anterior to posterior as well as the apical to basal axis, the latter due to cells downregulating HES5 as they differentiate (see Appendix Supplementary Methods). The rearranged Figure EV1f shows an example of an experimental radial gradient from one image. Reassuringly, we found that positive correlations between cells within 25 μ m still persist after removing the trend (**Figure 1j**). In the revised manuscript we have added new data to show this is also true for the Venus::HES5 spatial pattern at E9.5 and E11.5 (**Figure EV1 g,h**). A description of these new results has been added to the manuscript on page 6.

Whether these clusters comprise progenitors that are oscillating in phase and/or express persistent levels of HES5 remains at this point unknown.

Thank you for raising this question and giving us an opportunity to investigate it further in the experimental data. To interrogate this point, we exploited the availability of sparsely tracked single cell timeseries and phase reconstruction methodology previously shown in Manning et al 2019. In the revised manuscript, we include new analysis to show that pairs of cells staying in close proximity with each other show predominantly in-phase Venus::HES5 oscillations/fluctuations (amended **Figure 3 new panels k,l**). We have previously described that single cell HES5 timeseries show aperiodic fluctuations with only approximately 40% of progenitors passing a stringent oscillatory test (Manning et al. 2019). Within the newly added cell pairs dataset, this results in a noisy phase-mapping (**Figure 3l**) that nevertheless shows prevalence of in-phase activity and this was confirmed using cross-correlation analysis (new **Appendix Figure S2c**). Therefore, we conclude that microclusters comprise of cells that locally synchronise in their phase as well as positively correlated levels of Venus::HES5 due to changes in the long-term trends of the levels of Venus::HES5 over time (i.e. the microcluster “persistence” time, discussed further below). These revisions have been added to the manuscript on page 10.

Of note, the paragraph title, 'microclusters ... have positively correlated HES5 levels', is too obvious to be a conclusion given that microclusters were presumably visually identified on this basis. In sum, the existence of microclusters is not convincingly demonstrated. The authors need to perform a rigorous unbiased approach aiming at identifying nuclei with positively correlated HES5 levels before examining the spatial distribution of these clusters

Our method for microcluster detection is a rigorous and unbiased approach. First, the nuclei were manually segmented, which although laborious means that each one was individually inspected. Second, we used clearly-defined and objective criteria to identify and count the number of cells per cluster (see Appendix Supplementary Methods). Briefly, the intensity distributions of mean Venus::HES5 nuclei intensity were quantile normalised between experiments. Then a look-up table was applied so that cells within 80-120% intensity range of each other were given the same colour. We have chosen this range because it matches the amplitude of Venus::HES5 fluctuations (see Manning et al. 2019).

We have however modified the title of section “Microclusters of spinal cord progenitor cells have positively correlated HES5 levels” to “Positive correlations in Venus::HES5 intensity are indicative of microclusters in spinal cord tissue”. This is found in the manuscript on page 5.

The authors then switched methods and performed low-resolution live imaging (Fig 2). In these conditions, nuclei were not identified and microclusters were identified as peaks in detrended HES5 intensity profiles. The median spatial period corresponds to approx 4 cells. The authors therefore suggest that clusters of 2-3 cells in a high HES5 state in synchrony are periodically found every 4 cells (Fig 2). However, while the periodicity is clear in the sample shown in Fig 2c, it is much less obvious in the other samples (Fig 3f and S3b).

We agree with the reviewer that spatial periodicity is variable between samples, most likely because it is a highly dynamic process. In the original manuscript, we defined a statistical criterion for spatial periodicity based on randomising intensities to produce confidence bounds and identify in a quantitative manner significant peaks in the auto-correlation function (Appendix Supplementary Methods). Our criterion is unbiased and has > 95% statistical significance level and in the original manuscript we have provided evidence that 100% of DMSO control kymographs show significant periodicity using this criterion (Figure 4f). In the revised version, we have included a new **Appendix Table S1** that shows that 100% of kymographs from untreated tissue have significant auto-correlation peaks detected at multiple timepoints. At the suggestion of reviewer 2 we have also confirmed our findings using an independent frequency analysis method. Indeed the Lomb-Scargle Periodogram analysis for apical, medium and basal kymograph data provides high confidence levels for the presence of periodicity (new **Figure EV2f** and **Appendix Figure S1b**).

Also, the extent to which the automatically-defined peaks (Fig 2) correspond to the visually-defined microclusters (Fig 1) remains to be examined. Interestingly, this analysis may have been performed in Fig 7: 34 microclusters have been identified following **nuclear segmentation and detrended intensity profiles were studied**. How peaks and clusters compare is, however, not presented.

We would like to highlight that peaks in the auto-correlation show that the signal has similarity to itself, and hence periodicity, at the distance lag shown by the x-value of the peak. Therefore, unlike the raw spatial Venus::HES5 intensity profile, these peaks do not directly map to peaks in the spatial Venus::HES5 intensity profile. Thank you for giving us the opportunity to clarify this methodological detail (revised manuscript, page 7).

However, we understand the spirit of the question and therefore, in the revised version we present a direct comparison of a segmented Venus::HES5 nuclear intensity image (new **Figure EV2b,c**) and a ventral-to-dorsal Venus::HES5 spatial profile. This shows that, as expected, peaks in the spatial profile map nicely to the high cells in the microcluster plots.

Next, the authors examined the time evolution of the non-detrended HES5 intensity obtained from low-resolution imaging (Fig 3a-d). The kymograph data are interpreted to suggest that groups of 2-3 cells (defined arbitrarily using 20 micron-wide ROIs) can interconvert between low and high HES5 states every 6-8 hours (Fig 3a-d).

The 20 μ m ROI is not chosen arbitrarily, but it represents half of the 40 μ m spatial periodicity, chosen so that it can capture the size of a microcluster. We have clarified this in the text and methods. However, the location of the 20 μ m region may not correspond to clusters. To account for this, we provide a new method where we performed detection of high/low regions (above or below mean) from the kymograph and take an ROI around this region. The microcluster persistence time is very similar between the two methods (**Figure 3d** and new **Appendix Figure S2d**). Thus, we conclude that the Venus::HES5 spatial pattern is

dynamic across the tissue (additional clustergrams added to **Appendix Figure S2c**) and the method used to capture regions in the kymograph does not affect the persistence time.

Second, the authors extracted the mean phase values of the underlying single-cell oscillators from the detrended HES5 intensity values (Fig 3f-h). The latter analysis was performed at higher spatial resolution but lower temporal resolution. Again, this analysis supports the notion that clusters of cells can interconvert between low and high HES5 states. The authors therefore conclude that the spatial pattern of HES5 is dynamic over time. Two observations need, however, to be clarified. First, it is unclear whether clusters exhibit periodic expression in time. Given the temporal persistence of the high and low states, a 12-16h period may be expected. How do the authors relate this periodicity to the 3-4h period of HES5 expression seen at the single-cell level?

We realise that we did not explain well in the original manuscript how the persistence time of the microclusters (6-8hrs) relates to the single cell periodicity of (3-4hrs). Therefore, we have added a new section to clarify this in the revised manuscript. As we discuss in page 10, the persistence of HES5 in a high or low state microclusters is a new finding that has emerged from the tissue level analysis described in this manuscript and has not been reported before. In the revised manuscript, we suspected that it may correspond to “longer-term” trends in the Venus::HES5 mean level. These have been previously observed at the single cell level, on top of the ultradian oscillations/fluctuations (3-4hrs), but not analysed further. Thus, we investigated the time scale of the longer-term fluctuations of Venus::HES5 expression in single cells to see if it matched the microcluster persistence time. Indeed, using the same methods to analyse persistence we observe similar values in persistence time between microclusters and single cell long-term Venus::HES5 trends. Taken together, with the analysis of cell pairs (above) these results suggest that changes in the levels of Venus::HES5 in microclusters are produced at single cell level through in-phase ultradian activity as well as changes in the HES5 levels.

The second question of whether clusters exhibit periodic expression over time with a 6-8hrs periodicity is more difficult to answer conclusively at this time because most of our movies are 12-14h. In some movies, we do see clusters that go from high expression to low and then high again (highlighted in **Figure 3c** and new **Appendix Figure S2c** mentioned in text page 9), implying periodicity. To address this question with certainty, experimental improvements would be required to increase tissue slice viability and we feel that this is beyond the scope of the current study.

Also, the authors assume that the x,y coordinates of the recorded cells/clusters do not change over the time course of the live imaging experiment. In other words, the living tissue must not locally expand, contract, deform etc... over the 12-24h period of recording: is this really the case? Actually, the kymograph shown in Fig 3b may suggest that **the shape of the tissue**, hence cell positions, change over time, hence generating these non-reproducible traveling waves of HES5 expression. In sum, the authors need to exclude this possibility to convincingly show that the HES5 spatial pattern is dynamic over time.

Indeed, one has to exclude artifacts introduced from tissue deformation. We are aware of this pitfall and we have addressed it in the original manuscript in several ways 1. The timelapse data undergoes a rigorous quality control that requires tissue viability for >12hrs and a maintenance of tissue and ventricle integrity 2. Image registration/drift correction by tracking a distinct landmark, as detailed in Methods. Therefore, *global tissue movement* is accounted for. Furthermore, we account for changes in single cell positions and *local tissue deformations* in 3 ways;

1. In the revised manuscript, we have introduced a new, additional, quality control method by measuring the average motility of tracked single cells over time in the D-V axis and comparing it to patterns/waves of Venus::HES5 intensity in kymograph acquired from the same movie. A threshold of 20 μm for average single cell displacement in 12hrs is used and any movies with greater average single cell displacement are not considered for the analysis of HES5 spatial pattern dynamics. 1 out of the 10 timelapse movies failed the quality control threshold and is highlighted in the new **Appendix Table S2** together with examples that were accepted. The timelapse movie that fails quality control has average single cell motility with a persistent drift over time and a mean displacement of 22.8 μm . The example passing timelapse movie has minimal drift in mean single cell motility and a pattern over time that is not reflected in the patterns/waves in the kymograph. Even with the new quality control measures, we still observe convincing high-to-low changes in Venus::HES5 expression in the kymographs and conclude that the HES5 spatial pattern is indeed dynamic over time.
2. By using a 15 μm wide (in apical-basal axis) line to generate the kymograph. Single nuclei have a mean displacement of 7.9 μm in apical-basal direction in 2.5h therefore we ensure that on average nuclei would not leave the region of interest in the 2h window used for downstream averaging of the kymograph. This information has been included in methods (section).
3. By using a 20 μm wide (in D-V axis) region of interest (ROI) applied to the kymograph in order to measure the persistence time of the Venus::HES5 spatial pattern. Single nuclei have a mean displacement of 5.6 μm in dorso-ventral direction in 2.5 hrs therefore it is highly unlikely that microclusters will leave this ROI.

Next, the authors looked at the role of Notch using drugs. As expected, the expression levels of the Notch target HES5 are down (Fig 4a) and this effect is associated with reduced amplitude of the peaks upon detrending (Fig 4b). However, whether Notch contributes to the dynamic high-to-low and low-to-high switches at the cluster level may be hard to judge when HES5 levels are overall reduced **Thus, the contribution of Notch to the dynamic switching remains to be clarified.**

Our analysis is focused on the influence of DBZ on the phase re-sets observed between subsequent de-trended kymograph Venus::HES5 intensities in time (local fluctuations) rather than the trend or the amplitude. The phase detection method (Hilbert transform) is not dependent on the level of expression and so the reduction in HES5 levels in DBZ does not affect our analysis of the dynamic microcluster high-to-low and low-to-high phase switches. We have clarified this in the revised manuscript (page 11).

Then, the authors switched to computational modelling to test whether varying the strength of cell-cell coupling between oscillators is sufficient to produce spatial and temporal periodicity (Fig 5). Exploration of the parameter space identified conditions that allow for the emergence of in-phase HES5 expression in small clusters of cells (Fig 5g). **This analysis is nice and clear.** Nevertheless, as the authors note, this modelling does not produce the kind of periodic spatial patterns that are reported in Fig 2. The authors further observed that both the amplitude of simulated HES5 variations in time and space depends on the coupling strength (Fig 6c). Assuming that the probability to differentiate increases when HES5 levels are below a threshold, the rate of differentiation is then expected to vary with the coupling strength (Fig 6b). To test whether this idea may apply in vivo, the authors correlated the differentiation rate observed in two domains of the spinal cord with the 'spatial amplitude'

measured in these two domains in snapshot views of HES5. The observed correlation is consistent with the idea that the rate of differentiation is regulated by the coupling strength. Obviously, other regional differences may account for the observed differences in the rate of differentiation.

Yes, we agree, other regional differences may contribute the observed differences in the rate of differentiation. An exciting possibility is that other factors, such as the presence of OLIG2, which represses HES5 in the MN domain (Sagner et al. PLOS Biology 2018) converge to the coupling strength, however, this is not known and it is an interesting question for the future. In the revised manuscript, we have included a more balanced discussion on other possible factors influencing spatial differences in the rate of differentiation to highlight this point. This is found on page 24.

In sum, the ideas explored in this study are very interesting but the data do not support well enough the conclusions.

We thank the reviewer for their detailed reading of the manuscript and feel that the additional analysis has strengthened the support of the main conclusions of the paper.

Reviewer #2:

In the current manuscript, the authors build on the previous finding that Hes5 expression is oscillatory in neuronal progenitors (Manning et al., 2019) and try to address how these oscillations unfold in the tissue level. Examining the spatial expression of Hes5 they uncover so-called microclusters of correlated expression that appear regularly arranged along the D/V axis of the developing neural tube. Moreover, these patterns do not appear static in time, as the levels of Hes5 follow different dynamics. Pharmacological manipulation with Notch signalling inhibitors reveal that communication through Notch is critical for such temporal dynamics of the microclusters while the spatial arrangement appear less affected. The authors employ a mathematical model to formalize assumptions and hypothesis and test the potential origin and impact of microclusters on neuronal differentiation. According to this analysis the coupling strength between cells can partially explain the observed dynamics and the authors show that different spatial patterns correlate with differentiation rates. The authors attempt to address an important and timely question, and towards these they employ an impressive toolbox applying mathematical and live imaging analysis. Their main findings are the existence of microclusters and their role in differentiation. The presence of microclusters is quite clear, although their origin is not investigated deeply enough. How much of the pattern can be attributed to clonal expansion of some cells, and what role does the cell migration play ?

We thank the reviewer for stating that the evidence we provide makes the 'presence of microclusters quite clear'.

With regards to the contribution of clonal expansion we assume that the reviewer means that microclusters could be set up early on with fewer or even single cells, which then expand through cell division. One might then expect that the cluster size would increase over developmental time as more cell division occurs. In the revised manuscript we have looked at earlier stages, however we did not observe any significant changes in either the number of cells per microcluster (in D-V axis) (amended **Figure 1d**), or the spatial periodicity in Venus::HES5 expression (new panels in **Figure 2h**) between E9.5, E10.5, and E11.5. The cell cycle length of neural progenitors in the mouse spinal cord at E10 is around 13hrs (Kicheva et al. 2014), so E9.5-E11.5 should encapsulate multiple cell divisions. We conclude that clonal expansion is not responsible for generating the periodic HES5 microclusters along the D-V axis. Conversely, our new data also shows that cell divisions are

not detrimental to the pattern as cluster size in DV seems to be preserved across different stages.

The reviewer also asks whether what role might cell migration play in formation of the microcluster pattern. We have not found any published evidence of co-ordinated nuclear migration at a scale that could explain the formation of cellular clusters, but in the revised manuscript we looked at correlation between single cell nuclear positions in the tissue. We found a weak positive correlation in the movement of nuclei in apico-basal axis between cell pairs less than 20 μ m apart, but there was a large variation in correlations, and the correlation dropped between cells further apart (new **Appendix Figure S1e**). This weak correlation in apical-basal nuclear movement may contribute somewhat to maintaining the microcluster pattern. We incorporated a sentence in the results on page 8.

While the authors emphasize Notch signalling to explain their formation, it is clear from their data that the spatial pattern cannot be attributed to this. Seeing the expression pattern of *olig2*, and *ngn* in Suppl fig 7 it is clear that some form of spatial pattern is definitely not specific to *Hes5*. The authors skip critical questions on the microclusters, i.e. what other genes show similar pattern? Are they correlating/anticorrelating with *Hes5*? Some of the interesting candidates would be *Delta*. If *Delta* expression doesn't follow the *Hes5* expression as they imply in the discussion, it will be clear that the pattern is not emerging through Notch signalling interaction.

In the revised manuscript, we provide new and exciting experimental evidence showing that the periodic spatial pattern we describe is a composite pattern consisting of single cells that co-ordinate both *HES5* levels and temporal phase over time (revised **Figure 3g-l**). Using previously reported single cell data from Manning et al 2019, we now show that at single cell level the activity of *HES5* between neighbouring cells is indeed predominantly in-phase (new results in **Figure 3l** and **Appendix Figure S4b,d**) which is consistent with the model prediction at weak coupling. This allows us to draw a more robust conclusion that Notch promotes in-phase activity in the microclusters. The previously shown DBZ experiments also inform us that in the absence of Notch signalling microclusters are no longer maintained in a dynamic way. We apologise if this was not clear and we have made this clearer in the discussion page 12.

However, we do agree that the spatial pattern is not specific to *HES5*, an exciting observation that shows the wider applicability of our findings to neurogenesis overall. In the revised manuscript, we have investigated the spatial expression of the pro-neural factor *NGN2*. We have added a new figure (**Figure 8**) which shows, using both *NGN2* antibody staining and a *NGN2:mScarlet* fusion reporter mouse, that *NGN2* also has a spatially periodic expression pattern, with around half the spatial period of *Venus::HES5*. We show that the *NGN2* and *Venus::HES5* patterns correlate through the cross-correlation function and that the lag between the *NGN2* and *Venus::HES5* pattern is less than a single-cell width. This indicates that the two spatial oscillators are super-imposed on each other with no phase shift. The spatial period of *NGN2* is also smaller in the motoneuron domain with a mean period of 21 μ m supporting the conclusion that a hi-lo-hi-lo pattern exists in the motoneuron domain. We also use the pseudo-colour representations of nuclear *Venus::HES5* and *NGN2* levels to show that *HES5* microclusters only contain 1 or 2 *NGN2* high cells. We conclude that at least *NGN2* in the Notch-Delta regulatory network shows a spatial periodic pattern of half the period to *Venus::HES5* resulting in *NGN2* high cells occurring in *HES5* high microclusters and in *HES5* low microclusters. We hypothesise that high *NGN2* is expressed in a sustained manner and hence maintains expression even when the microcluster has switched to a low state. Therefore we propose that microclusters act to reliably select one or two cells that go on to express sustained *NGN2* and differentiate.

In the revised manuscript we have also addressed the spatial pattern of Dll1 using single-molecule fluorescent in-situ hybridisation (smiFISH). Dll1 and Jag1 mRNA large-scale expression patterns from chromogenic in-situ hybridisation have been previously published in Marklund et al. Development 2010. Dll1 shows a broad expression domain in the ventral spinal cord that covers the motor neuron domain and the ventral-most part of the interneuron domain. Alternate stripes of Jag1 and Dll1 are observed in the intermediate spinal cord, which covers the remaining part of the interneuron domain. Thus, there is domain pattern of Dll, Jag1 (and Hes5) which occurs on a larger spatial scale than the microclusters (**diagram in Figure EV5**).

In the revised manuscript we have looked inside one of these large spatial domains and we show that in the region where HES5 is expressed in microclusters, Dll1 mRNA is expressed in stripes that are a couple of cells wide (new **Figure EV5**). Precise correlation of the spatial patterns of Dll1 and HES5 (i.e. whether in phase or anti-phase) is challenging with smiFISH and presently, we do not have the endogenous protein fusions reporter mice needed to address this question in the same robustly quantitative way that we have done with NGN2 and HES5. Thus, while we recognise the importance of this question, it is outside the scope of this study.

The role that the micropatterns play is even more blurry and safe conclusions are difficult to draw. Important questions on the fate of one microcluster's cells remain unanswered. The implication from the authors' model is that the cells of the microcluster, having coherent Hes5 levels, would differentiate in unison. It would be important to examine if this is the case

We have investigated whether multiple cells within a cluster differentiate in unison in the revised manuscript. Our newly added HES5/NGN2 data shows that within a HES5 microcluster, fewer cells (and in the MN domain only one cell per cluster) upregulate NGN2, suggesting that cells in a cluster do not differentiate in unison (new **Figure 8h**). Secondly, we have independently investigated this question in our stochastic multi-cellular HES5 model. Consistent with the experimental observations, we find that in the majority of clusters with 3 or 4 cells wide in D-V axis, a single cell from that cluster is likely to differentiate. The distribution of the number of cells differentiating per cluster is shown in **Appendix Figure S7b,c**. With these new findings, we conclude that the most likely role of the microclusters is to facilitate or enable one cell per cluster to differentiate (see discussion, page 19).

The mathematical model is also sophisticated and is built on reasonable assumptions but the comparison to experimental results is problematic. In particular, the authors see that different patterns correlate with different differentiation rates. They are then too fast to assign a causal link through the model, without any experimental validation. According to their hypothesis the pattern depends on the coupling strength, the ultimate cause of the pattern and the variable differentiation rate. However, the coupling strength is not probed. **Since the coupling is expected to be mediated by notch signalling, one would expect that its manipulation can change the pattern and differentiation rate. Such experiments are critical in order to substantiate the links drawn through the model's hypotheses.**

In our model, the coupling strength is described by a repression threshold and Hill coefficient. The repression threshold represents the amount of HES5 protein required to reduce Hes5 transcription in a neighbouring cell by half. Therefore, a low repression threshold results in a large coupling strength as not much HES5 is required to affect the neighbouring cell. The Hill coefficient indicates how steep the response curve of Hes5 transcription is to a change in HES5 protein in the neighbouring cell, with higher values corresponding to increased nonlinearity. While conceptually simple, in molecular terms, the coupling strength encompasses multiple molecular events and factors (i.e. such as co-

repressors) that are presently unknown. Thus, it is not clear how to experimentally manipulate the coupling strength in the context of neurogenesis.

Nevertheless, we recognise that testing the influence of the coupling strength on the microcluster pattern is important and we had to think of alternative ways to strengthen our hypothesis. We reasoned that if our model is correct, then domains where the coupling strength changes should have a different microcluster pattern. We have already shown in the original version of the manuscript that microcluster size and spatial periodicity differs in the motoneuron from the interneurons domains, so we hypothesised that these domains may have different coupling strengths.

In the revised manuscript, we have now provided more evidence that the coupling strength is changing between the motoneuron and interneuron domains by further analysis of quantitative and spatial FCS data presented in Manning et al. 2019. Consistent with the model prediction, the motoneuron domain has a lower nuclear Venus::HES5 concentration, to be expected from a region with higher coupling strength, as at higher coupling strength lower number of HES5 molecules are needed to affect Hes5 transcription in the neighbouring cell. The model also predicts that weak coupling, as predicted in the interneuron domain would generate smaller cell-cell HES5 concentration differences compared to strong coupling (new **Appendix Figure S7a**). This is because weakly coupled cells have less ability to repress the transcription of their neighbours and so are more similar in levels. This relationship persists even after correcting for mean level in each condition. The correction by mean is important as variability in expression is expected to scale with the mean. Indeed, when we calculate the difference in Venus::HES5 concentration relative to the mean between neighbouring cell-pairs from FCS-calibrated single cell expression maps of spinal cord, it is lower in the interneuron domain compared to the motoneuron domain (revised **Figure 7f**). This is further experimental evidence of changes in coupling strength between IN and MN. A discussion of this has been incorporated in page 29.

Overall, I find the attempt very interesting but not very convincing. The authors would benefit from deeper examination of the origin and role of the microclusters and I would recommend publication after serious revisions. Other specific points that need to be addressed are following:

1. The authors remove the signal trend by fitting a polynomial. What is the polynomial's grade?

We used a polynomial order 4 to 6 for all of the analysis and we have explicitly stated this in the Appendix Supplemental Methods and legends where appropriate.

2. In figure 2d an interesting application of the LS periodogram is exploited to detect the spacing of the microclusters. It may be possible to use this further and examine which, if any, of the significant peaks are detected in multiple samples. This will give further credibility on the uncovered periodicity.

Thank you for the suggestion, we have added performed Lomb Scargle periodogram analysis of the spatial data and we found it agrees well with our initial findings. An example of average spectra of multiple kymographs from one experiment is included in the new **Appendix Figure S1b**. One can observe that when averaged across multiple observations, the spectral analysis shows a dominant peak at frequencies around 40um and the peak is well above a stringent statistical significance test.

3. In figure 2 g-i there is analysis of the intermediate and basal domains along the D/V axis, in addition to the apical. It may be worth performing LS periodograms for these signals and compare the detected peaks.

We have now performed LS periodogram on the intermediate and basal domains (new **Figure EV 2f**) showing the spatial periodicities are similar in the apical and intermediate

domains and appear reduced in the basal domain. These new findings agree with our auto-correlation analysis and we have additionally amended the graph in **Figure 2h** to include additional technical repeats per experiment.

4. For figure 3, the analysis is done in blocks of 20 μ m in width. While this is in principle fine, it is not clear why this particular segmentation is the one to perform. What would happen if the windows are shifted by 2 or 3 μ m? I think a better approach would be to apply a gaussian blur filter on the kymograph.

We have revised the clustering of dynamics and persistence time analysis in **Figure 3 c-e** to incorporate a 2 μ m Gaussian blur pre-processing step of kymograph data and this is described in Appendix Supplementary Methods. As expected the Gaussian blur produces a smoother version of the kymograph data (new **Appendix Figure S2b** vs **Figure S2b**). These changes did not produce any notable differences from our initial results (revised **Figure 3 d,e**).

5. In figure 3c there are 2 big groupings of the temporal behaviour, as increasing or decreasing. I am wondering whether this is an oversimplification, as buried in these patterns are oscillatory trends. In fact, such dynamics are commented on, in the main text ("...revealed changes from low to high Venus::HES5, high to low, or fluctuations between the two states...") but are not highlighted in the figure.

We have highlighted fluctuating areas in **Figure 3c** and provided clustering examples of independent experiments in **Appendix Figure S2c**. All clustergrams showed fluctuating clusters in addition to High-Low and Low-High. We added a sentence to describe this in the main at page 8.

6. I would recommend a rainbow LUT (HSV in MATLAB) instead of the one used from blue to yellow. It would have the advantage of continuity between 0 and 2 π .

We have changed the LUT in **Fig 4c** and **Fig EV 3b** as suggested.

7. The term "salt and pepper" is not the most informative for the described case in the mathematical model. The term "checkerboard-like" that the authors first use in the discussion is much more accurate and should be used throughout the manuscript. We understand that salt-and-pepper implies random and have removed it. We like checkerboard for the model description, however it implies periodicity along apico-basal axis. Our analysis shows that the D-V spatial period changes slightly between apical and basal regions suggesting a checkerboard pattern is not entirely accurate to describe the data. Therefore we have changed "salt and pepper" to "alternating high-low" throughout the paper.

8. When explaining the Kuramoto order parameter in the materials and methods section (p.28), the authors state that "A value of 0 indicates no synchronization, and in complex space would appear as a random distribution of phases that average to a point at the origin". This is not correct and I would remove the word "random", because a value of 0 can emerge from properly arranged phases, i.e. dividing the 2 π cycle in equal n intervals and placing n oscillators' phases at the interval boundaries. Importantly, the oscillators are synchronized out-of-phase and thus a value 0 indicates no in-phase synchronization.

The reviewer is correct and we thank you for highlighting this error/inaccuracy. We have clarified this in the main text and (page 12) removed the word 'random'.

9. In Suppl fig.1, panel g is not accounted for in the legend or the main text.

Thank you for highlighting this error, we have addressed this and added to the legend and main text.

Reviewer #3:

It has been shown that HES5 expression oscillates in neural progenitors, but its multiscale

pattern is not known in the developing nervous system. The authors sought to determine the HES5 expression patterns in the developing spinal cord by live imaging and found the existence of spatially periodic and temporally dynamic microclusters of HES5 expression. Using the computational modeling and experimental data, the authors showed that Notch signaling-mediated local synchronization is responsible for formation of such microclusters, and that the HES5 spatial patterns correlate with the rate of cell differentiation.

By taking advantage of a powerful live imaging technique, the authors for the first time demonstrated the local synchronization of HES5 expression in the developing nervous system. This is an interesting and elegant work showing the significance of the HES5 spatial patterns in cell differentiation. The results are of general interest but need some clarification. Specific comments are indicated below.

1. In Figure 1, the authors should show whether or not **HES5 microclusters continue like stripes along the z axis (anterior-posterior axis)**. If this is the case, it is interesting to image HES5 expression from the apical side by opening the neural tube.

To address this interesting question, we have taken longitudinal cryo-sections of the spinal cord and performed auto-correlations of the Venus::HES5 spatial profile along the A-P axis. Peaks in the auto-correlation show spatial periodicity in A-P axis of around 30um (new result in **Figure EV2i**). We also interrogated how far microclusters may extend in the anterior-posterior (A-P) axis using our existing kymograph data. To do this, we have correlated kymographs from non-overlapping subsequent z-stacks (extending in the A-P direction) and arranged the correlation coefficients by distance in AP. We find that kymographs are highly correlated and correlations drop off with increasing distance (new **Figure EV2j**). Combining these two approaches we conclude that the scale of cluster size in the AP is comparable to that observed in DV. We have added this in the manuscript on page 8.

2. The authors showed that Notch inhibition disrupts HES5 microclusters (Figure 4), but it is not clear how Notch signaling regulates different HES5 spatial patterns. The authors suggested that different coupling strength leads to different patterns like microclusters or salt-and-pepper conditions (Figure 5g), **but this could be explained by different levels or types of Notch ligands. The authors should examine this possibility.**

We agree that different levels or types of Notch ligands *could* lead to different patterns and perhaps also change the coupling strength. In the revised manuscript we have addressed the spatial pattern of Dll1 using single-molecule fluorescent in-situ hybridisation (smiFISH) (new result in **Figure EV5**). Notch ligands are expressed in different domains along the D-V axis of the spinal cord as reported in Marklund et al. 2010. They show a broad expression domain of Delta in the ventral spinal cord that covers the motor neuron domain and the ventral-most part of the interneuron domain. Alternate stripes of Jag1 and Delta are observed in the intermediate spinal cord, which covers the remaining part of the interneuron domain. In the revised manuscript we show that in the region where HES5 is expressed in microclusters, Dll1 mRNA is expressed in stripes that are only 1-2 cells wide. We think these are microstripes within the broader expression domain.

In the updated manuscript we have provided more evidence that the coupling strength is changing between the motorneuron and interneuron domains by analysing quantitative and spatial FCS data presented in Manning et al. 2019. Consistent with the model prediction, the motorneuron domain has a lower nuclear Venus::HES5 concentration, to be expected from a region with higher coupling strength. Using the model, we predict that weak coupling would generate smaller cell-cell concentration differences compared to strong coupling. This tendency persists even after correcting for mean level in each condition. The correction by mean is important as variability in expression is expected to scale with the mean. Indeed we

can now show that the difference in Venus::HES5 concentration relative to the mean between neighbouring cell-pairs is lower in the motorneuron domain compared to the interneuron domain (new result in **Figure 7f**). This is further experimental evidence of changes in coupling strength between IN and MN.

3. In the mathematical model in Figure 5, cells are steadily present in the same place. However, in the apical region, cells only in late G2-M-early G1 are present. Thus, it is likely that many cells come and go due to **interkinetic nuclear migration** every hour or so in each microcluster. The authors should clarify whether this mathematical model represents the in vivo condition of the developing spinal cord.

The model is representative of the apical domain and assumes a fixed number of cells that maintain their neighbour contacts over time but does not include interkinetic nuclear migration. As such, it is not a biophysically accurate description of the in vivo conditions but rather provides a conceptual understanding of dynamic regimes arising in complex multicellular systems. We have clarified these assumptions in the updated manuscript, page 32.

4. In Figure 5, depending on the inter-cellular time delay, $2\pi/3$ phase difference of HES5 oscillations could be stable. Did the authors observe such phase difference between cells, particularly when HES5 microclusters are absent?

We have not specifically looked at the phase shift distributions at different parameter points of the model. Therefore we do not know the answer to the question and in the interest of prioritising questions that are directly relevant to our findings, we regret that we have not addressed this.

Thank you for sending us your revised manuscript. We have now heard back from the two reviewers who were asked to evaluate your study. Overall, the reviewers think that the study has improved as a result of the performed revisions.

However, as you will see below, reviewer #1 still raises some remaining concerns. During our pre-decision cross-commenting process (in which the reviewers are given the chance to make additional comments, including on each other's reports), reviewer #2 agreed that there are several questions that remain open but mentioned that nevertheless in their opinion the findings merit publication in *Molecular Systems Biology*. Reviewer #2 recommended that you are given the chance to address the remaining issues. As such, we would like to offer you an exceptional second round of revision. I include the additional comments of reviewer #2 below the reviewers' reports, as they can be helpful for addressing the remaining concerns.

On a more editorial level, we would like to ask you to address the following issues.

Reviewer #1:

The authors made several changes in response to the various points raised in the first reviewing round. As a result, this revised version is somewhat improved, mostly through clarifications. I review below the new data with a specific focus on the authors response to my comments.

1. Fig 1. The authors did not really address the issue of the lack of automatic detection method of the micro-clusters. The authors indicate that they 'used clearly-defined and objective criteria to identify and count the number of cells per cluster' and referred to the Appendix Supplementary Methods (ASM). Unfortunately, I failed to read these criteria in this section of the paper and only found a section on diameter calculation of the synthetic micro-clusters (ASM, p.21), and the criteria given in the Methods section (p.28, l.926-9) are not sufficiently clear to understand whether and how micro-clusters were identified in an unbiased manner. Thus, the initial definition of the local clusters of cells with similarly high levels of HES5 may still be seen as somewhat arbitrary (Fig 1a-c). It is therefore still not clear how their size was determined (Fig 1d). In this regard, the authors should measure the correlation coefficient with neighbors located farther away (# of neighbors>4). Indeed, since micro-clusters are stated to include 3-7 cells, one would expect this correlation coefficient to decrease beyond the sixth neighbors. Is this observed? This analysis would also probably require to increase the n number in the analysis (only 5 micro-clusters appeared to be studied in Fig 1e). If so, would it be possible to apply this criterium to automatically identify micro-clusters? Otherwise, the authors did perform local detrending on the HES5 data shown in Fig 1. However, it seems that two different detrending methods were applied to the data shown in Figs 1 and 2. Is there a particular reason for this? I also did not understand why an analysis of the spatial periodicity of the HES5 signal was not performed on the 40x segmented images (Fig 1b). Last, while I understand that intensity peaks do not necessarily match with high auto-correlation values, it might still be helpful for the reader to see whether periodicity/auto-correlation peaks match the positions of the HES5 micro-clusters identified in Fig 1 as 'local clusters of cells with similar levels of HES5' (as now plotted in EV2b,c).

2. Fig 3i-l (new data based on cell pair tracking). I presume that the pairs of cells selected for tracking were those that exhibited a strong HES5 signal allowing for easier tracking. If so, one may wonder whether these selected pairs are enriched in sister cells exhibiting similarly high levels of HES5, hence implying that history rather than position may contribute to similarity in dynamics. Can the authors exclude this possibility? If not, the conclusion that micro-clusters include 'local in-phase' cells with correlated levels of HES5 would be weakened. Also, if nearly all (13/14) cell pairs exhibit local co-ordination in phase, what would be the expected behavior at the tissue scale? Is the spatial phase map shown in Fig 2k consistent with the notion of local co-ordination in phase? Major point, panels i-j: to what extent the correlation measured for the raw Venus-HES5 values merely reflects the global decrease in signal intensity seen over time? Is correlation maintained for detrended values?

Minor point, l.329: the only pair of cells shown in Fig 3g does not appear to maintain close proximity (<20microns) over the time course of the movie as stated in the text. Indeed, at the start, the distance is clearly >20microns.

3. Fig 2c and 3f. How do the authors explain the variability in spatial periodicity?

4. I still find difficult to reconcile the observations that micro-clusters (defined by local high HES5 levels) persist for 6-8 hr and that these micro-clusters contain 'in-phase' cells showing fluctuations of HES5 with a 3-4 hr period.

5. local tissue deformation. The quality control and drift correction steps discussed by the authors in their rebuttal do not really address the local tissue deformation issue raised in my review. This issue is however now tentatively addressed by measuring the average mobility of tracked single cells. This approach raises two questions. First, the authors state that nuclei have a mean displacement of 7.9/5.6 microns in the a-b/d-v directions but do not detail how these values were obtained: how many cells were tracked? what is the observed movie-to-movie variability? A second more relevant point: these displacement values support the possibility that some of the high-to-low switches (see for instance the central box ** in Fig 3b) may in part result from cells with high HES5 levels leaving the binned area. Thus, it seems possible that local deformations may indeed complicate the interpretation of the data.

6. Contribution of Notch to the dynamic switching. I understand that the phase detection method is not dependent on the level of expression, but I believe that this statement is correct in the absence of significant noise, which may not be the case when HES5 levels are low.

7. The new findings on the spatial periodicity of NGN2 (Fig 8) do not really shed new light on any of the issues raised by the referees but rather bring about a new layer of complexity as it is interpreted to suggest the existence of two spatial oscillators super-imposed on each other. I would recommend to leave this set of new observations for a future study.

Reviewer #2:

The authors provided an updated manuscript, where significant effort has been put to improve it. They have addressed the majority of point raised. More specifically:

- The measured fixed size of the microcluster size at different stages of development indicates that they are unlikely to be just a product of clonal expansion

- Additional data on Ngn2 and Dll1/Jag1 expression enrich the knowledge about microclusters.

Although their expression patterns are not always easy to reconcile, they are interesting findings for future investigation. For example it could be examined why only 1 cell of the cluster expresses Ngn2 and whether the position of this cell is stereotypical in a cluster...

- The clarifications about the meaning of coupling strength are useful, but highlight the significant lack of knowledge that the Hill functions encapsulates in a black box.

- All the minor points I raised have been adequately addressed.

From the additions, I found particularly illuminating and important the lines 246-263, while the added section in lines 310-350 is more difficult to follow. The authors assign a "longer-term" trend in the Venus::Hes5 level, which in fact implies the existence of a second periodicity (longer period thus "longer-term" trend) on top of the described 3-4hrs. It may have been helpful to build this argument with a cleaner description of what this "long-term" trend actually is.

Overall, the revised manuscript addresses the great majority of deficiencies, although at times it leaves more open questions than satisfactory explanations. Nevertheless, the findings presented are substantiated and I believe merit publication in Molecular Systems Biology. I expect this line of research will be followed.

Additional comments by reviewer #2 after cross-commenting:

I see the value of many of the points referee 1 raises in the renewed revision. In particular, the idea that the manuscript is "overall improved, mostly through clarifications" somewhat echoes my comment that it leaves more open questions than satisfactory explanations. I would like to comment on some aspects of his/her critique:

1. He/She was not convinced of the existence of microclusters. From my point of view this was not a major issue at the beginning, and while I agree that some aspects could be done better (I agree with the importance of measuring the correlation coefficient beyond the 4th neighbour, expectation being it will drop) there are supporting evidence for the microcluster's presence and relevance. More specifically, the correlation with NGN2 expression patterns adds on their credibility. Thus, I would think including these data is important against point 7 suggestion.

2. I found the addition of the single cell tracking data a significant improvement. In fact, such experiments are quite demanding from a technical point of view, and thus I anticipate that only a selection of pairs are admissible for analysis. As long as the criteria for selecting the pairs are clearly stated, I wouldn't see any further problems.

4. This is the point with which I agree most with reviewer 1, as it resonates with my critique on the section 310-350. I believe the argument of the authors boils down to the existence of a second periodicity at 6-8 hrs (interestingly a harmonic) that is more evident due to biological (or technical?) reasons at the population level. I think the authors should be given the opportunity to further clarify/explain this.

Overall, I would recommend giving the authors another chance to address these issues. While the picture emerging from the work is somewhat blurry, I think the results would be of value to the community and they suggest new avenues of research.

Thank you for giving us the chance to respond to the remaining reviewer's concerns. As a result, we have made several changes to the figures and manuscript text which we outline in our point-by-point response below.

The authors made several changes in response to the various points raised in the first reviewing round. As a result, this revised version is somewhat improved, mostly through clarifications. I review below the new data with a specific focus on the authors response to my comments.

1. Fig 1. The authors did not really address the issue of the lack of automatic detection method of the micro-clusters. The authors indicate that they 'used clearly-defined and objective criteria to identify and count the number of cells per cluster' and referred to the Appendix Supplementary Methods (ASM). Unfortunately, I failed to read these criteria in this section of the paper and only found a section on diameter calculation of the synthetic micro-clusters (ASM, p.21), and the criteria given in the Methods section (p.28, l.926-9) are not sufficiently clear to understand whether and how micro-clusters were identified in an unbiased manner. Thus, the initial definition of the local clusters of cells with similarly high levels of HES5 may still be seen as somewhat arbitrary (Fig 1a-c). It is therefore still not clear how their size was determined (Fig 1d).

We have now introduced automation to the existing method for segmenting a microcluster based on intensity thresholding. Using automated counting, we now report a total number of cells in the microcluster (Figure EV1d), which is very similar to those obtained from manual counting and does not change the message of the paper. We have also used the automation to measure the distance between microclusters and find it to be approximately 3-4 cells (Figure EV2f), validating our periodicity measure from auto-correlation in Figure 2.

To increase confidence in our method we have performed two control experiments: In Randomisation 1 (Rnd1) we randomly shuffled the existing intensities assigned to each nucleus and in Randomisation 2 (Rnd2) we randomly sampled from a distribution of intensities with the same mean and standard deviation as the data. Using the automated method, we find that microclusters in both sets of randomised data are predominantly doublets and are significantly smaller than microclusters of HES5 in the real data, showing that the microclusters detected in the data are not expected by chance (Figure EV1d).

We apologise that the criteria for cluster identification and counting was difficult to find and insufficient. In the amended manuscript, we have provided a step-by-step description of automated microcluster quantification in a new Materials and Methods section entitled "Microcluster quantification" (page 29; lines 951-996), signposted on page 6, line 170 of the updated manuscript. We have also made the Matlab code available on GitHub as outlined in Data Availability.

In this regard, the authors should measure the correlation coefficient with neighbors located farther away (# of neighbors>4). Indeed, since micro-clusters are stated to include 3-7 cells, one would expect this correlation coefficient to decrease beyond the sixth neighbors. Is this observed? This analysis would also probably require to increase the n number in the analysis (only 5 micro-clusters appeared to be studied in Fig 1e). If so, would it be possible to apply this criterium to automatically identify micro-clusters?

As suggested by both reviewers 1 and 2 we have included measurements of the correlation coefficient of Venus::HES5 intensity extended to neighbour numbers 5-8 and indeed the coefficient decreases (updated Figure 1e). The correlation in HES5 between a cell and it's 5th neighbour is significantly lower than between the same cell and it's 1st neighbour. Regarding the n number for the analysis, each datapoint in Figure 1e is a correlation coefficient calculated from all segmented cells in one image of embryonic spinal cord (>100 cells per image) and therefore encompasses multiple microclusters. These 5 datapoints represent 5 different tissue replicates and a sufficient n.

Overall, the authors feel that **1.** the automated quantification of microclusters **2.** the addition of the randomised control for the microcluster quantification method **3.** the presence of positive correlations between nuclei that decreases over increasing distance, even after detrending and **4.** the presence of spatial periodicity in Venus::HES5 intensity with a period of greater than 1 cell gives a high degree of confidence in our approach to identify and characterise microclusters of cells with similar HES5 levels.

Otherwise, the authors did perform local detrending on the HES5 data shown in Fig 1. However, it seems that two different detrending methods were applied to the data shown in Figs 1 and 2. Is there a particular reason for this?

This is because the data in Figure 1 is 2-dimensional and includes two gradients of Venus::HES5 that need to be removed, one in the apical-basal direction and one in the dorsal-ventral direction. The data in Figure 2 is 1-dimensional, along the dorsal-ventral direction, so there is no influence of the apico-basal gradient. Therefore, it only needs de-trending in the D-V axis. A section regarding detrending methods and their motivation has been added to the Materials and Methods (page 38; lines 1237-1250).

I also did not understand why an analysis of the spatial periodicity of the HES5 signal was not performed on the 40x segmented images (Fig 1b).

The 40x segmented images are non-continuous due to the application of the nuclear mask during image segmentation and therefore cannot be used for auto-correlation. However, we have performed an auto-correlation analysis of HES5 signal from non-segmented 40x images. This is included in the rebuttal for the benefit of the reviewer (Rebuttal Figure 1). It shows that spatial periodicity of Venus::HES5 along the dorso-ventral axis in 40x snapshot images and 20x movies is very similar indicating that microcluster periodicity detection is not affected by the imaging set-up.

Rebuttal Figure 1. Spatial periodicity of Venus::HES5 from 20x and 40x images.

Peak-to-peak distance in autocorrelation of Venus::HES5 spatial intensity profile from 40x non-segmented snapshot images or 20x movie frames. Each datapoint represents a single image. NS indicates no significance in Mann-Whitney test.

Last, while I understand that intensity peaks do not necessarily match with high auto-correlation values, it might still be helpful for the reader to see whether periodicity/auto-correlation peaks match the positions of the HES5 micro-clusters identified in Fig 1 as 'local clusters of cells with similar levels of HES5' (as now plotted in EV2b,c).

As the reviewer states, the side-by-side graphs of EV2b and EV2c allow the reader to see how the autocorrelation peaks match the positions of the microclusters. We have adjusted the figure legend to highlight that EV2b is from a segmented image and EV2c is from a non-segmented image (page 54; lines 1818-1819).

2. Fig 3i-l (new data based on cell pair tracking). I presume that the pairs of cells selected for tracking were those that exhibited a strong HES5 signal allowing for easier tracking. If so, one may wonder whether these selected pairs are enriched in sister cells exhibiting similarly high levels of HES5, hence implying that history rather than position may contribute to similarity in dynamics. Can the authors exclude this possibility? If not, the conclusion that micro-clusters include 'local in-phase' cells with correlated levels of HES5 would be weakened.

Cells were not selected for tracking based on the strength of the HES5 signal. Cells were tracked on the H2B::mCherry signal, generated through a Sox1:CreERT2 conditional transgene, to avoid biasing for cells with a strong Venus::HES5 signal. We have clarified this in the Materials and Methods (page 27; lines 875-886). The length of our movies does not allow us to evaluate whether division history may contribute to similarity in dynamics between cell pairs. We have added a sentence in the discussion to this effect (page 21; lines 689-690). However, the conclusion of correlated levels in cell pairs still holds.

Also, if nearly all (13/14) cell pairs exhibit local co-ordination in phase, what would be the expected behavior at the tissue scale? Is the spatial phase map shown in Fig 2k consistent with the notion of local co-ordination in phase?

We interpret this question to be referring to the phase-phase mapping found in Figure 3. Indeed, this showed that cell pairs found in close proximity frequently displayed in-phase expression of Venus::HES5 when activity at all timepoints is being considered. However, the instantaneous phase differs for every pair at any one time in the same tissue. The behaviour at the tissue scale has been inferred by the Kuramoto Order

Parameter analysis shown in Appendix Figure S4h,i which looks at many cells in the tissue. Our data is consistent with the notion that local in phase expression does not translate to global synchrony.

We have additionally performed a control analysis using random cross-pairing of cells in the same experiment (resulting in an analysis of cell pairs located further than 20um away) and in this case the phase mapping shows no predominance of in phase activity reproducibly across 3 experiments (new Appendix Figure S4f,g). This further supports the conclusion that phase synchrony is observed locally and not at the population level. We discussed this in the text (page 11; lines 353-359) and added a description in Materials and Methods (page 35; lines 1156-1165).

Major point, panels i-j: to what extent the correlation measured for the raw Venus-HES5 values merely reflects the global decrease in signal intensity seen over time? Is correlation maintained for detrended values?

The correlation coefficient analysis of detrended temporal Venus::HES5 expression shows that the positive correlation between cells in close proximity persists but is lower than in non-detrended data. It remains statistically significant compared to a control H2B::mCherry signal from the same cells (Rebuttal Figure 2 - right panel). This further supports our conclusion that microclusters are a composite pattern consisting of coordination in levels of Venus::HES5 as a result of long-term slow fluctuations (positive correlations in non-detrended signal) and co-ordination in ultradian oscillations (positive correlations in de-trended signal and in-phase behaviour).

Rebuttal Figure 2. Correlation coefficient of Venus::HES5 and H2B::mCherry control signal in cell pairs in close proximity.

Correlation coefficient analysis of non-detrended Venus::HES5 (left panel) and detrended Venus::HES5 (right panel) in cell pairs. Each datapoint represents median correlation coefficient for a single experiment paired to the control H2B::mCherry signal collected at the same time . ** indicates $p < 0.01$ in two-tailed paired t-test. Data in the left panel included in amended Figure 3g.

However we would caution against interpreting the detrended oscillatory data through correlation coefficients alone as they do not provide a fair picture. Correlation coefficients are known to fail in detecting complex nonlinear relationships such as in phase activity. This is why our preferred methods for analysing the detrended signals are phase-phase mapping and cross-correlation (Figure 3 and Appendix Figure S4) and we have not added the correlation analysis of detrended timeseries to the manuscript.

Minor point, I.329: the only pair of cells shown in Fig 3g does not appear to maintain close proximity (<20microns) over the time course of the movie as stated in the text. Indeed, at the start, the distance is clearly >20microns.

We defined cell pairs as having a median 3D Euclidean distance <20 μ m over 12 hrs. Therefore, at some points the distance may be greater than 20 μ m, but over the course of the movie they are on average closer than 20 μ m. We have clarified this in the text (page 11; lines 337-338) and Materials and Methods (page 35; lines 1155-1156).

3. Fig 2c and 3f. How do the authors explain the variability in spatial periodicity?

We explain the variability in spatial period through biological noise and the stochastic dynamic behaviour of gene regulatory networks. Our previous analysis of single cell Venus::HES5 expression timeseries indicates that a stochastic model of auto-negative feedback is 160 times more likely to describe HES5 expression statistics than a deterministic model (Manning et al. 2019), indicating the presence of stochasticity in single cell expression. The spatial periodicity is generated through coupling of these stochastic processes and so we postulate that the variability is due to stochasticity of the underlying gene network.

4. I still find difficult to reconcile the observations that micro-clusters (defined by local high HES5 levels) persist for 6-8 hr and that these micro-clusters contain 'in-phase' cells showing fluctuations of HES5 with a 3-4 hr period.

Thank you for highlighting this issue, it is clearly an important but confusing point as it was also picked up by Reviewer 2. We have now clarified in the text that single cell Venus::HES5 expression is a composite of 1. low amplitude, short periodicity, noisy, oscillations and 2. larger amplitude and slower varying fluctuations in mean expression. Both were observed in Manning et al., 2019 but their relationship is analysed for the first time in the present paper. We show here that cell pairs have both in-phase activity of the lower amplitude oscillations and correlations in mean levels, suggesting both oscillations and slower varying fluctuations contribute to formation of microclusters. The persistence time of the microclusters matches the timescale of the slower varying fluctuations in mean expression of single-cell Venus::HES5 (Manning et al., 2019).

We have re-written and clarified the text (pages 10-11; lines 318-365) to provide a clearer description of the slower varying fluctuations in mean levels of single cell Venus::HES5 and how these relate to the persistence time of the microclusters.

Further we have measured amplitudes of ultradian oscillations and the slower varying fluctuations in mean Venus::HES5 levels to add to the explanation (new Appendix Figure S4a). We have also added a new diagrammatic figure (new Figure 3k) to provide a visual explanation.

5. local tissue deformation. The quality control and drift correction steps discussed by the authors in their rebuttal do not really address the local tissue deformation issue raised in my review. This issue is however now tentatively addressed by measuring the average mobility of tracked single cells. This approach raises two questions. First, the authors state that nuclei have a mean displacement of 7.9/5.6 microns in the a-b/d-v directions but do not detail how these values were obtained: how many cells were tracked? what is the observed movie-to-movie variability?

As the reviewer states, the average motility of tracked single cells partly addresses the issue of local tissue deformation. We find it unlikely that local tissue deformation would be of large scale or a significant problem because it would be picked up by either the motility of single cells, the deformation quality controls that we imposed or in local z-movement of the tissue. As the reviewer asks, we have added the methodology, cell numbers and variability for the mean nuclear displacement in the Materials and Methods section entitled "Generation of expression profiles and kymographs" (page 32; lines 1046-1070) and show the experiment-by-experiment variation in new Appendix Table S3.

A second more relevant point: these displacement values support the possibility that some of the high-to-low switches (see for instance the central box ** in Fig 3b) may in part result from cells with high HES5 levels leaving the binned area. Thus, it seems possible that local deformations may indeed complicate the interpretation of the data.

As we said in the previous round of rebuttal we have quantified the nuclear migration and it is smaller than the observational window over 2-2.5 hrs. Nevertheless, we cannot exclude the possibility that some nuclei leave the ROI and have included a sentence in the Discussion to this point (page 22; lines 722-724). However, we do not see how this behaviour could generate any of the reproducible phenomena we report in the paper, particularly considering the reproducible difference observed with the DBZ Notch inhibitor conditions. To reiterate, microclusters are less dynamic in DBZ even though nuclei have increased motility in the basal direction in these exact same DBZ Notch inhibitor movies (Manning et al. 2019).

6. Contribution of Notch to the dynamic switching. I understand that the phase detection method is not dependent on the level of expression, but I believe that this statement is correct in the absence of significant noise, which may not be the case when HES5 levels are low.

To address this point, we have utilised the existing analysis that determines statistical significance of spatial periodicity using auto-correlation (height of significant auto-correlation peaks is above noise levels). When the spatial period is significant, we assume there is no substantial noise. We have now used this statistical criterion to

restrict the phase analysis only to the DBZ time intervals that show significant spatial periodicity. This resulted in phase analysis up to 8h-10h in DBZ and we compared it to DMSO over the same time period (up to 10h). The exclusion due to significant spatial periodicity did not remove any DMSO data, however previously we had considered the DMSO data in its entirety whereas now we apply a similar 10h cut-off. These modifications (amended Figure 4e) made the result clearer because they increased the significance in the difference in dynamic switching between DMSO and DBZ Notch inhibitor conditions. Thus, this new analysis strengthened our conclusion that the spatial pattern in DBZ is not dynamic over time even at timepoints before it becomes extinguished by amplitude death. We added a sentence describing these changes to the main (page 12; lines 389-391) and a description in Materials and Methods (page 38; lines 1232-1235).

7. The new findings on the spatial periodicity of NGN2 (Fig 8) do not really shed new light on any of the issues raised by the referees but rather bring about a new layer of complexity as it is interpreted to suggest the existence of two spatial oscillators super-imposed on each other. I would recommend to leave this set of new observations for a future study.

Yes, we agree that the spatial periodicity of NGN2 brings about a new layer of complexity. However, we have decided to keep it in the paper because it was a direct question from reviewers 2 and 3 in the first round of review.

Reviewer #2:

The authors provided an updated manuscript, where significant effort has been put to improve it. They have addressed the majority of point raised. More specifically:

- The measured fixed size of the microcluster size at different stages of development indicates that they are unlikely to be just a product of clonal expansion
- Additional data on Ngn2 and Dll1/Jag1 expression enrich the knowledge about microclusters. Although their expression patterns are not always easy to reconcile, they are interesting findings for future investigation. For example it could be examined why only 1 cell of the cluster expresses Ngn2 and whether the position of this cell is stereotypical in a cluster...
- The clarifications about the meaning of coupling strength are useful, but highlight the significant lack of knowledge that the Hill functions encapsulates in a black box.
- All the minor points I raised have been adequately addressed.

From the additions, I found particularly illuminating and important the lines 246-263, while the added section in lines 310-350 is more difficult to follow. The authors assign a "longer-term" trend in the Venus::Hes5 level, which in fact implies the existence of a second periodicity (longer period thus "longer-term" trend) on top of the described 3-4hrs. It may have been helpful to build this argument with a cleaner description of what this "long-term" trend actually is.

This issue of the longer-term trend in Venus::HES5 and how it relates to microcluster persistence was also picked up by Reviewer 1. Indeed, the single-cell longer-term trends and transitions in microclusters from HES5 hi-lo-high states suggests the

existence of a second periodicity. We have now clarified in the text that single cell Venus::HES5 expression is a composite of 1. low amplitude, short periodicity, noisy, oscillations and 2. larger amplitude and slower varying fluctuations in mean expression. Nested oscillations, where oscillations of different timescales are superimposed, are present in biology including neuronal oscillations of the brain and we included a discussion of this including references in the manuscript (page 21; lines 735-738). However our movies are not long enough to show whether the single cell long-term trends in Venus::HES5 are periodic and so we cannot definitively prove a second periodicity.

On your suggestion we have re-written and clarified the text (pages 10-11; lines 318-365) to provide a clearer description of the longer-term trend in the single cell Venus::HES5 level and have renamed the trend "slow varying fluctuations" to emphasise their non-linearity. Further we have clarified that the time-scale of this slow varying trend matches the persistence time of the microclusters (Figure 3e) and has greater amplitude than the 3-4hrs oscillations (new Appendix Figure S4a), We have added a new figure to provide a visual explanation (new Figure 3k).

Overall, the revised manuscript addresses the great majority of deficiencies, although at times it leaves more open questions than satisfactory explanations. Nevertheless, the findings presented are substantiated and I believe merit publication in Molecular Systems Biology. I expect this line of research will be followed.

Additional comments by reviewer #2 after cross-commenting:

I see the value of many of the points referee 1 raises in the renewed revision. In particular, the idea that the manuscript is "overall improved, mostly through clarifications" somewhat echoes my comment that it leaves more open questions than satisfactory explanations. I would like to comment on some aspects of his/her critique:

1. He/She was not convinced of the existence of microclusters. From my point of view this was not a major issue at the beginning, and while I agree that some aspects could be done better (I agree with the importance of measuring the correlation coefficient beyond the 4th neighbour, expectation being it will drop) there are supporting evidence for the microcluster's presence and relevance. More specifically, the correlation with NGN2 expression patterns adds on their credibility. Thus, I would think including these data is important against point 7 suggestion.
2. I found the addition of the single cell tracking data a significant improvement. In fact, such experiments are quite demanding from a technical point of view, and thus I anticipate that only a selection of pairs are admissible for analysis. As long as the criteria for selecting the pairs are clearly stated, I wouldn't see any further problems.
4. This is the point with which I agree most with reviewer 1, as it resonates with my critique on the section 310-350. I believe the argument of the authors boils down to the existence of a second periodicity at 6-8 hrs (interestingly a harmonic) that is more evident due to biological (or technical?) reasons at the population level. I think the authors should be given the opportunity to further clarify/explain this.

Overall, I would recommend giving the authors another chance to address these

issues. While the picture emerging from the work is somewhat blurry, I think the results would be of value to the community and they suggest new avenues of research.

Thank you for taking the time to cross-comment, we appreciate your efforts.

Thank you again for sending us your revised manuscript. We think that the performed revisions have satisfactorily addressed the remaining concerns of the reviewers. I am therefore pleased to inform you that your paper has been accepted for publication.

Corresponding Author Name: Nancy Papalopulu and Cerys Manning

Manuscript Number: MSB-20-9902